# BEARD: BENCHMARKING THE ADVERSARIAL ROBUSTNESS FOR DATASET DISTILLATION

## ABSTRACT

Dataset Distillation (DD) compresses large-scale datasets into smaller synthesized datasets, enabling efficient model training while preserving high test performance. However, existing DD methods primarily focus on accuracy and largely neglect adversarial robustness, potentially exposing models to security risks in critical applications. Evaluating robustness is therefore essential but remains challenging due to complex interactions among distillation methods, model architectures, and attack strategies. Moreover, current benchmarks provide only partial coverage and lack a unified perspective in the DD domain. To address this gap, we introduce BEARD, an open and unified benchmark for systematically evaluating the adversarial robustness of models trained on distilled datasets from representative DD methods such as DM, IDM, and BACON. BEARD supports diverse adversarial attacks, including FGSM, PGD, and C&W, and widely used datasets such as CIFAR-10/100 and Tiny-ImageNet. Using an *adversarial game framework*, we define three key metrics: Robustness Ratio (RR), Attack Efficiency Ratio (AE), and Comprehensive Robustness-Efficiency Index (CREI). We conduct systematic evaluations and analyses across unified benchmarks, varying images-per-class (IPC) settings and adversarial training strategies, showing that dataset distillation consistently enhances adversarial robustness, with adversarial training providing further improvements. The leaderboard is available at `https://beard-leaderboard.github.io/`, along with a library of model and dataset pools to support reproducible research. Code is accessible at `https://anonymous.4open.science/r/BEARD-6B8A/`.

## 1 INTRODUCTION

Deep Neural Networks (DNNs) (LeCun et al., 2015) have revolutionized various domains by leveraging large-scale datasets to learn sophisticated representations for specific tasks (Krizhevsky et al., 2012; Vaswani et al., 2017). However, training on massive datasets involves substantial computational overhead and large storage requirements, which pose practical challenges, especially in resource-constrained environments (Yu et al., 2023; Lei & Tao, 2023). These challenges motivate methods that improve training efficiency and reduce memory usage while preserving model performance.

Dataset Distillation (DD) (Wang et al., 2018; Zhao et al., 2020) offers a promising approach to mitigating the computational and storage challenges of training on large datasets by compressing them into smaller, synthetic subsets, thereby improving training efficiency and reducing memory usage. This approach is particularly valuable in resource-constrained scenarios where training on full datasets is impractical (Li et al., 2020; Yang et al., 2023). Considerable progress in DD has been driven by a range of algorithmic innovations, which can be broadly categorized into Meta-Model Matching (e.g., DD (Wang et al., 2018), KIP (Nguyen et al., 2020; 2021)), Gradient Matching (e.g., DC (Zhao et al., 2020), TESLA (Cui et al., 2023), FTD (Du et al., 2023)), and Distribution Matching (e.g., DM (Zhao & Bilen, 2023), CAFE (Wang et al., 2022), BACON (Zhou et al., 2024a)).

Despite the efficiency and memory advantages of dataset distillation, models trained on distilled datasets remain highly vulnerable to adversarial attacks, which are small, deliberately crafted, and human-imperceptible perturbations capable of misleading classifiers (Szegedy et al., 2014; Goodfellow et al., 2015a; Madry et al., 2018a; Zhou et al., 2022), as illustrated in Figure 1. These vulnerabilities threaten the reliability of security-critical applications such as face recognition (Wei et al., 2022a;b), object detection (Zhou et al., 2024b; Hu et al., 2021), and autonomous driving (Wang

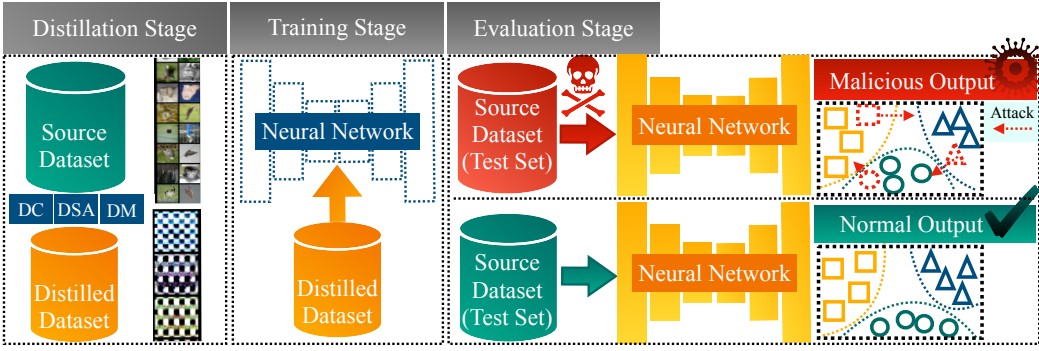

Figure 1: Evaluation of adversarial robustness for dataset distillation. The process has three stages: (1) *Distillation Stage*: DD methods such as DC (Zhao et al., 2020), DSA (Zhao & Bilen, 2021), and DM (Zhao & Bilen, 2023) are used to generate distilled datasets; (2) *Training Stage*: models are trained on these datasets; (3) *Evaluation Stage*: targeted and untargeted attacks (e.g., FGSM (Goodfellow et al., 2015b), PGD (Madry et al., 2018b), C&W (Carlini & Wagner, 2017)) are applied to test sets of standard datasets (CIFAR-10/100, TinyImageNet (Krizhevsky, 2009; Deng et al., 2009)). Performance under clean and adversarial conditions is measured using dedicated robustness metrics.

et al., 2021; Yuan et al., 2023), demonstrating that gains in training efficiency do not guarantee adversarial robustness. This highlights the need for principled frameworks and evaluation protocols to systematically assess and enhance the adversarial robustness of models trained on distilled datasets.

**Research Gap.** Although recent studies (Xue et al., 2025; Ma et al., 2025; Chen et al., 2023) have begun exploring ways to improve the adversarial robustness of DD, they do not provide a systematic framework for evaluating the robustness of models trained on distilled datasets. Assessing robustness remains challenging due to the complex interactions among distillation methods, model architectures, and attack strategies. Existing benchmarks offer only partial solutions. Specifically, DD-RobustBench (Wu et al., 2024) evaluates each IPC setting independently without providing a unified perspective, whereas RobustBench (Croce et al., 2021) primarily focuses on conventional models rather than DD. Collectively, these limitations underscore the need for a principled framework and specialized metrics explicitly tailored for DD, enabling comprehensive and unified assessment of adversarial robustness.

To address this gap, we introduce `BEARD`, an open and unified benchmark designed to systematically evaluate the adversarial robustness of models trained using DD methods. We conduct extensive evaluations using various representative DD techniques, including DC (Zhao et al., 2020), DSA (Zhao & Bilen, 2021), DM (Zhao & Bilen, 2023), MTT (Cazenavette et al., 2022), IDM (Zhao et al., 2023), and BACON (Zhou et al., 2024a), across a range of datasets, such as TinyImageNet (Deng et al., 2009) and CIFAR-10/100 (Krizhevsky, 2009). To assess robustness, we incorporate a broad spectrum of attack methods, including FGSM (Goodfellow et al., 2015b), PGD (Madry et al., 2018b), C&W (Carlini & Wagner, 2017), DeepFool (Moosavi-Dezfooli et al., 2016), and AutoAttack (Croce & Hein, 2020). Using the *adversarial game framework*, which provides a unified perspective for evaluating the adversarial robustness of DD methods across diverse IPC settings and attack strategies, we further introduce three key evaluation metrics: Robustness Ratio (RR), Attack Efficiency Ratio (AE), and Comprehensive Robustness-Efficiency Index (CREI). Additionally, we develop a straightforward evaluation protocol with a Dataset Pool and a Model Pool. Through large-scale experiments, we evaluate the adversarial robustness of models trained using DD under targeted and untargeted attacks with `BEARD`. Analysis of the results demonstrates that dataset distillation improves robustness across unified benchmarks and diverse IPC settings, with adversarial training providing additional gains.

Our **contributions** are summarized as follows:

- We present `BEARD`, a unified benchmark for evaluating the adversarial robustness of models trained on distilled datasets, built upon an *adversarial game framework* to systematically assess dataset distillation methods under diverse attack scenarios.
- We introduce novel robustness metrics and a leaderboard that ranks existing dataset distillation methods based on their performance against various adversarial attacks.

- We release open-source code with detailed documentation, along with extensible Model and Dataset Pools to support reproducible and standardized robustness evaluations.

- We perform an extensive comparative analysis of benchmark results, providing practical insights and recommendations for improving adversarial robustness in dataset distillation.

## 2 RELATED WORK

**Dataset Distillation.** Dataset distillation aims to synthesize compact datasets that preserve the essential information of large-scale datasets. Wang et al. (2018) pioneered a bi-level optimization approach, modeling network parameters based on synthetic data, though it incurs high computational costs. To address this, Zhao et al. (2020) proposed Dataset Condensation (DC), a gradient matching method aligning gradients from original and synthetic datasets, while DSA (Zhao & Bilen, 2021) enhances it with data augmentation to produce more informative synthetic images and improve training performance. MTT (Cazenavette et al., 2022) mimics long-range training dynamics, while distribution matching methods such as DM (Zhao & Bilen, 2023) and IDM (Zhao et al., 2023) leverage distributional similarity metrics. BACON (Zhou et al., 2024a) employs a Bayesian framework to improve efficiency. Additional directions include multi-size distillation (He et al., 2024), trajectory-based lossless distillation (Guo et al., 2024), and SRe2L (Yin et al., 2024), RDED (Sun et al., 2024). DC-Bench (Cui et al., 2022) provides an initial benchmark for DD performance evaluation. Despite these advances, existing methods primarily emphasize training efficiency and accuracy, and a systematic evaluation of adversarial robustness remains largely unexplored.

**Adversarial Dataset Distillation.** Adversarial Robust Distillation (ARD) (Goldblum et al., 2020) demonstrated that robustness can be transferred from teacher to student within the framework of knowledge distillation. Building on the notion of robust features (Ilyas et al., 2019), Wu et al. (2022) proposed constructing robust datasets, such that classifiers trained on these datasets exhibit inherent adversarial robustness. Subsequent studies have investigated adversarial robustness specifically in the context of dataset distillation, with Ma et al. (2025) examining the efficiency and reliability of DD tasks using TrustDD, Chen et al. (2023) analyzing associated security risks, and Xue et al. (2025) exploring approaches to embed adversarial robustness in distilled datasets to enhance robustness without compromising accuracy. Nevertheless, these investigations typically focus on specific attack types or experimental settings and do not provide a unified framework for systematically comparing adversarial robustness across diverse distilled datasets, IPC settings, and attack scenarios.

**Adversarial Robustness Benchmarks.** Existing benchmarks provide limited insights into the adversarial robustness of dataset distillation, due to the challenges of systematically evaluating interactions among distillation methods, model architectures, and attack strategies. Wu et al. (2024) proposed DD-RobustBench as an initial framework, evaluating multiple IPC settings independently (e.g., IPC-1, IPC-10, IPC-50). While it offers useful initial analysis, it lacks a unified perspective across IPCs and does not explicitly account for attack efficiency, which is particularly relevant in resource-constrained environments such as edge computing. In terms of attack strategies, it primarily considers white-box untargeted attacks, leaving black-box and targeted attacks less explored. Moreover, the benchmark does not provide a leaderboard for straightforward comparison across methods. Prior to DD-RobustBench, Croce et al. (2021) proposed RobustBench, which focuses on conventional models and provides comprehensive attack evaluations with standardized metrics, but it does not specifically address the unique challenges of dataset distillation, such as multi-IPC evaluation or efficiency considerations with condensed datasets.

Building on these considerations, BEARD was developed as an open and unified benchmark specifically designed to systematically evaluate the adversarial robustness of dataset distillation methods across multiple datasets. Inspired by Dai et al. (2023), an *adversarial game framework* is employed to unify the evaluation process and three key metrics are introduced: Robustness Ratio (RR), Attack Efficiency Ratio (AE), and Comprehensive Robustness-Efficiency Index (CREI). These metrics provide a unified perspective for assessing models trained on distilled datasets across different IPC settings, capturing both attack effectiveness and attack efficiency. The overall evaluation is summarized by the CREI, with individual metrics such as RR for attack effectiveness and AE for attack efficiency. Multiple leaderboards are also constructed to offer an intuitive display of adversarial robustness. A detailed comparison between BEARD and existing benchmarks is provided in Appendix B.5.

# 3 ADVERSARIAL ROBUSTNESS FOR DATASET DISTILLATION AGAINST MULTIPLE ATTACKS

## 3.1 PRELIMINARY

**Motivation.** Adversarial robustness in dataset distillation has received limited systematic evaluation. Prior studies (Wang et al., 2018; Zhao et al., 2020) focused on model accuracy, offering only a limited view of robustness. Subsequent work (Wu et al., 2024) considered robustness evaluation but lacked a unified perspective across IPC settings and largely overlooked attack efficiency. Another study (Croce et al., 2021) proposed comprehensive robustness benchmarks but did not specifically address the unique challenges of dataset distillation. Moreover, robustness evaluation is further complicated by the interactions among distillation methods, model architectures, and diverse attack strategies. To address these challenges, we introduce a unified *adversarial game framework* that explicitly considers both attack effectiveness and efficiency, enabling a comprehensive assessment of how distilled datasets perform under various adversarial conditions. This framework provides a principled foundation for designing specialized metrics and leaderboards that systematically compare the robustness of different dataset distillation methods across multiple datasets and IPC settings.

**Notations.** Consider a large dataset $\mathcal{T} = \{(x_i, y_i)\}_{i=1}^N$, where $x_i \in \mathcal{X} \subseteq \mathbb{R}^d$ denotes input samples and $y_i \in \mathcal{Y} = \{1, \ldots, C\}$ denotes corresponding labels. DD aims to generate a synthetic dataset $\mathcal{S} = \{(\tilde{x}_j, \tilde{y}_j)\}_{j=1}^M \subseteq \mathcal{X} \times \mathcal{Y}$, with $M \ll N$, such that a model $m \in \mathcal{M}$, $m : \mathcal{X} \to \mathcal{Y}$, trained on $\mathcal{S}$ performs comparably to one trained on $\mathcal{T}$. The defender function $\mathcal{D}$ encompasses DD methods with diverse IPC settings, outputting $\mathcal{S}$. The attacker function set $\mathcal{A}$ generates adversarial examples with perturbation budget $\epsilon \in \mathcal{P}$, where $\mathcal{P} = \{\epsilon \mid \epsilon \geq 0\}$ defines allowable perturbation sizes.

## 3.2 A UNIFIED ADVERSARIAL GAME FRAMEWORK FOR EVALUATING ADVERSARIAL ROBUSTNESS IN DATASET DISTILLATION

**Definition 3.1** (Attacker Function). Let $\mathcal{L} : \mathcal{Y} \times \mathcal{Y} \to \mathbb{R}$ be a loss function (e.g., cross-entropy). An attacker function $a \in \mathcal{A}$ maps an input-label pair $(x, y) \in \mathcal{X} \times \mathcal{Y}$ and a model $m \in \mathcal{M}$ to an adversarial example $\hat{x} = a(x, y, m)$ that maximizes the loss under an $L_p$-bounded perturbation:

$$a(x, y, m) = \arg \max_{\substack{\hat{x} \in \mathcal{X} \\ \|\hat{x} - x\|_p \leq \epsilon}} \mathcal{L}(m(\hat{x}), y). \tag{1}$$

Assuming $\mathcal{X}$ is bounded and $\mathcal{L}$ is continuous, the constraint set $\{\hat{x} \in \mathcal{X} : \|\hat{x} - x\|_p \leq \epsilon\}$ is compact, ensuring the maximizer exists. In practice, $a(x, y, m)$ is approximated by standard attacks (Goodfellow et al., 2015b; Madry et al., 2018b).

**Definition 3.2** (Defender Function). Let $\mathcal{L} : \mathcal{Y} \times \mathcal{Y} \to \mathbb{R}$ be a loss function. A defender function $d \in \mathcal{D}$ aims to synthesize a distilled dataset $\mathcal{S}$ to minimize the worst-case adversarial risk across a set of attack functions $\mathcal{A}$. This leads to a bi-level optimization problem:

$$d(\mathcal{A}) = \arg \min_{\mathcal{S} \subseteq \mathcal{X} \times \mathcal{Y}} \max_{a \in \mathcal{A}} \mathbb{E}_{(x,y) \sim \mathcal{T}} \mathcal{L}(m(a(x, y, m)), y),$$

$$\text{where} \quad m = \arg \min_{m \in \mathcal{M}} \mathbb{E}_{(\tilde{x}, \tilde{y}) \sim \mathcal{S}} \mathcal{L}(m(\tilde{x}), \tilde{y}). \tag{2}$$

The inner optimization trains a defended model $m$ on the synthetic dataset $\mathcal{S}$, while the outer optimization aims to enhance robustness against worst-case attacks.

**Definition 3.3** (Attack Success Rate (ASR)). Let $(x, y) \in (\mathcal{X}, \mathcal{Y}) \sim \mathcal{T}$ be an input-label pair, $a \in \mathcal{A}$ an adversarial attack, and $m \in \mathcal{M}$ a defended model. The attack success rate is the probability that the model correctly classifies the original input but misclassifies the adversarially perturbed input:

$$\mathcal{ASR}(m; a) = \mathbb{E}_{(x,y) \sim \mathcal{T}} \mathbf{1}\{m(a(x, y, m)) \neq y \wedge m(x) = y\}, \tag{3}$$

where $\mathbf{1}\{\cdot\}$ is the indicator function, which is 1 if the model misclassifies the perturbed input and 0 otherwise. The ASR quantifies vulnerability, with higher values indicating greater susceptibility.

**Definition 3.4** (Attack Success Time (AST)). Let $(x, y) \in (\mathcal{X}, \mathcal{Y}) \sim \mathcal{T}$ be an input-label pair, $a \in \mathcal{A}$ an adversarial attack, and $m \in \mathcal{M}$ a defended model trained via dataset distillation. Let $t$ denote the GPU time taken by $a$ to generate a perturbed input $a(x, y, m)$ such that $m(a(x, y, m)) \neq y$. The attack success time is defined as the expected time required to successfully fool the model:

$$\mathcal{AST}(m; a) = \mathbb{E}_{(x,y) \sim \mathcal{T}} [t \mid m(a(x, y, m)) \neq y]. \tag{4}$$

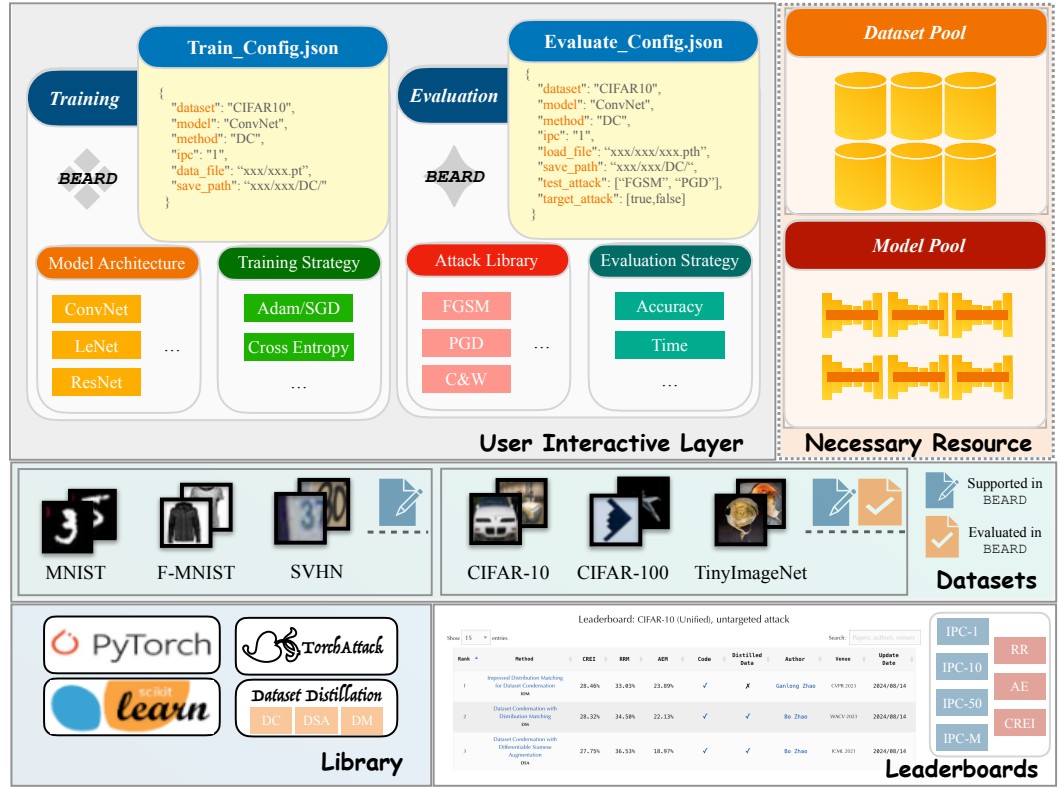

Figure 2: Illustration of BEARD. A distilled dataset pool is first generated from source datasets using various dataset distillation methods, including DC (Zhao et al., 2020), DSA (Zhao & Bilen, 2021), DM (Zhao & Bilen, 2023), IDM (Zhao et al., 2023), and BACON (Zhou et al., 2024a). Neural networks are then trained on these distilled datasets to form a model pool of pretrained models. The adversarial robustness of these models is subsequently evaluated using a variety of attack methods, such as FGSM (Goodfellow et al., 2015b), PGD (Madry et al., 2018b), C&W (Carlini & Wagner, 2017), DeepFool (Moosavi-Dezfooli et al., 2016), and AutoAttack (Croce & Hein, 2020). Leaderboards are created to provide an intuitive comparison of robustness across different methods and datasets.

**Definition 3.5** (Adversarial Game Framework). Given conceptual thresholds $\gamma$ and $\beta$, which define the conditions for evaluating attack success rate and attack success time, respectively, and a set $\mathcal{A}$ of perturbation functions that may occur during test-time, the performance of the model is evaluated based on its ASR and AST under these perturbations. The model is considered robust if:

$$\frac{\mathbb{E}_{m\in\mathcal{M}}\mathbb{E}_{a\in\mathcal{A}}\mathcal{ASR}(m;a)}{\max\limits_{m^*\in\mathcal{M},a^*\in\mathcal{A}}\mathcal{ASR}(m^*;a^*)} \le \gamma \quad \text{and} \quad \frac{\mathbb{E}_{m\in\mathcal{M}}\mathbb{E}_{a\in\mathcal{A}}\mathcal{AST}(m;a)}{\max\limits_{m^*\in\mathcal{M},a^*\in\mathcal{A}}\mathcal{AST}(m^*;a^*)} \ge \beta. \tag{5}$$

Here, $\gamma$ and $\beta$ are conceptual thresholds specifying conditions for the defender's victory, requiring a low ASR and a high AST, respectively. They are not assigned specific values in practice but serve to formalize the adversarial game dynamics and define robustness criteria.

*Remark* 3.6. In the *adversarial game framework*, the defender wins if two conditions are met: (1) the attack success rate $\mathcal{ASR}(m;a)$ is minimized below the threshold $\gamma$, and (2) the attack success time $\mathcal{AST}(m;a)$ is maximized above the threshold $\beta$. If either condition fails, the attacker wins. A win for the defender indicates effective robustness against adversarial perturbations, while a win for the attacker reveals vulnerabilities that need addressing. This game is defined over a set of models $m \in \mathcal{M}$, each trained on distinct distilled datasets $(\tilde{x}, \tilde{y}) \in (\tilde{\mathcal{X}}, \tilde{\mathcal{Y}}) \subseteq \mathcal{S}$, and subjected to various attacks $a \in \mathcal{A}$. When a specific model $m$ or attack $a$ is chosen, the multi-adversary game simplifies to a single-adversary game focused on their interaction. The numerator represents the average performance across all model-attack pairs and the denominator the worst-case scenario, providing a fair basis for comparison across different IPC settings and attacks.

## 3.3 METRIC FOR EVALUATING ADVERSARIAL ROBUSTNESS

Building on the Definition 3.5 and Remark 3.6, we propose metrics that aggregate accuracy across both single and multiple attacks, as well as models trained on different IPC distilled datasets. This section introduces two key criteria for evaluating adversarial robustness: (1) attack effectiveness and (2) attack efficiency, with their respective metrics.

**Definition 3.7** (Robustness Ratio (RR)). Given a neural network model $m \in \mathcal{M}$ with defensive distillation $d \in \mathcal{D}$ and an adversarial attack function $a \in \mathcal{A}$, the robustness ratio is defined as:

$$\mathrm{RR}(m; a) = 100 \times \left[ 1 - \frac{\mathbb{E}_{m \in \mathcal{M}} \mathbb{E}_{a \in \mathcal{A}} \mathcal{ASR}(m; a)}{\max\limits_{m \in \mathcal{M}, a \in \mathcal{A}} \mathcal{ASR}(m; a)} \right]. \tag{6}$$

*Remark* 3.8. The purpose of using "$1-$" in the formula is to emphasize model robustness rather than attack success. A higher attack success rate indicates a more effective attack but a less robust model. Therefore, by subtracting the normalized attack success rate from 1, the formula inversely represents robustness. This way, when ASR is high, the robustness ratio will be low, and when ASR is low, the model is considered more robust. The formula also normalizes the ASR by dividing it by the maximum possible ASR to provide a standardized measure of robustness.

**Definition 3.9** (Attack Efficiency Ratio (AE)). Given a neural network model $m \in \mathcal{M}$ with defensive distillation $d \in \mathcal{D}$ and an adversarial attack function $a \in \mathcal{A}$, the attack efficiency ratio is defined as:

$$\mathrm{AE}(m; a) = 100 \times \left[ \frac{\mathbb{E}_{m \in \mathcal{M}} \mathbb{E}_{a \in \mathcal{A}} \mathcal{AST}(m; a)}{\max\limits_{m \in \mathcal{M}, a \in \mathcal{A}} \mathcal{AST}(m; a)} \right]. \tag{7}$$

**Definition 3.10** (Comprehensive Robustness-Efficiency Index (CREI)). Given the robustness ratio RR and the attack efficiency ratio AE, with an adjustable coefficient $\alpha$, the comprehensive robustness-efficiency index combines them into a unified metric:

$$\mathrm{CREI} = \alpha \times \mathrm{RR} + (1 - \alpha) \times \mathrm{AE}, \tag{8}$$

where $\alpha$ controls the trade-off between robustness and efficiency. In our experiments, we set $\alpha = 0.5$.

*Remark* 3.11. The *adversarial game framework* can shift between multi-adversary and single-adversary scenarios, where "single-adversary" refers to a model facing one attack strategy, while "multi-adversary" involves multiple attack strategies. In this context, the metrics adjust: Robustness Ratio (RR) and Attack Efficiency (AE) become Single-Adversary Robustness Ratio (RRS) and Single-Adversary Attack Efficiency Ratio (AES) for single-adversary situations, and Multi-Adversary Robustness Ratio (RRM) and Multi-Adversary Attack Efficiency Ratio (AEM) for multi-adversary contexts. The defender aims to minimize the attack success rate, which aligns with maximizing RR, while optimizing AE corresponds to maximizing attack success time. Conversely, the attacker seeks to maximize AE and minimize RR. Analyzing these metrics enables a clearer assessment of dataset distillation, capturing both the robustness of models and the efficiency of attacks. These metrics are directly evaluated in our experiments (Section 5) to quantify how different DD methods, IPCs, and attack strategies affect model robustness and attack efficiency.

# 4 ADVERSARIAL ROBUSTNESS BENCHMARK FOR DATASET DISTILLATION

## 4.1 OVERVIEW OF BEARD

BEARD consists of two main stages: the *Training Stage* and the *Evaluation Stage*, as illustrated in Figure 2. In the training stage (Section 4.1.1), models are trained on datasets from the dataset pool. The evaluation stage (Section 4.1.2) involves applying adversarial perturbations to test images from an attack library to assess model robustness. The benchmark comprises three key components: *Dataset Pool*, *Model Pool*, and *Evaluation Metrics*. More details are provided in Appendix A.

### 4.1.1 TRAINING STAGE

In the training stage, we primarily focus on CIFAR-10 (Krizhevsky, 2009), CIFAR-100 (Krizhevsky, 2009), and TinyImageNet (Deng et al., 2009) due to their widespread use and diverse performance

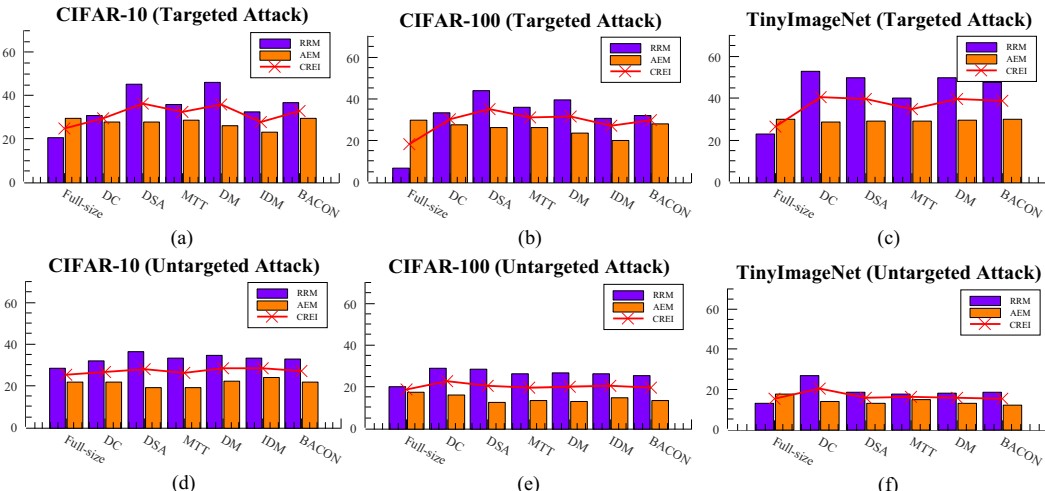

Figure 3: Performance of various dataset distillation methods under targeted and untargeted adversarial attacks on CIFAR-10, CIFAR-100, and TinyImageNet. The first row depicts targeted attacks with unified IPC settings, while the second row shows performance under untargeted attacks. Metrics used include Multi-Adversary Robustness Ratio (RRM), Multi-Adversary Attack Efficiency Ratio (AEM), and Comprehensive Robustness-Efficiency Index (CREI).

in dataset distillation. Additionally, we extend our experiments to include MNIST (LeCun et al., 1998), FashionMNIST (Xiao et al., 2017), and SVHN (Netzer et al., 2011) to further evaluate the adversarial robustness of dataset distillation methods on simpler datasets. We evaluate six prominent dataset distillation methods: DC (Zhao et al., 2020), DSA (Zhao & Bilen, 2021), DM (Zhao & Bilen, 2023), MTT (Cazenavette et al., 2022), IDM (Zhao et al., 2023), and BACON (Zhou et al., 2024a), which represent different approaches, including gradient matching (Zhao et al., 2020; Zhao & Bilen, 2021), distribution matching (Zhao & Bilen, 2023; Zhao et al., 2023; Zhou et al., 2024a), and trajectory matching (Cazenavette et al., 2022). Synthetic datasets are generated using IPC-1, IPC-10, and IPC-50 settings, ensuring consistency with DC-bench (Cui et al., 2022) for hyperparameters. These datasets, primarily sourced from existing *open-source* distilled datasets, are produced by the six dataset distillation methods across various IPC settings and form the **Dataset Pool**. This pool is critical for evaluating the performance of different dataset distillation methods and ensures a comprehensive comparison across various distillation approaches.

### 4.1.2 EVALUATION STAGE

In the evaluation stage, the **Model Pool** repository is utilized to streamline the assessment of robust models trained on distilled datasets. By integrating metrics derived from the *adversarial game framework*, including RR, AE, and CREI, this evaluation can more effectively measure the models' robustness against adversarial attacks within the competitive dynamics of the game setting. The repository facilitates the analysis of model performance and broader trends by consolidating checkpoints from various sources. However, challenges arise in unifying these models due to differing architectures and normalization techniques. After generating distilled datasets from the dataset pool, multiple models are trained from scratch using various distillation methods, IPC settings, and the Adam optimizer for 1,000 epochs. The models with the highest validation accuracy are selected and added to the model pool. Adversarial robustness is assessed using a diverse attack library compatible with *torchattacks* (Kim, 2020), including methods like FGSM (Goodfellow et al., 2015b), PGD (Madry et al., 2018b), C&W (Carlini & Wagner, 2017), DeepFool (Moosavi-Dezfooli et al., 2016), and AutoAttack (Croce & Hein, 2020). Both targeted and untargeted attacks are conducted with a uniform perturbation budget of $\epsilon = \frac{8}{255}$ for most methods, with exceptions for DeepFool and C&W.

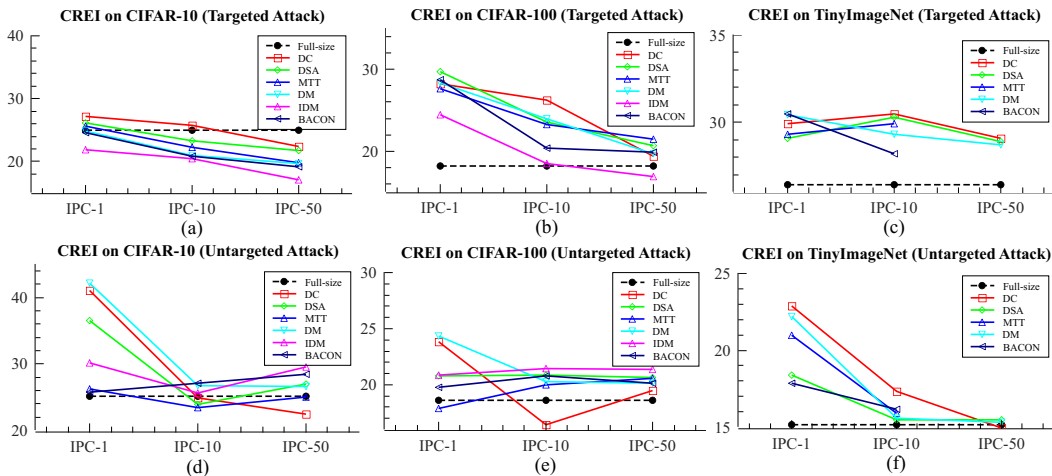

Figure 4: CREI trends under targeted and untargeted attacks across three datasets: CIFAR-10, CIFAR-100, and TinyImageNet. The x-axis represents the number of IPC, while the y-axis displays CREI values. Six DD methods (DC, DSA, MTT, DM, IDM, BACON) are compared to full-size datasets at IPC-1, IPC-10, and IPC-50, highlighting their robustness and efficiency across various attacks.

## 4.2 LEADERBOARDS

We provide 12 leaderboards for CIFAR-10 (Krizhevsky, 2009), CIFAR-100 (Krizhevsky, 2009), and TinyImageNet (Deng et al., 2009), covering IPC-1, IPC-10, and IPC-50 settings. These leaderboards rank methods based on robustness and efficiency metrics, including RR, AE, and CREI. The leaderboard evaluates six DD methods: DC (Zhao et al., 2020), DSA (Zhao & Bilen, 2021), DM (Zhao & Bilen, 2023), MTT (Cazenavette et al., 2022), IDM (Zhao et al., 2023), and BACON (Zhou et al., 2024a). In terms of adversarial attacks, the leaderboards integrate various methods such as FGSM (Goodfellow et al., 2015b), PGD (Madry et al., 2018b), C&W (Carlini & Wagner, 2017), DeepFool (Moosavi-Dezfooli et al., 2016), and AutoAttack (Croce & Hein, 2020), all compatible with *torchattacks* (Kim, 2020). Evaluating adversarial robustness is challenging due to the diversity of settings and attack types, with no unified framework available. As illustrated in Figure 2, our leaderboards fill this gap by offering a unified evaluation of adversarial robustness in dataset distillation.

## 5 ANALYSIS

### 5.1 ROBUSTNESS EVALUATION USING PROPOSED METRICS

The results in Figure 3 show that models trained on synthetic datasets generated by DD methods achieve higher Multi-Adversary Robustness Ratio (RRM) under both targeted and untargeted attacks, indicating improved adversarial robustness. Among these DD methods, DSA exhibited the best or second-best robustness under both attack types, while DC outperformed most other methods such as MTT, DM, IDM, and BACON in most cases. Moreover, as the dataset size increased, adversarial robustness improved under targeted attacks but decreased under untargeted attacks. Regarding efficiency, models trained on full-size datasets show higher Multi-Adversary Attack Efficiency Ratio (AEM), meaning attacks require more effort and time, reflecting robustness from the efficiency perspective. The Comprehensive Robustness-Efficiency Index (CREI), which combines RRM and AEM, provides a unified measure of both adversarial robustness and attack efficiency, highlighting that DD methods, particularly DSA and DC achieve a more balanced performance. We further examine the trade-off between robustness and clean accuracy, which arises from mechanisms similar to adversarial training. Additional details are in Appendices B.1 and B.6.

To further understand these robustness patterns, we analyze the characteristics of different DD methods. DSA and DC, as representative gradient-matching methods, predominantly capture early-stage features during dataset distillation. Compared to distribution-matching methods such as DM, IDM, and BACON, gradient-matching methods capture more information from the data, which

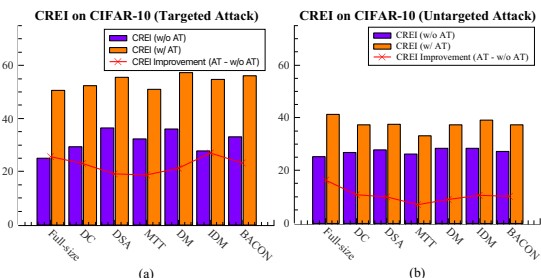

(a)          (b)

Figure 5: Illustration of CREI trends on CIFAR-10 under targeted attacks with (w/) or without (w/o) Adversarial Training (AT). The x-axis shows DD methods (Full-size, DC, DSA, MTT, DM, IDM, BACON) under unified IPC, while the y-axis displays CREI values measuring adversarial robustness. CREI improvement indicates the difference between models with and without AT.

Table 1: Robustness evaluation of dataset distillation methods with and without adversarial training on CIFAR-100 under targeted attacks (%).

| Method | RR | AE | CREI |
|---|---|---|---|
| DM | 9.38 | 30.02 | 19.7 |
| DM+AT | 33.98 | 30.3 | 32.14 |
| IDM | 11.6 | 22.13 | 16.86 |
| IDM+AT | 97.69 | 28.03 | 62.86 |
| BACON | 8.99 | 30.73 | 19.86 |
| BACON+AT | 87.6 | 30.29 | 58.94 |

facilitates the learning of robust features, as highlighted in Ilyas et al. (2019), and thereby enhances adversarial robustness. Moreover, relative to trajectory-matching methods like MTT, which learn features over the entire training trajectory, gradient-matching methods emphasize early-stage features, which have been shown to contribute more substantially to adversarial robustness (Zhang et al., 2020). Furthermore, the impact of dataset size on robustness further reveals the interplay between local and global robustness. Targeted attacks mainly evaluate local robustness, which benefits from the smoother decision boundaries learned from larger distilled datasets. In contrast, untargeted attacks reflect global robustness, which may decline as increasing data complexity introduces more globally vulnerable directions in the input space.

## 5.2 ROBUSTNESS EVALUATION WITH DIVERSE IPCS

We identify two key observations from Figure 4. First, increasing IPC generally decreases adversarial robustness, as reflected by lower CREI values. Second, enlarging the dataset scale improves robustness under targeted attacks but reduces it under untargeted attacks, consistent with the patterns observed in Figure 3. Among all methods, DC and DSA maintain strong robustness across IPC settings, outperforming most other DD approaches in the majority of cases, although their performance becomes slightly inferior to DM, IDM, and BACON under the IPC-50 configuration. These trends hold consistently across CIFAR-10, CIFAR-100, and TinyImageNet, demonstrating the robustness and generalizability of DD methods. Collectively, these findings highlight the complex relationship among IPC, dataset size, and adversarial robustness, affirming the effectiveness of specific DD methods in enhancing model robustness under various scenarios. Further details on robustness under multiple attacks with varying IPCs are provided in Appendix B.2, while single-attack robustness under different perturbation budgets is discussed in Appendix B.4.

To interpret these trends, it is useful to examine how different DD methods leverage data features and gradients. Increasing IPC provides more features per class, enriching global representations and slightly benefiting untargeted attacks, but also producing less smooth local decision boundaries that reduce performance under targeted attacks. Gradient-matching methods like DC and DSA excel at low to medium IPCs by focusing on early-stage features, yet their advantage diminishes as IPC or dataset scale grows and feature complexity increases. In contrast, distribution-matching methods such as DM, IDM, and BACON, which integrate the full data distribution, can better exploit the abundant information in high-IPC or large-dataset settings, resulting in comparatively higher adversarial performance. These patterns highlight the nuanced interplay among dataset characteristics, attack type, and distillation methodology in shaping model robustness.

## 5.3 ROBUSTNESS EVALUATION WITH ADVERSARIAL TRAINING

Figure 5 shows that Adversarial Training (AT) substantially improves model robustness against both targeted and untargeted attacks under a unified IPC in a multi-adversary setting. For targeted attacks, models with AT (orange bars) achieve higher CREI values than those without AT (purple bars), particularly for methods such as DSA and BACON (Figure 5(a)). In contrast, models trained on full-size datasets exhibit lower CREI values without AT, consistent with the trend of declining robustness as IPC increases. For untargeted attacks, full-size models benefit most from AT, while all methods experience reduced robustness without it (Figure 5(b)). Notably, the effect of AT increases with dataset scale, as illustrated by the red curve. These trends are further confirmed on CIFAR-100 (Table 1), where DM+AT, IDM+AT, and BACON+AT achieve CREI values of 32.14%, 62.86%, and 58.94%, consistently surpassing their non-AT counterparts. The results indicate that combining DD methods with AT consistently enhances robustness across attack types and dataset scales. Additional details are provided in Appendix B.3

These observations can be interpreted by considering the complementary roles of dataset distillation and adversarial training. AT explicitly smooths the decision boundaries and strengthens local robustness, which benefits targeted attacks and mitigates vulnerabilities introduced by higher IPC or complex datasets. While distilled datasets already capture essential gradient or distribution information, full-size datasets provide richer feature diversity, allowing AT to exert greater effect. Gradient-matching methods like DSA can leverage early-stage features for robustness at lower IPCs, but as dataset scale grows, AT helps both distilled and full-size models better exploit complex feature spaces. This synergy explains why AT consistently improves CREI across methods and dataset scales, highlighting the importance of aligning distillation strategy, dataset size, and adversarial training to maximize model robustness.

## 6 CONCLUSION

A standardized benchmark is crucial for advancing the evaluation of adversarial robustness in dataset distillation methods. However, existing benchmarks provide only partial coverage and lack a unified perspective for systematically evaluating adversarial robustness across different dataset distillation settings. To address this gap, we propose BEARD, an open and unified benchmark designed to assess adversarial robustness in dataset distillation. This benchmark includes a dataset pool, a model pool, and novel metrics (RR, AE, and CREI). It also provides a leaderboard that ranks models based on their performance across three standard datasets under six adversarial attacks. Currently, the leaderboard includes 18 models trained on distilled datasets from six dataset distillation methods with three IPC settings. Evaluation with BEARD shows that dataset distillation methods vary in inherent adversarial robustness, while adversarial training consistently improves CREI. The benchmark highlights the strengths and weaknesses of different dataset distillation methods, supporting the development of more robust, secure, and efficient techniques. BEARD is an *open*, *community-driven* platform that is *continuously updated* for evaluating and comparing the adversarial robustness of distilled datasets.

## ETHICS STATEMENT

This work does not involve human subjects, animals, or personally identifiable information. All datasets used are publicly available and ethically released. The study focuses on developing a benchmark for evaluating the adversarial robustness of dataset distillation methods. Potential societal risks are minimal, and measures were taken to ensure fair evaluation and prevent misuse. Specifically, the benchmark does not provide training scripts that could be directly used for malicious purposes, and all evaluation protocols are designed to avoid favoring specific methods. All authors have complied with the ICLR Code of Ethics.

## REPRODUCIBILITY STATEMENT

All results reported in this work are fully reproducible. The code, benchmark data, and scripts are publicly available at `https://anonymous.4open.science/r/BEARD-6B8A/`, and the BEARD leaderboard can be accessed at `https://beard-leaderboard.github.io`. Detailed descriptions of the experimental setup, hyperparameters, and attack implementations are provided in Appendix A, followed by the definitions and theoretical derivations of the evaluation

metrics (RR, AE, and CREI) in Section 3 of the main text, enabling the use of the benchmark for evaluating the adversarial robustness of DD methods. Dependencies and environment specifications (e.g., framework versions) are included in the repository to allow full verification of all claims.

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

# APPENDIX

This appendix provides supplementary materials supporting the `BEARD` benchmark for evaluating the adversarial robustness of dataset distillation methods. The content is organized as follows:

- Appendix A presents an overview of `BEARD`, including the experimental setup, implementation details, and configuration settings.
- Appendix B provides detailed robustness analyses, including unified benchmarks, IPC variations, additional datasets and attack methods, the effect of adversarial training, robustness curves, comparisons with non-distilled methods, metric analyses, and attack convergence.
- Appendix C discusses the limitations of `BEARD`, outlines future directions, and highlights the practical applications and potential impact of the benchmark.
- Appendix D presents the statement on the use of large language models, clarifying their role in manuscript preparation.

## A    OVERVIEW OF `BEARD`

### A.1    EXPERIMENTAL SETUP

#### A.1.1    DATASETS AND DISTILLATION METHODS

**Dataset.**    In our experiments, we use three standard image classification datasets: MNIST (LeCun et al., 1998), FashionMNIST (Xiao et al., 2017), SVHN (Netzer et al., 2011), CIFAR-10 (Krizhevsky, 2009), CIFAR-100 (Krizhevsky, 2009), and TinyImageNet (Deng et al., 2009). Each dataset has been selected for its relevance and complexity in the context of dataset distillation and adversarial robustness evaluation.

- **MNIST** (LeCun et al., 1998) comprises 60,000 training images and 10,000 testing images of grayscale handwritten digits ranging from 0 to 9. It consists of 10 classes, and each image is $28 \times 28$ pixels in size.
- **FashionMNIST** (Xiao et al., 2017) consists of 10 classes of grayscale fashion items. The training set contains 60,000 images, and the test set contains 10,000 images. Each image is also in a $28 \times 28$ pixel format.
- **SVHN** (Netzer et al., 2011) contains 73,257 training images and 26,032 test images of house numbers captured from Google Street View. It includes digit sequences ranging from 0 to 9, with each image being $32 \times 32$ pixels in size.
- **CIFAR-10** (Krizhevsky, 2009) contains 60,000 $32 \times 32$ color images in 10 classes, with 50,000 images for training and 10,000 for testing. The images are preprocessed to normalize pixel values to the range [0, 1].
- **CIFAR-100** (Krizhevsky, 2009) Similar to CIFAR-10 but with 100 classes, this dataset contains 60,000 images, divided into 50,000 training and 10,000 testing images. Each image is resized to $32 \times 32$ pixels and normalized.
- **TinyImageNet** (Deng et al., 2009) A subset of the large-scale ImageNet dataset, TinyImageNet contains 200 classes with 100,000 training images and 10,000 images each for validation and testing. Images are resized to $64 \times 64$ pixels and normalized.

**Dataset Distillation Methods.**    Our benchmark evaluates six representative dataset distillation methods: DC (Zhao et al., 2020), DSA (Zhao & Bilen, 2021), DM (Zhao & Bilen, 2023), MTT (Cazenavette et al., 2022), IDM (Zhao et al., 2023), and BACON (Zhou et al., 2024a). These methods encompass a range of distillation techniques commonly employed in recent research, including gradient matching (Zhao et al., 2020; Zhao & Bilen, 2021), distribution matching (Zhao & Bilen, 2023; Zhao et al., 2023; Zhou et al., 2024a), and trajectory matching (Cazenavette et al., 2022).

- **DC** (Zhao et al., 2020) formulates dataset distillation as a bi-level optimization problem, focusing on matching the gradients of deep neural networks trained on the original dataset $\mathcal{T}$ and the synthetic dataset $\mathcal{S}$.

- **DSA** (Zhao & Bilen, 2021) improves distillation by incorporating data augmentation, enabling the generation of more informative synthetic images, which enhances the performance of models trained with these augmentations.
- **DM** (Zhao & Bilen, 2023) offers a straightforward yet impactful method for generating condensed images by aligning the feature distributions of synthetic images $\mathcal{S}$ with those of the original training set $\mathcal{T}$ across multiple sampled embedding spaces.
- **MTT** (Cazenavette et al., 2022) introduces trajectory matching as a distillation technique, condensing large datasets into smaller ones by aligning the training trajectories of models trained on both the synthetic $\mathcal{S}$ and original $\mathcal{T}$ datasets.
- **IDM** (Zhao et al., 2023) proposes a novel dataset condensation approach based on distribution matching, which proves to be both efficient and promising for DD tasks.
- **BACON** (Zhou et al., 2024a) applies a Bayesian framework to dataset distillation, framing it as a risk minimization problem. By using priors to estimate posterior probabilities, it enhances both performance and efficiency.

**Training Details.** Neural networks are trained from scratch on each distilled dataset, following a standardized training process across all experiments to ensure fair comparisons:

- **Optimizer:** The Adam optimizer is used with default settings, including a learning rate of 1e-4 and beta values of 0.9 and 0.999, ensuring stable and efficient optimization.
- **Epochs:** Models are trained for 1,000 epochs to ensure sufficient convergence and allow for the full learning potential of each distilled dataset.
- **Batch Size:** A batch size of 128 is employed to balance computational efficiency with model performance, optimizing resource usage without sacrificing accuracy.
- **Model Selection:** After training, the model with the highest validation accuracy on the original test set is selected and incorporated into the model pool for subsequent adversarial evaluations.

**Adversarial Attack Methods.** All attacks are implemented using the *torchattacks* library (Kim, 2020), which includes a comprehensive set of current adversarial attack methods. To ensure fair comparisons, we apply consistent parameters across different models. Our attack library encompasses a range of methods, including FGSM (Goodfellow et al., 2015b), PGD (Madry et al., 2018b), C&W (Carlini & Wagner, 2017), DeepFool (Moosavi-Dezfooli et al., 2016), AutoAttack (Croce & Hein, 2020), DIFGSM (Xie et al., 2019), MIFGSM (Dong et al., 2018), TIFGSM (Dong et al., 2019), EOTPGD (Liu et al., 2019), UPGD, and others. In the evaluation stage, adversarial perturbations are applied to assess the robustness of distilled datasets generated by various distillation methods. Both targeted and non-targeted attacks are performed to evaluate adversarial robustness. To ensure consistency, all trained models are subjected to identical parameters, with a perturbation budget set to $\epsilon = \frac{8}{255}$ for all methods except DeepFool and C&W.

- **FGSM** (Goodfellow et al., 2015b): This attack generates adversarial examples by perturbing the input in the direction of the gradient of the loss function. In our experiments, the perturbation budget is set to $\epsilon = \frac{8}{255}$.
- **PGD** (Madry et al., 2018b): A more powerful extension of FGSM, PGD applies iterative steps to generate adversarial examples. The perturbation budget $\epsilon$ is set to $\frac{8}{255}$, with the number of steps `step_nums` set to 10 and step size `step_size` set to $\frac{2}{255}$.
- **C&W** (Carlini & Wagner, 2017): This attack focuses on optimizing adversarial examples to minimize perturbation while ensuring misclassification, providing a robust evaluation of model robustness. In our setup, the box-constraint parameter is set to 1, and the number of steps is set to 100.
- **DeepFool** (Moosavi-Dezfooli et al., 2016): DeepFool estimates the minimal perturbation required to induce misclassification, offering insights into the model's sensitivity to adversarial perturbations. We use `steps` set to 50 and `overshot` set to 0.02.
- **AutoAttack** (Croce & Hein, 2020): A strong ensemble of multiple attacks, AutoAttack provides a comprehensive evaluation of model robustness. In our experiments, we set the perturbation budget $\epsilon$ to $\frac{8}{255}$.

- **DIFGSM** (Xie et al., 2019): This attack extends FGSM by applying random input diversification before each gradient step, improving transferability of adversarial examples. In our experiments, the diversification probability is set to 0.5, and the perturbation budget is $\epsilon = \frac{8}{255}$.

- **MIFGSM** (Dong et al., 2018): Momentum Iterative FGSM introduces a momentum term into the gradient update, stabilizing the optimization and enhancing attack strength. We use a decay factor of 1.0 and set the perturbation budget to $\epsilon = \frac{8}{255}$.

- **TIFGSM** (Dong et al., 2019): Translation-Invariant FGSM improves transferability by convolving gradients with a Gaussian kernel, making the attack less sensitive to spatial shifts. The kernel size is set to 15, and the perturbation budget is fixed at $\epsilon = \frac{8}{255}$.

- **EOTPGD** (Liu et al., 2019): Expectation Over Transformation PGD computes gradients over multiple stochastic transformations, enabling attacks against randomized or stochastic defenses. We set the number of EOT samples to 2 and use $\epsilon = \frac{8}{255}$.

- **UPGD**: An enhanced variant of PGD (Madry et al., 2018b) that integrates momentum (Dong et al., 2018), gradient normalization, Expectation Over Transformation (EOT) (Athalye et al., 2018), and multiple loss objectives including CE, margin, and DLR. In our experiments, we set $\epsilon = \frac{8}{255}$, step size $= \frac{2}{255}$, number of steps $= 10$, and momentum decay $= 1.0$.

We generate synthetic images under IPC-1, IPC-10, and IPC-50 settings from three datasets: CIFAR-10, CIFAR-100, and TinyImageNet. To assess the effectiveness of our approach, we train models on these synthetic images and evaluate their performance on the original test sets. All methods utilize the default data augmentation strategies provided by the original authors to ensure consistency in distillation performance evaluation. For a fair comparison in generalization, we use the synthetic datasets released by the authors.

After training, we apply a range of adversarial attacks to the models trained on the synthetic datasets and report the mean accuracy across 5 runs, with models randomly initialized and trained for 1,000 epochs. The evaluation metrics employed in our experiments are designed to provide a comprehensive assessment of adversarial robustness. These metrics include:

- **Single-Adversary Robustness Ratio (RRS):** Measures how effectively the models resist adversarial attacks under a single adversary.

- **Multi-Adversary Robustness Ratio (RRM):** Assesses the model's robustness against attacks from multiple adversaries.

- **Single-Adversary Attack Efficiency Ratio (AES):** Quantifies the efficiency of single adversarial attacks in terms of the time required to succeed.

- **Multi-Adversary Attack Efficiency Ratio (AEM):** Evaluates the efficiency of attacks involving multiple adversaries.

- **Comprehensive Robustness-Efficiency Index (CREI):** Integrates both robustness and attack efficiency into a unified metric, offering a balanced evaluation of model performance under adversarial conditions.

## A.2 EXPERIMENTAL SETTINGS

**Networks Architectures.** In our experiments, we employed the ConvNet architecture (Sagun et al., 2017) for dataset distillation, following methodologies from prior studies, including DC-bench (Cui et al., 2022) and BACON (Zhou et al., 2024a). The ConvNet consists of three identical convolutional blocks followed by a final linear classifier. Each block features a convolutional layer with 128 kernels of size $3 \times 3$, instance normalization, ReLU activation, and average pooling with a stride of 2 and a pooling size of $3 \times 3$. This architecture configuration is consistent with the settings outlined in DC-bench and BACON, ensuring adherence to established practices in dataset distillation.

**Evaluation Protocol.** We generate synthetic images under IPC-1, IPC-10, and IPC-50 settings from three datasets: CIFAR-10, CIFAR-100, and TinyImageNet. To assess the effectiveness of our approach, we train models on these synthetic images and evaluate their performance on the original test sets. All methods utilize the default data augmentation strategies provided by the original authors

Table 2: Performance comparison of dataset distillation methods under various adversarial attacks. Metrics include Multi-Adversary Robustness Ratio (RRM), Multi-Adversary Attack Efficiency Ratio (AEM), and Comprehensive Robustness-Efficiency Index (CREI). The Targ. Att. and Untarg. Att. denote the Targeted Attack and Untargeted Attack, respectively. The best results are highlighted in **bold**, while the second-best results are underlined.

| Evaluation | | | Dataset Distillation (%) | | | | | | |
|---|---|---|---|---|---|---|---|---|---|
| Metric | Attack Type | Dataset | Full-size | DC | DSA | MTT | DM | IDM | BACON |
| RRM | Targ. Att. | CIFAR-10 | 20.42 | 30.79 | 45.22 | 36.00 | **46.01** | 32.35 | 36.83 |
| | | CIFAR-100 | 6.77 | 33.11 | **43.97** | 36.06 | 39.32 | 30.79 | 31.81 |
| | | TinyImageNet | 22.99 | **52.62** | 49.87 | 40.05 | 49.57 | / | 47.57 |
| | Untarg. Att. | CIFAR-10 | 20.42 | 30.79 | 45.22 | 36.00 | **46.01** | 32.35 | 36.83 |
| | | CIFAR-100 | 6.77 | 33.11 | **43.97** | 36.06 | 39.32 | 30.79 | 31.81 |
| | | TinyImageNet | 22.99 | **52.62** | 49.87 | 40.05 | 49.57 | / | 47.57 |
| AEM | Targ. Att. | CIFAR-10 | **29.39** | 27.91 | 27.64 | 28.52 | 26.01 | 23.15 | 29.27 |
| | | CIFAR-100 | **29.59** | 27.50 | 26.05 | 26.25 | 23.31 | 19.89 | 27.76 |
| | | TinyImageNet | 29.83 | 28.80 | 28.97 | 29.26 | 29.55 | / | **29.96** |
| | Untarg. Att. | CIFAR-10 | 21.91 | 21.53 | 18.97 | 19.21 | 22.13 | **23.89** | 21.53 |
| | | CIFAR-100 | **17.29** | 16.06 | 12.26 | 13.23 | 12.83 | 14.44 | 13.34 |
| | | TinyImageNet | **17.31** | 14.04 | 12.77 | 14.71 | 13.08 | / | 12.09 |
| CREI | Targ. Att. | CIFAR-10 | 24.91 | 29.35 | **36.43** | 32.26 | 36.01 | 27.75 | 33.05 |
| | | CIFAR-100 | 18.18 | 30.31 | **35.01** | 31.16 | 31.32 | 27.16 | 29.78 |
| | | TinyImageNet | 26.41 | **40.71** | 39.42 | 34.66 | 39.56 | / | 38.76 |
| | Untarg. Att. | CIFAR-10 | 25.12 | 26.70 | 27.75 | 26.26 | 28.32 | **28.46** | 27.20 |
| | | CIFAR-100 | 18.60 | **22.40** | 20.40 | 19.65 | 19.78 | 20.36 | 19.30 |
| | | TinyImageNet | 15.15 | **20.46** | 15.67 | 16.13 | 15.51 | / | 15.24 |

to maintain consistency in distillation performance evaluation. For fair comparisons in generalization, we employ the synthetic datasets released by the authors.

Following model training, we apply adversarial attacks to evaluate the robustness of the models trained on the various synthetic datasets. We report the mean accuracy across 5 runs, with models randomly initialized and trained for 1,000 epochs. The performance is measured using the condensed set as the primary evaluation metric.

### A.3 IMPLEMENTATION DETAILS

The `BEARD` benchmark builds upon the software foundation established by BACON (Zhou et al., 2024a). For generating synthetic images in the dataset pool, we use the Stochastic Gradient Descent (SGD) optimizer with a learning rate of 0.2 and a momentum of 0.5, applied to synthetic datasets with IPC-1, IPC-10, and IPC-50 settings. In the subsequent model training phase, we employ the same SGD optimizer, but adjust the learning rate to 0.01, momentum to 0.9, and apply a weight decay of 0.0005. The batch size is set to 256. All experiments, including both the generation of synthetic datasets and the training of models, are conducted using NVIDIA RTX 2080 Ti GPU clusters. Additionally, we provide a configuration JSON file to facilitate the setup and management of experimental parameters.

## B ANALYSIS

### B.1 ROBUSTNESS EVALUATION USING RR, AE, AND CREI METRICS

The Table 2 compares the performance of various dataset distillation methods using three key metrics: Multi-Adversary Robustness Ratio (RRM), Multi-Adversary Attack Efficiency Ratio (AEM), and Comprehensive Robustness-Efficiency Index (CREI). The evaluation covers both targeted and untargeted adversarial attacks across three datasets: CIFAR-10, CIFAR-100, and TinyImageNet. The best results are highlighted in bold, while the second-best results are underlined.

Table 3: Comparison of adversarial robustness using CREI for different dataset distillation methods under targeted attacks across various datasets and IPC settings. The best results are highlighted in **bold**, while the second-best results are underlined.

| Dataset | IPC | Dataset Distillation (%) | | | | | |
|---------|-----|-------|-------|-------|-------|-------|-------|
| | | DC | DSA | MTT | DM | IDM | BACON |
| CIFAR-10 | Full-size | 24.91 | | | | | |
| | 1 | **27.19** | 26.11 | 25.67 | 24.87 | 21.92 | 24.62 |
| | 10 | **25.73** | 23.31 | 22.21 | 20.90 | 20.41 | 20.83 |
| | 50 | **22.36** | 21.75 | 19.82 | 19.70 | 17.11 | 19.10 |
| CIFAR-100 | Full-size | 18.18 | | | | | |
| | 1 | 28.23 | **29.74** | 27.65 | 28.30 | 24.44 | 28.71 |
| | 10 | **26.23** | 23.58 | 23.23 | 23.97 | 18.47 | 20.38 |
| | 50 | 19.40 | 20.64 | **21.51** | 19.70 | 16.86 | 19.86 |
| TinyImageNet | Full-size | 26.41 | | | | | |
| | 1 | 29.94 | 29.08 | 29.33 | 30.44 | / | **30.50** |
| | 10 | **30.46** | 30.28 | 29.93 | 29.30 | / | 28.18 |
| | 50 | **29.10** | 28.89 | / | 28.72 | / | / |

**Targeted Attacks.** Dataset distillation methods demonstrate substantial improvements in robustness compared to full-size models. For example, in CIFAR-10, DM achieves a RRM of 46.01%, a significant increase from the 20.42% of the full-size model. Similarly, DSA achieves a RRM of 45.22%. These enhancements are evident across CIFAR-100 and TinyImageNet, where DM and DSA continue to outperform full-size models. For instance, in CIFAR-100, DM has a RRM of 39.32%, compared to 6.77% for the full-size model. On TinyImageNet, DM achieves a RRM of 49.57%, compared to 22.99% for the full-size model. Despite these improvements in robustness, distillation methods like DC and DSA show a slight reduction in AEM values. For example, in CIFAR-10, DC has an AEM of 27.91% and DSA has 27.64%, compared to 29.39% for the full-size model. This indicates a trade-off between robustness and efficiency. The CREI scores further illustrate this balance: DM and DSA achieve high CREI values, with DM reaching 36.01% in CIFAR-10 and DSA 36.43%, showcasing their effective trade-off between robustness and efficiency.

**Untargeted Attacks.** The robustness improvements with distillation methods are less pronounced compared to targeted attacks. For instance, in CIFAR-10, while DM and DSA still offer high RRM (45.22% and 46.01%, respectively), the gap between these methods and full-size models is narrower. The AEM values for distillation methods are generally lower, indicating that these methods require less time and computational resources for adversarial attacks compared to full-size models. For example, the AEM for DC in CIFAR-10 under untargeted attacks is 21.53%, compared to 21.91% for the full-size model. Similarly, DSA shows an AEM of 18.97% in CIFAR-10, which is lower than the 21.91% for the full-size model. The CREI scores reflect this trend, with methods like DM and DSA achieving reasonable CREI values, such as 27.75% for DM in CIFAR-10, demonstrating a balanced performance between robustness and efficiency despite the slight trade-off in robustness.

B.2 ROBUSTNESS EVALUATION WITH DIVERSE IPCS

The robustness evaluation, conducted across targeted and untargeted attacks, reveals two key observations: (1) increasing the number of IPC decreases adversarial robustness, as evidenced by lower CREI values across various methods and datasets; and (2) increasing the dataset scale enhances adversarial robustness when using dataset distillation methods, particularly when comparing distilled datasets to full-size datasets.

**Targeted Attacks.** In the context of targeted attacks, increasing IPC values typically leads to reduced adversarial robustness, as seen from the declining CREI scores. For instance, on CIFAR-10, methods like DC and DSA perform best at IPC = 1, showing strong robustness, but their performance decreases with larger IPC values. Similarly, for CIFAR-100, BACON outperforms other methods

Table 4: Comparison of adversarial robustness using CREI for different dataset distillation methods under untargeted attacks across various datasets and IPC settings. The best results are highlighted in **bold**, while the second-best results are underlined.

| Dataset | IPC | Dataset Distillation (%) | | | | | |
|---|---|---|---|---|---|---|---|
| | | DC | DSA | MTT | DM | IDM | BACON |
| CIFAR-10 | Full-size | 25.12 | | | | | |
| | 1 | 41.11 | 36.59 | 26.25 | **42.21** | 30.18 | 25.79 |
| | 10 | 24.85 | 23.90 | 23.40 | 26.73 | 25.60 | **27.09** |
| | 50 | 22.50 | 27.00 | 25.06 | 26.61 | **29.59** | 28.48 |
| CIFAR-100 | Full-size | 18.60 | | | | | |
| | 1 | 23.87 | 20.81 | 17.91 | **24.32** | 20.85 | 19.76 |
| | 10 | 16.44 | 20.83 | 19.97 | 20.28 | **21.43** | 20.78 |
| | 50 | 19.46 | 20.67 | 20.54 | 20.10 | **21.34** | 20.11 |
| TinyImageNet | Full-size | 15.15 | | | | | |
| | 1 | **22.86** | 18.41 | 20.98 | 22.20 | / | 17.88 |
| | 10 | **17.33** | 15.50 | 15.90 | 15.59 | / | 16.18 |
| | 50 | 14.96 | **15.50** | / | 15.30 | / | / |

at IPC = 1, though its robustness diminishes at higher IPC levels. Importantly, as the dataset scale increases, the advantage of dataset distillation methods over Full-size datasets becomes more pronounced. For example, in TinyImageNet at IPC = 1, DC and DSA maintain high CREI scores, surpassing the Full-size model, emphasizing that dataset distillation methods can achieve better robustness with smaller dataset sizes under targeted attacks. The detailed results are presented in Table 3. The best results are highlighted in bold, while the second-best results are underlined.

Table 5: Comparison of adversarial robustness using CREI for different dataset distillation methods with and without adversarial training under targeted and untargeted attacks. The best results are highlighted in **bold**

| Attack Type | Adversarial Training | Dataset Distillation (%) | | | | | | |
|---|---|---|---|---|---|---|---|---|
| | | Full-size | DC | DSA | MTT | DM | IDM | BACON |
| Targeted | w/ AT | **50.54** | **52.30** | **55.56** | **50.96** | **57.21** | **54.67** | **56.18** |
| | w/o AT | 24.91 | 29.35 | 36.43 | 32.26 | 36.01 | 27.75 | 33.05 |
| Untargeted | w/ AT | **41.33** | **37.39** | **37.59** | **33.13** | **37.34** | **39.05** | **37.25** |
| | w/o AT | 25.12 | 26.70 | 27.75 | 26.26 | 28.32 | 28.46 | 27.20 |

Table 6: Robustness evaluation of dataset distillation methods, with and without adversarial training, on CIFAR-100 with IPC-50 under targeted and untargeted attacks (%).

| Method | Targeted | | | Untargeted | | |
|---|---|---|---|---|---|---|
| | RR | AE | CREI | RR | AE | CREI |
| DM | 9.38 | 30.02 | 19.7 | 22.44 | 17.75 | 20.1 |
| DM+AT | 33.98 | 30.3 | 32.14 | 30.7 | 30.31 | 30.5 |
| IDM | 11.6 | 22.13 | 16.86 | 24.68 | 17.98 | 21.33 |
| IDM+AT | 97.69 | 28.03 | 62.86 | 34.29 | 23.39 | 28.84 |
| BACON | 8.99 | 30.73 | 19.86 | 22.82 | 17.4 | 20.11 |
| BACON+AT | 87.6 | 30.29 | 58.94 | 34.17 | 22.32 | 28.25 |

**Untargeted Attacks.** Under untargeted attacks, the trend of decreasing robustness with increasing IPC is also observed, but the effects are less severe compared to targeted attacks. For CIFAR-10, DC

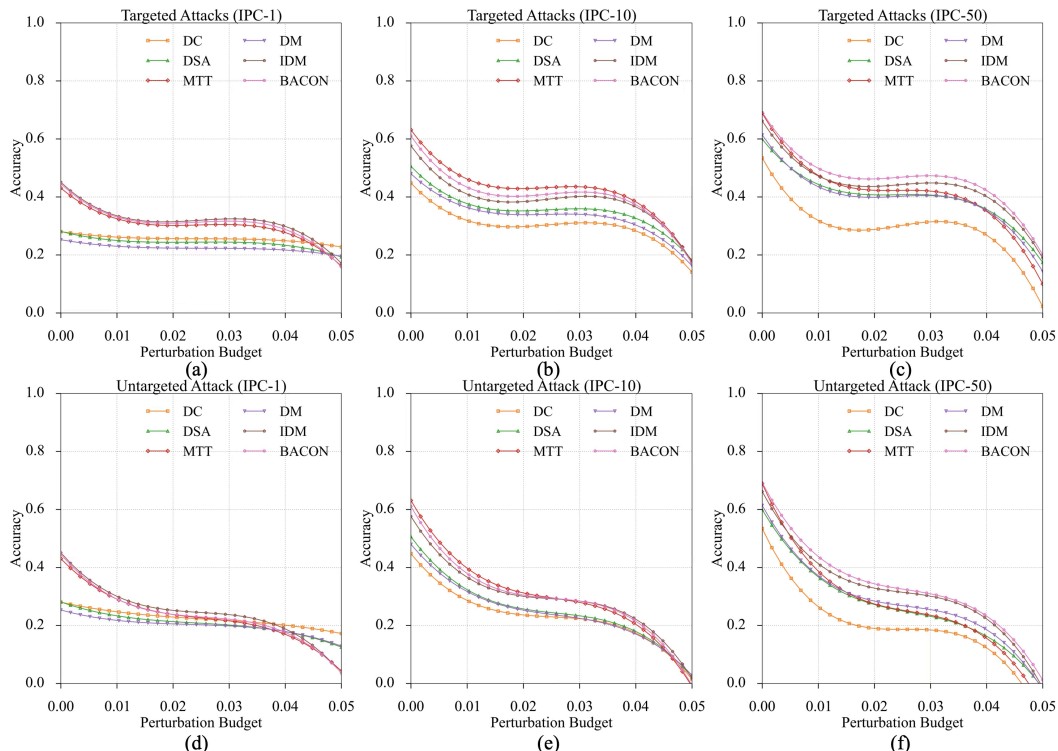

Figure 6: Robustness curves of DD methods on CIFAR-10 under targeted and untargeted PGD attacks (IPC-{1,10,50}), with the x-axis representing perturbation budget $\epsilon$ and the y-axis representing classification accuracy.

and DM perform strongly at IPC = 1, with DC achieving the highest CREI score. Notably, as the dataset size increases (e.g., TinyImageNet), the gap in robustness between distillation methods and Full-size datasets becomes more evident. For instance, DC consistently outperforms the Full-size model in TinyImageNet at IPC = 1, while showing comparable or better robustness even at higher IPC values. This reinforces the observation that dataset distillation methods not only excel with fewer images per class but also offer greater robustness in larger datasets, especially when facing untargeted attacks. Detailed results are provided in Table 4. The best results are highlighted in bold, while second-best results are underlined.

### B.3 ROBUSTNESS EVALUATION WITH ADVERSARIAL TRAINING

**Targeted Attacks.** Adversarial training substantially improves robustness under targeted attacks. On CIFAR-10 (Table 5), dataset distillation methods such as BACON and DM achieve high CREI values of 56.18% and 57.21%, respectively, compared to 50.54% for the full-size dataset, highlighting the effectiveness of dataset distillation in enhancing adversarial robustness. Even without AT, dataset distillation methods maintain moderate robustness, with DM and BACON outperforming other methods, indicating inherent adversarial robustness. These observations are further confirmed on the larger CIFAR-100 dataset under the IPC-50 setting (Table 6), where dataset distillation methods with AT, including DM+AT, IDM+AT, and BACON+AT, achieve CREI values of 32.14%, 62.86%, and 58.94%, consistently outperforming their non-AT counterparts.

**Untargeted Attacks.** Adversarial training also improves robustness under untargeted attacks, though the absolute gains in CREI are smaller than those observed for targeted attacks. On CIFAR-10 (Table 5), the Full-size dataset achieves a CREI of 41.33%, slightly higher than other methods, while dataset distillation methods such as BACON and DM attain CREI values of 37.25% and 37.34%, respectively. Without AT, all methods exhibit reduced robustness, with DSA and DM maintaining relatively higher CREI values of 27.75% and 28.32%, indicating that dataset distillation provides some inherent adversarial robustness even without AT. These observations are further confirmed on

Table 7: Comparison of dataset distillation methods on CIFAR-10 under targeted attacks using IPC-1, IPC-10, and IPC-50. Results are evaluated with DD-RobustBench and `BEARD`. All values indicate accuracy (%) except those corresponding to the RRM metric.

| Dataset | IPC | Attack | DD-RobustBench | | | | | BEARD | | | | |
|---|---|---|---|---|---|---|---|---|---|---|---|---|
| | | | DC | DSA | MTT | DM | IDM | DC | DSA | MTT | DM | IDM |
| | | **RRM across IPCs ⇒** | | | | | | **16.55** | **16.10** | **11.72** | **15.07** | **5.50** |
| | | **RRM ⇒** | | | | | | **12.44** | **7.46** | **11.59** | **7.24** | **3.17** |
| | | None(Clean) | 29.73 | 29.27 | 45.74 | 26.75 | 46.14 | 28.11 | 28.23 | 43.44 | 25.34 | 46.01 |
| | | FGSM | 22.12 | 19.78 | 26.21 | 20.08 | 22.86 | 22.49 | 20.83 | 24.55 | 19.95 | 27.06 |
| | 1 | PGD | 19.78 | 16.82 | 19.79 | 18.2 | 21.55 | 21.32 | 19.75 | 20.22 | 18.98 | 23.57 |
| | | C&W | 18.41 | 16.63 | 16.83 | 18.01 | 21.26 | 19.32 | 15.85 | 16.4 | 16.31 | 16.73 |
| | | Autoattck | 18.19 | 16.47 | 16.41 | 17.91 | 20.82 | 17.91 | 16.5 | 17.38 | 16.07 | 20.06 |
| | | **RRM ⇒** | | | | | | **9.29** | **7.86** | **10.51** | **7.09** | **2.87** |
| CIFAR-10 | | None(Clean) | 46.07 | 52.93 | 60.98 | 49.81 | 58.84 | 45.31 | 51.06 | 63.76 | 48.45 | 58.47 |
| | | FGSM | 24.4 | 28.12 | 30.99 | 22.96 | 25.54 | 25.48 | 28.21 | 35.34 | 26.38 | 32.31 |
| | 10 | PGD | 18.27 | 20.01 | 21.95 | 14.74 | 23.34 | 20.83 | 21.98 | 25.96 | 20.98 | 26.34 |
| | | C&W | 16.5 | 19.83 | 19 | 14.95 | 23.17 | 15.88 | 13.21 | 15.71 | 11.05 | 14.49 |
| | | Autoattck | 16 | 19.11 | 17.95 | 14.18 | 22.63 | 17.75 | 12.82 | 23.13 | 18.29 | 23.61 |
| | | **RRM ⇒** | | | | | | **8.06** | **7.33** | **8.55** | **8.21** | **2.87** |
| | | None(Clean) | 55.08 | 61.14 | 70.45 | 63.12 | 67.82 | 54.31 | 60.48 | 69.99 | 62.06 | 67.07 |
| | | FGSM | 24.54 | 24.33 | 32.32 | 30.61 | 25.82 | 24.79 | 33.42 | 34.62 | 33.97 | 36.84 |
| | 50 | PGD | 14.82 | 14.81 | 19.19 | 19.36 | 22.47 | 16.38 | 21.32 | 21.29 | 23.53 | 28.42 |
| | | C&W | 14.23 | 13.84 | 18.87 | 19.47 | 22.74 | 8.37 | 6.78 | 7.08 | 8.36 | 9.5 |
| | | Autoattck | 13.38 | 12.95 | 17.41 | 18.23 | 21.92 | 13.33 | 19.19 | 18.95 | 21.14 | 26.23 |

the larger CIFAR-100 dataset with IPC-50 (Table 6), where dataset distillation methods combined with AT, including DM+AT, IDM+AT, and BACON+AT, achieve CREI values of 30.5%, 28.84%, and 28.25%, consistently outperforming their non-AT counterparts. Overall, these results demonstrate that dataset distillation methods, particularly when combined with adversarial training, offer consistent improvements in robustness across both attack types and dataset scales.

## B.4 ROBUSTNESS EVALUATION WITH ROBUSTNESS CURVE

Figure 6 presents the robustness curves of different dataset distillation methods on CIFAR-10 under targeted and untargeted PGD attacks, evaluated with IPC values of 1, 10, and 50. Overall, as the perturbation budget increases, the accuracy of all methods declines, with untargeted attacks causing more severe degradation than targeted attacks.

Among the methods, BACON consistently demonstrates the strongest robustness, especially at higher IPC settings. In contrast, DC and DSA exhibit the weakest robustness across all scenarios, while IDM, DM, and MTT offer some resistance but still experience notable performance drops. These findings highlight the importance of enhancing data quality and employing more robust distillation strategies, such as BACON, to improve adversarial robustness. Furthermore, the greater impact of untargeted attacks suggests they should be prioritized in robustness evaluations.

## B.5 THE DIFFERENCES BETWEEN BEARD AND OTHER BENCHMARKS

`BEARD` introduces key innovations compared to benchmarks like DD-RobustBench Wu et al. (2024) and RobustBench Croce et al. (2021). It combines conceptual innovation, providing a unified evaluation perspective across multiple IPCs and attacks grounded in an adversarial game framework, with engineering innovation, employing a modular software architecture with flexible model and dataset pools that facilitate easy integration of custom DD methods, attacks, and datasets. These innovations enable a systematic, comprehensive, and interpretable evaluation of DD adversarial robustness.

While DD-RobustBench evaluates dataset distillation methods under different IPC settings, `BEARD` provides a more holistic assessment by aggregating results across multiple IPC values using unified metrics such as RRM (Table 7). Additionally, `BEARD` evaluates both targeted and untargeted attacks, whereas DD-RobustBench primarily focuses on attack effectiveness under untargeted settings.

Table 8: Comparison of Dataset Distillation Methods on CIFAR-10 IPC-50 under Targeted Adversarial Attacks (%).

| Attack | Full-size | DC | DSA | MTT | DM | IDM | BACON |
|--------|-----------|-------|-------|-------|-------|-------|-------|
| Clean | 87.41 | 54.31 | 60.48 | 69.66 | 62.06 | 66.91 | 69.95 |
| DIFGSM | 41.11 | 30.02 | 39.71 | 41.20 | 39.21 | 43.56 | 45.92 |
| MIFGSM | 38.87 | 29.38 | 39.22 | 39.93 | 38.71 | 43.25 | 45.77 |
| TIFGSM | 52.78 | 30.62 | 42.40 | 45.31 | 41.66 | 45.48 | 48.38 |
| EOTPGD | 37.78 | 29.58 | 39.20 | 40.00 | 38.59 | 43.23 | 45.57 |
| UPGD | 38.87 | 29.38 | 39.22 | 39.93 | 38.71 | 43.25 | 45.77 |

Table 9: Comparison of dataset distillation methods under combined adversarial attacks on CIFAR-10 across IPC-M for both targeted and untargeted attacks. RR, AE, and CREI (%) are used to evaluate adversarial robustness.

| Method | Targeted | | | Untargeted | | |
|--------|-------|-------|-------|-------|-------|-------|
| | RR | AE | CREI | RR | AE | CREI |
| Full-size | 22.79 | 16.92 | 19.85 | 25.97 | 12.34 | 19.16 |
| DC | 23.86 | 20.35 | 22.10 | 33.47 | 16.55 | 25.01 |
| DSA | 42.61 | 18.48 | 30.55 | 37.53 | 12.25 | 24.89 |
| MTT | 32.02 | 19.12 | 25.57 | 33.49 | 13.99 | 23.74 |
| DM | 43.10 | 19.14 | 31.12 | 35.64 | 14.11 | 24.88 |
| IDM | 31.15 | 19.68 | 25.42 | 34.61 | 14.98 | 24.79 |
| BACON | 34.51 | 18.93 | 26.72 | 34.59 | 14.35 | 24.47 |

Unlike RobustBench, which assesses conventional models, BEARD is specifically designed for resource-constrained DD methods, ensuring fair and meaningful robustness evaluation. We plan to enhance BEARD by incorporating more DD methods, larger datasets, and more sophisticated attacks, with potential integration into RobustBench. As an open-source and continuously updated framework, BEARD integrates seamlessly with various DD methods and datasets, featuring a leaderboard to foster further research in the field.

## B.6 TRADE-OFF BETWEEN ADVERSARIAL ROBUSTNESS AND MODEL PERFORMANCE

In our analysis, we also investigate the trade-off between adversarial robustness and model performance. It is important to note that while DD methods enhance adversarial robustness, they may also lead to a reduction in model performance on clean data. Specifically, dataset distillation techniques involve extracting key features from the original dataset, which can introduce non-robust features associated with certain classes. This process can be seen as a form of adversarial training, as it exposes the model to these non-robust features. As discussed by Ilyas *et al.* Ilyas et al. (2019), distilling a dataset essentially incorporates both robust and non-robust features from the original data, thereby improving adversarial robustness but also potentially diminishing performance on clean data. This explains the observed trade-off where DD methods achieve higher robustness at the cost of some loss in model performance. In our experiments, this balance between robustness and performance is evident. Models trained on distilled datasets demonstrate improved robustness to adversarial attacks, but also experience a slight degradation in accuracy on clean, non-adversarial inputs. As shown in Table 2, the distilled datasets generated by DD methods exhibit greater robustness to adversarial attacks compared to models trained on the original datasets. However, this improved robustness comes at the cost of reduced performance on clean, unperturbed data.

## B.7 EXPERIMENTS ON ADDITIONAL ADVERSARIAL ATTACKS, DATASETS, AND ADVERSARIAL TRAINING STRATEGIES

Table 10: Adversarial robustness of dataset distillation methods on MNIST, FashionMNIST, and SVHN under targeted and untargeted attacks. RR, AE, and CREI (%) are reported.

| Dataset | Method | Targeted | | | Untargeted | | |
|---|---|---|---|---|---|---|---|
| | | RR | AE | CREI | RR | AE | CREI |
| MNIST | DC | 82.26 | 19.90 | 51.08 | 79.69 | 13.08 | 46.38 |
| | DSA | 87.59 | 19.54 | 53.56 | 80.68 | 13.98 | 47.33 |
| | BACON | 71.55 | 23.51 | 47.53 | 79.92 | 13.95 | 46.94 |
| FashionMNIST | DC | 61.60 | 18.76 | 40.18 | 64.49 | 13.74 | 39.11 |
| | DSA | 69.18 | 18.88 | 44.03 | 66.31 | 16.84 | 41.58 |
| | BACON | 43.71 | 19.66 | 31.69 | 63.24 | 14.59 | 38.91 |
| SVHN | DC | 52.66 | 19.46 | 36.06 | 37.05 | 16.03 | 26.54 |
| | DSA | 46.81 | 20.20 | 33.50 | 33.51 | 12.59 | 23.05 |
| | BACON | 30.70 | 19.43 | 25.06 | 35.67 | 14.63 | 25.15 |

Table 11: Adversarial robustness of dataset distillation methods on CIFAR-10 (IPC-50) under various adversarial training strategies. Reported metrics are RR, AE, and CREI (%).

| Method | Targeted | | | Untargeted | | |
|---|---|---|---|---|---|---|
| | RR | AE | CREI | RR | AE | CREI |
| DM | 9.50 | 31.16 | 20.33 | 29.51 | 30.14 | 29.83 |
| DM + PGD_AT (Src) | 71.43 | 29.52 | 50.47 | 47.85 | 25.57 | 36.71 |
| DM + TRADES | 8.67 | 31.71 | 20.19 | 30.35 | 31.37 | 30.86 |
| DM + TRS | 7.85 | 35.51 | 21.68 | 30.22 | 32.44 | 31.33 |
| DM + MART | 7.76 | 31.95 | 19.85 | 29.73 | 30.01 | 29.87 |
| DM + PGD_AT (Loss) | 8.86 | 30.75 | 19.80 | 30.09 | 29.96 | 30.03 |
| IDM | 9.01 | 31.95 | 20.48 | 31.51 | 27.77 | 29.64 |
| IDM + PGD_AT (Src) | 39.59 | 29.64 | 34.61 | 46.92 | 25.26 | 36.09 |
| IDM + TRADES | 9.91 | 32.94 | 21.42 | 31.80 | 29.56 | 30.68 |
| IDM + TRS | 10.05 | 33.28 | 21.66 | 31.03 | 29.54 | 30.29 |
| IDM + MART | 10.24 | 34.11 | 22.17 | 31.10 | 27.63 | 29.37 |
| IDM + PGD_AT (Loss) | 10.75 | 31.59 | 21.17 | 30.43 | 27.85 | 29.14 |
| BACON | 7.08 | 30.88 | 18.98 | 36.43 | 28.03 | 32.23 |
| BACON + PGD_AT (Src) | 49.66 | 29.27 | 39.46 | 48.19 | 22.75 | 35.47 |
| BACON + TRADES | 7.48 | 29.70 | 18.59 | 32.39 | 32.68 | 32.53 |
| BACON + TRS | 7.73 | 29.60 | 18.66 | 32.79 | 32.47 | 32.63 |
| BACON + MART | 7.28 | 29.93 | 18.61 | 32.69 | 33.26 | 32.97 |
| BACON + PGD_AT (Loss) | 7.53 | 31.34 | 19.43 | 32.43 | 31.36 | 31.89 |

### B.7.1 Experiments on Additional Adversarial Attacks

Table 8 presents the raw accuracies of dataset distillation methods on CIFAR-10 IPC-50 under the extended targeted attacks (DIFGSM, MIFGSM, TIFGSM, EOTPGD, and UPGD), providing the basis for our analysis. To offer a more comprehensive and interpretable evaluation, we introduce combined robustness metrics, with Table 9 reporting the resulting adversarial robustness on CIFAR-10 for IPC-M across both targeted and untargeted attacks. In the targeted setting, DM achieves the highest CREI, followed by DSA, reflecting that distribution-based methods like DM tend to learn smoother decision boundaries, which improves robustness against transferable attacks. Gradient-based methods such as DSA capture more detailed information from the training distribution but exhibit sharper boundaries, making them slightly less robust under these attacks. In untargeted attacks, methods like DC perform competitively, as these attacks are generally less sensitive to boundary smoothness. The

Table 12: Comparison of average accuracy and robustness metrics (%) under various IPCs and targeted attacks.

| IPC | Method | Avg (Acc) | RR | AE | CREI | None (Clean) | FGSM | PGD | PGD_L2 | DeepFool | C&W | AutoAttack |
|---|---|---|---|---|---|---|---|---|---|---|---|---|
| IPC-1 | DM | 23.24 | 20.77 | 28.98 | 24.87 | 25.34 | 22.18 | 22.03 | 23.43 | / | 23.23 | / |
| | IDM | 37.08 | 19.31 | 24.53 | 21.92 | 46.01 | 32.18 | 33.04 | 35.01 | / | 39.17 | / |
| | BACON | 35.09 | 19.78 | 29.46 | 24.62 | 44.95 | 29.59 | 30.61 | 33.73 | / | 36.58 | / |
| IPC-10 | DM | 37.35 | 12.41 | 29.39 | 20.90 | 48.45 | 32.61 | 32.83 | 37.51 | / | 35.35 | / |
| | IDM | 44.26 | 15.82 | 25.01 | 20.41 | 58.47 | 37.37 | 38.44 | 41.57 | / | 45.45 | / |
| | BACON | 45.99 | 12.94 | 28.71 | 20.83 | 61.73 | 39.13 | 40.10 | 44.06 | / | 44.93 | / |
| IPC-50 | DM | 45.13 | 9.54 | 29.86 | 19.70 | 62.06 | 39.45 | 38.66 | 44.53 | / | 40.93 | / |
| | IDM | 49.61 | 9.44 | 24.78 | 17.11 | 67.07 | 42.97 | 43.91 | 48.92 | / | 45.18 | / |
| | BACON | 51.15 | 6.99 | 31.21 | 19.10 | 69.95 | 44.68 | 45.59 | 50.81 | / | 44.71 | / |

Table 13: Comparison of average time (s) and robustness metrics (%) under various IPCs and targeted attacks.

| IPC | Method | Avg | RR | AE | CREI | None (Clean) | FGSM | PGD | PGD_L2 | DeepFool | C&W | AutoAttack |
|---|---|---|---|---|---|---|---|---|---|---|---|---|
| IPC-1 | DM | 23.54 | 20.77 | 28.98 | 24.87 | 3.1 | 2.88 | 10.21 | 10.24 | / | 91.29 | / |
| | IDM | 18.42 | 19.31 | 24.53 | 21.92 | 6.49 | 3.56 | 7.45 | 7.32 | / | 67.29 | / |
| | BACON | 19.34 | 19.78 | 29.46 | 24.62 | 2.5 | 3.01 | 8.52 | 8.71 | / | 73.94 | / |
| IPC-10 | DM | 23.93 | 12.41 | 29.39 | 20.90 | 2.91 | 2.94 | 10.3 | 11.18 | / | 92.34 | / |
| | IDM | 18.87 | 15.82 | 25.01 | 20.41 | 6.45 | 3.27 | 8.04 | 8.06 | / | 68.54 | / |
| | BACON | 19.30 | 12.94 | 28.71 | 20.83 | 3.06 | 2.85 | 8.32 | 8.5 | / | 73.75 | / |
| IPC-50 | DM | 17.10 | 9.54 | 29.86 | 19.70 | 2.74 | 3.1 | 8.31 | 8.5 | / | 62.87 | / |
| | IDM | 20.16 | 9.44 | 24.78 | 17.11 | 6.7 | 3.37 | 8.07 | 8.05 | / | 74.61 | / |
| | BACON | 21.21 | 6.99 | 31.21 | 19.10 | 3.04 | 3.01 | 11.96 | 12.22 | / | 75.82 | / |

table illustrates how different distillation strategies interact with attack types and emphasizes the role of boundary smoothness in enhancing robustness against transferable attacks.

### B.7.2 EXPERIMENTS ON ADDITIONAL DATASETS

Table 10 reports the adversarial robustness of dataset distillation methods on MNIST, FashionMNIST, and SVHN under both targeted and untargeted attacks. Across these simpler datasets, DSA generally achieves the highest CREI under targeted attacks, indicating its ability to capture rich information from the training distribution. On MNIST, all methods perform well, while on FashionMNIST, DSA still outperforms DC and BACON, though the gap narrows due to increased texture complexity. On SVHN, a more natural and complex dataset, distribution-based methods like DSA maintain relative robustness under targeted attacks, whereas gradient-based methods such as DC show stronger performance under untargeted attacks. These results demonstrate BEARD's applicability across datasets of varying difficulty and highlight the interaction between dataset characteristics, distillation strategies, and attack types.

### B.7.3 EXPERIMENTS ON ADDITIONAL ADVERSARIAL TRAINING STRATEGIES

We evaluate the robustness of dataset distillation methods on CIFAR-10 (IPC-50) under multiple adversarial training strategies, including PGD_AT, TRADES, MART, and TRS. PGD_AT(Src) generates adversarial examples directly from the input without modifying the loss, providing strong guidance for learning robust decision boundaries. PGD_AT(Loss) additionally aligns predictions of clean and adversarial examples with the labels, yielding more conservative regularization. TRADES balances robustness and accuracy via cross-entropy and KL divergence, MART emphasizes misclassified examples, and TRS promotes gradient diversity and model smoothness through multiple PGD adversarial examples. Table 11 shows that PGD_AT(Src) consistently produces the largest improvements under both targeted and untargeted attacks, while PGD_AT(Loss) offers moderate gains, and TRADES, MART, and TRS provide alternative trade-offs between robustness and attack efficiency. BACON exhibits higher baseline robustness than DM and IDM due to richer feature representations, and PGD_AT(Src) reduces the gap across methods. All methods show larger gains under untargeted attacks because targeted attacks explicitly push the model toward a specific incorrect

class, making defense more challenging. Overall, these results demonstrate that dataset distillation methods effectively leverage adversarial training strategies, with PGD_AT(Src) providing the strongest enhancement through direct adversarial guidance, and other strategies offering controlled or diversity-based improvements.

## B.8 METRICS ANALYSIS

### B.8.1 ANALYSIS OF AVERAGE AND ROBUSTNESS METRICS

Table 14: CREI values of different dataset distillation methods under varying hyperparameter $\alpha$ (%).

| Method | RR | AE | $\alpha = 0.0$ | $\alpha = 0.2$ | $\alpha = 0.4$ | $\alpha = 0.6$ | $\alpha = 0.8$ | $\alpha = 1.0$ |
|--------|------|------|------|------|------|------|------|------|
| DC | 31.87 | 21.53 | 21.53 | 23.60 | 25.67 | 27.73 | 29.80 | 31.87 |
| DSA | 36.53 | 18.97 | 18.97 | 22.48 | 25.99 | 29.51 | 33.02 | 36.53 |
| MTT | 33.30 | 19.21 | 19.21 | 22.03 | 24.85 | 27.66 | 30.48 | 33.30 |
| DM | 34.50 | 22.13 | 22.13 | 24.60 | 27.08 | 29.55 | 32.03 | 34.50 |
| IDM | 33.03 | 23.89 | 23.89 | 25.72 | 27.55 | 29.37 | 31.20 | 33.03 |
| BACON | 32.87 | 21.53 | 21.53 | 23.80 | 26.07 | 28.33 | 30.60 | 32.87 |

We analyze the metrics used for evaluating model robustness under varying IPC settings and adversarial attacks. Specifically, we focus on how different metrics, including average accuracy (Avg (Acc)), Robustness Ratio (RR), Attack Efficiency Ratio (AE), and average attack time (Avg (Time)), reflect models' performance and resilience. Avg (Acc) is highly influenced by IPC settings and the inherent differences between dataset distillation (DD) methods, making it sensitive to variations in clean accuracy. In contrast, RR and AE isolate the effects of adversarial attacks, offering more consistent and comparable measures of robustness across different models. Table 12 illustrates this by showing that while Avg (Acc) varies significantly with IPC settings, RR remains relatively stable. For instance, at IPC-1, Avg (Acc) ranges from 23.24 for DM to 37.08 for IDM, while RR changes only slightly from 20.77 to 19.31. This stability makes RR a more reliable metric for comparing robustness independent of clean accuracy variations across IPC settings.

Further analysis in Table 13 reveals that Avg (Time), being sensitive to hardware differences, is less reliable as a robustness metric. For example, the clean inference times for DM and IDM are 3.1s and 6.49s, respectively, which introduces significant variability in Avg (Time) due to hardware disparities. This makes Avg (Time) unsuitable for fair comparisons, as it does not normalize for hardware differences. In contrast, AE normalizes the results across multiple adversarial attack types, mitigating the effect of hardware variations and providing a more stable and consistent measure of robustness. Therefore, RR and AE offer a more comprehensive and unbiased evaluation of adversarial resilience, enabling fairer comparisons across DD methods, while Avg (Acc) and Avg (Time) are heavily influenced by external factors such as IPC settings and hardware configurations, limiting their utility in adversarial robustness evaluation.

### B.8.2 ANALYSIS OF HYPERPARAMETER VARIATION ON CREI

The choice of the hyperparameter $\alpha$ in the CREI metric controls the balance between RR and AE, reflecting the trade-off between adversarial robustness and attack efficiency. A smaller $\alpha$ gives more weight to AE, favoring methods with higher attack efficiency, while a larger $\alpha$ emphasizes RR, favoring methods with stronger robustness. Setting $\alpha = 0.5$ provides an equal weight to both RR and AE, offering a balanced assessment, but the metric remains flexible for task-specific priorities. In resource-constrained or edge scenarios, where minimizing attack time is critical, a smaller $\alpha$ is ideal. In contrast, for tasks focused on evaluating robustness against adversarial attacks, a larger $\alpha$ should be preferred. To investigate the effect of varying $\alpha$, we conducted a sensitivity analysis (Table 14) on CIFAR-10 under combined IPC settings of $1, 10, 50$, demonstrating how the trade-off between

Table 15: Comparison of adversarial robustness between DD methods and random subsets under targeted attacks on CIFAR-10 with IPC-1,10,50.

| Method | RR | AE | CREI |
|---|---|---|---|
| Full-size | 67.24 | 29.55 | 48.39 |
| Subset | 54.47 | 32.29 | 43.38 |
| DC | 88.51 | 27.91 | 58.21 |
| DSA | 86.81 | 27.64 | 57.22 |
| MTT | 83.95 | 28.52 | 56.24 |
| DM | 85.76 | 26.01 | 55.89 |
| IDM | 87.07 | 23.15 | 55.11 |
| BACON | 84.37 | 29.27 | 56.82 |

Table 16: Comparison of adversarial robustness between DD methods and random subsets under untargeted attacks on CIFAR-10 at IPC-1, IPC-10, and IPC-50.

| Method | IPC-1 | | | IPC-10 | | | IPC-50 | | |
|---|---|---|---|---|---|---|---|---|---|
| | RR | AE | CREI | RR | AE | CREI | RR | AE | CREI |
| Full-size | 28.33 | 21.91 | 25.12 | 28.33 | 21.91 | 25.12 | 28.33 | 21.91 | 25.12 |
| Subset | 46.70 | 22.87 | 34.79 | 27.86 | 28.93 | 28.40 | 26.18 | 29.49 | 27.84 |
| DC | 54.15 | 28.07 | 41.11 | 26.45 | 23.25 | 24.85 | 20.32 | 24.69 | 22.50 |
| DSA | 46.93 | 26.25 | 36.59 | 24.61 | 23.19 | 23.90 | 31.95 | 22.06 | 27.00 |
| MTT | 28.78 | 23.71 | 26.25 | 24.94 | 21.86 | 23.40 | 28.37 | 21.75 | 25.06 |
| DM | 52.17 | 32.26 | 42.21 | 27.05 | 26.41 | 26.73 | 30.51 | 22.70 | 26.61 |
| IDM | 31.13 | 29.22 | 30.18 | 24.99 | 26.21 | 25.60 | 32.00 | 27.18 | 29.59 |
| BACON | 26.34 | 25.24 | 25.79 | 28.24 | 25.94 | 27.09 | 32.87 | 24.10 | 28.48 |

robustness and efficiency changes with different values of $\alpha$. This analysis provides guidance on selecting the appropriate weighting based on specific evaluation goals.

## B.9 ADVERSARIAL ROBUSTNESS ANALYSIS FOR NON-DISTILLED METHODS

We observe that dataset distillation methods significantly improve adversarial robustness compared to non-distilled baselines. Gradient-matching methods (e.g., DC, DSA) excel at lower IPC values by focusing on early-stage features, leading to higher CREI scores. In contrast, trajectory-matching (e.g., MTT) and distribution-matching methods (e.g., DM, IDM, BACON) perform better at higher IPC settings by capturing richer feature distributions, thereby enhancing robustness against adversarial perturbations. Across all IPC levels, DD methods consistently outperform full-size datasets and random subsets, demonstrating their ability to improve adversarial defense by focusing on informative features and reducing noise.

Empirical results in Table 15 confirm that DD methods outperform non-distilled baselines (full-size datasets and random subsets) across all metrics (RR, AE, CREI), validating the robustness benefits of dataset distillation under adversarial attacks. Further analysis of the Subset method across different IPC settings (Table 16) shows that while it performs reasonably well at IPC-1 and IPC-10, its performance drops significantly at IPC-50. This indicates that random subsets fail to capture the complex feature distributions needed for robust adversarial performance. In contrast, DD methods, especially distribution-matching methods, leverage richer and more complex feature information, allowing them to maintain higher robustness as IPC increases, explaining their superior performance at higher IPC levels.

## B.10 ATTACK CONVERGENCE ANALYSIS

Table 17: Accuracy under targeted PGD attacks with different iteration steps on CIFAR-10, IPC=50 (%).

| Method | Clean | PGD-1 | PGD-2 | PGD-3 | PGD-4 | PGD-5 | PGD-10 | PGD-15 | PGD-20 | PGD-25 | PGD-30 | PGD-35 | PGD-40 |
|---|---|---|---|---|---|---|---|---|---|---|---|---|---|
| Full-size | 87.41 | 55.85 | 51.20 | 45.11 | 41.26 | 39.88 | 37.60 | 37.44 | 37.33 | 37.33 | 37.23 | 37.34 | 37.23 |
| DC | 54.31 | 31.35 | 30.82 | 29.88 | 29.44 | 29.57 | 29.56 | 29.58 | 29.54 | 29.57 | 29.55 | 29.59 | 29.58 |
| DSA | 60.48 | 44.10 | 42.60 | 40.68 | 39.48 | 39.42 | 39.24 | 39.24 | 39.24 | 39.27 | 39.21 | 39.28 | 39.17 |
| MTT | 69.66 | 47.04 | 45.07 | 42.14 | 40.49 | 40.22 | 40.01 | 39.96 | 39.89 | 39.90 | 39.90 | 39.91 | 39.91 |
| DM | 62.06 | 43.17 | 41.74 | 40.03 | 38.94 | 38.87 | 38.67 | 38.69 | 38.68 | 38.69 | 38.71 | 38.69 | 38.67 |
| IDM | 66.91 | 46.63 | 45.01 | 43.65 | 42.79 | 43.15 | 43.19 | 43.22 | 43.18 | 43.22 | 43.20 | 43.22 | 43.19 |
| BACON | 69.95 | 49.13 | 47.75 | 45.95 | 45.25 | 45.65 | 45.58 | 45.56 | 45.57 | 45.56 | 45.57 | 45.57 | 45.56 |

Table 18: Accuracy under diverse targeted attacks on CIFAR-10, IPC = 50 (%).

| Attack | Full-size | DC | DSA | MTT | DM | IDM | BACON |
|---|---|---|---|---|---|---|---|
| Clean | 87.81 | 54.31 | 60.48 | 69.99 | 62.06 | 67.07 | 69.95 |
| FGSM | 46.65 | 29.14 | 40.07 | 41.36 | 39.45 | 42.97 | 44.68 |
| PGD | 37.36 | 29.56 | 39.24 | 40.02 | 38.66 | 43.91 | 45.59 |
| PGD_L2 | 56.42 | 32.09 | 45.71 | 48.87 | 44.53 | 48.92 | 50.81 |
| DeepFool | 87.81 | 11.26 | 17.21 | 18.76 | 16.96 | 17.65 | 18.28 |
| C&W | 31.47 | 41.13 | 37.68 | 41.46 | 40.93 | 45.18 | 44.71 |
| AutoAttack | 87.81 | 13.33 | 19.19 | 18.95 | 21.14 | 26.23 | 27.00 |

We analyze the convergence behavior of adversarial attacks applied to dataset distillation methods to ensure the chosen attack hyperparameters are effective and consistent across all evaluated methods. To maintain fairness, all methods were assessed under identical attack settings for PGD, FGSM, C&W, DeepFool, and AutoAttack. Table 17 shows that PGD attacks with 10 or more iterations have converged for all DD methods, as further increasing the number of iterations does not significantly reduce accuracy. This confirms that the chosen attack strength is sufficient for evaluating adversarial robustness. Additionally, Table 18 demonstrates that across a variety of attack types, the accuracy of models distilled by different DD methods consistently decreases compared to clean accuracy, indicating that the attacks are effective under our hyperparameter settings. These findings validate that our attack configurations are robust enough to provide reliable benchmarks for evaluating the adversarial resilience of DD methods, ensuring fair and meaningful comparisons.

## C LIMITATIONS, FUTURE DIRECTIONS, AND POTENTIAL IMPACT

### C.1 LIMITATIONS AND FUTURE PLANS

BEARD evaluates six representative DD methods and six adversarial attack strategies, but is currently limited by community availability in two main aspects. First, larger-scale distilled datasets (e.g., ImageNet) are not included, since our benchmark relies on open-source implementations to ensure fairness and reproducibility, and existing releases only support small- to medium-scale datasets such as CIFAR-10, CIFAR-100, and TinyImageNet. Privately reproducing large-scale results would require prohibitive computational resources and could introduce implementation bias. Second, the benchmark is confined to image classification, as publicly available distillation methods for other modalities (e.g., text, graphs, audio) are still immature or unavailable, making cross-domain evaluation presently infeasible. As the community develops more scalable and multimodal distillation methods, BEARD will be extended to incorporate larger datasets and more advanced adversarial attacks, thereby enabling a deeper and more systematic characterization of adversarial robustness across diverse domains. In addition, we plan to improve the evaluation metrics to better capture both worst-case and multi-attack performance, providing a more comprehensive and fair assessment of robustness across methods and IPC settings.

### C.2 PRACTICAL APPLICATIONS AND POTENTIAL IMPACT

The BEARD benchmark provides a standardized framework for evaluating the adversarial robustness of dataset distillation methods, supporting both research and practical deployment. It helps identify method-specific strengths and weaknesses, facilitating the development of more resilient techniques.

Given rising concerns over data security and privacy, BEARD also offers insights into the robustness of distilled datasets in adversarial settings.

## D  STATEMENT ON THE USE OF LARGE LANGUAGE MODELS

Large Language Models (LLMs) were used as a general-purpose assistive tool during the preparation of this paper. Specifically, LLMs were employed to help with language polishing, grammar correction, and minor rephrasing of sentences. All scientific content, experimental results, and interpretations were independently developed and verified by the authors. The authors take full responsibility for the accuracy and integrity of the content, including any text generated or edited with the assistance of LLMs. LLMs have not contributed to research ideation, experimental design, or analysis, and are not listed as authors.

