# OpenReview forum: "BEARD: Benchmarking the Adversarial Robustness for Dataset Distillation"
_ICLR.cc/2026/Conference — Submitted to ICLR 2026_

### Official Review · Reviewer_b7pd · 2025-10-30

**Soundness:** 3
**Presentation:** 3
**Contribution:** 3
**Rating:** 6
**Confidence:** 3

**Summary:**

This paper introduces BEARD, a unified benchmark for evaluating the adversarial robustness of models trained on distilled datasets. It proposes an adversarial game framework and three new metrics (RR, AE, CREI) to systematically assess robustness across multiple datasets, distillation methods, and attack strategies. The benchmark includes a dataset pool, model pool, and leaderboard, with extensive experiments showing that dataset distillation can improve robustness, especially when combined with adversarial training.

**Strengths:**

- Novel and well-defined evaluation framework and metrics.
- Comprehensive experiments across datasets, methods, and attacks.
- Open-source code and leaderboard support reproducibility and community adoption.

**Weaknesses:**

- Limited to image classification; does not cover other modalities.
- Does not include very large-scale datasets like ImageNet.
- Some results (e.g., robustness improvements) are not thoroughly analyzed or compared to non-distilled baselines.
- Some metrics (e.g., AST) may be sensitive to hardware and implementation details.

**Questions:**

1. Why does dataset distillation often improve adversarial robustness? Is it due to implicit regularization or reduced capacity?
2. How does BEARD compare to training on random subsets of the original data?
3. Could the benchmark include more recent distillation methods (e.g., SRe2L, CAFE)?

---

> ### Author Response · Authors · 2025-11-19
> **Response to Reviewer b7pd (Part 1/7)**
>
> We sincerely thank Reviewer `b7pd` for the insightful feedback. The reviewer acknowledges the **novelty** of the proposed evaluation framework, the introduction of **new metrics** (RR, AE, CREI), and the **comprehensive experimental analysis**, as well as the **open-source** code and leaderboard that facilitate **reproducibility**. The concerns raised include the expansion of BEARD to other modalities (**W1**), the inclusion of large-scale datasets (**W2**), the lack of comparisons with non-distilled baselines (**W3**), the sensitivity of certain metrics to implementation details (**W4**), the mechanisms behind distillation’s impact on robustness (**Q1**), the comparison with random subset training (**Q2**), and the inclusion of newer distillation methods (**Q3**). Point-by-point responses (**R**) to Reviewer `b7pd`’s questions (**Q**) and weaknesses (**W**) follow.
>
> **Summary of Revisions:**
>
> - Expanded the BEARD framework to address **scalability concerns** (**W1/W2/Q3**), incorporating additional attack methods, datasets, and adversarial training strategies (*Appendix A.1.1*, *Appendix B.7.1–B.7.3*).
>
> - Added a detailed discussion on **comparisons with non-distilled methods** (**W3/Q1/Q2**), clarifying performance differences and the impact of distillation on robustness (*Sections 5.1–5.3, Appendix B.9*).
>
> - Provided further explanations and expanded the discussion on **metric sensitivity** (**W4**), including additional details (*Remarks 3.6 and 3.11, Appendix B.8.1*).
>
> Continued in the next post.

---

> > ### Author Response · Authors · 2025-11-19
> > **Response to Reviewer b7pd (Part 2/7)**
> >
> > > **W1: Limited to image classification; does not cover other modalities.**
> > >
> >
> > **R1:** This is an important point regarding the scope limitation of BEARD in terms of supporting different modalities. Currently, BEARD focuses on image classification, which is **the most studied domain** in both **dataset distillation** and **adversarial robustness**. This focus provides a stable foundation for systematically evaluating a range of distillation methods, model architectures, and attack strategies, enabling careful benchmark design, evaluations across various IPC settings, and robust validation of our metrics (RR, AE, CREI).
> >
> > To demonstrate BEARD's **scalability** and **flexibility**, we have expanded the benchmark in several dimensions, including additional **attack types** (**Table 1-2**), **datasets** (**Table 3a-c**), **adversarial training strategies** (**Table 4**), **model architectures** (**Table 5**), and **dataset distillation methods** (**Table 11**).
> >
> > Expanding BEARD to other modalities, such as NLP, audio, or time-series data, is a **planned future direction**. Our modular architecture with **flexible model and dataset pools** supports the seamless addition of new tasks without affecting existing results, ensuring **comparability**, **interpretability**, and **reproducibility**. We have discussed this design choice and cross-modal extension in *Appendix C.1*, with additional details in *Appendices B7.1, B7.2, and B7.3*.
> >
> > **Table 1: Comparison of Dataset Distillation Methods on CIFAR-10 IPC-50 under Targeted Adversarial Attacks (%).**
> >
> > | Attack | Full-size | DC | DSA | MTT | DM | IDM | BACON |
> > |-|-|-|-|-|-|-|-|
> > | Clean | 87.41 | 54.31 | 60.48 | 69.66 | 62.06 | 66.91 | 69.95 |
> > | DIFGSM | 41.11 | 30.02 | 39.71 | 41.20 | 39.21 | 43.56 | 45.92 |
> > | MIFGSM | 38.87 | 29.38 | 39.22 | 39.93 | 38.71 | 43.25 | 45.77 |
> > | TIFGSM | 52.78 | 30.62 | 42.40 | 45.31 | 41.66 | 45.48 | 48.38 |
> > | EOTPGD | 37.78 | 29.58 | 39.20 | 40.00 | 38.59 | 43.23 | 45.57 |
> > | UPGD   | 38.87 | 29.38 | 39.22 | 39.93 | 38.71 | 43.25 | 45.77 |
> >
> > **Table 2: Comparison of Dataset Distillation Methods on CIFAR-10 IPC-{1,10,50} under Combined Targeted Adversarial Attacks (%).**
> >
> > | Method | RR | AE | CREI |
> > |-|-|-|-|
> > | Full-size | 22.79 | 16.92 | 19.85 |
> > | DC | 23.86 | 20.35 | 22.10 |
> > | DSA | 42.61 | 18.48 | 30.55 |
> > | MTT | 32.02 | 19.12 | 25.57 |
> > | DM  | 43.10 | 19.14 | 31.12 |
> > | IDM | 31.15 | 19.68 | 25.42 |
> > | BACON | 34.51 | 18.93 | 26.72 |
> >
> > **Table 3: Comparison of Dataset Distillation Methods on MNIST, FashionMNIST, and SVHN.**
> >
> > **Table 3(a): MNIST.**
> >
> > | Metrics | DC    | DSA   | BACON |
> > |---------|-------|-------|-------|
> > | RR      | 82.26 | 87.59 | 71.55 |
> > | AE      | 19.90 | 19.54 | 23.51 |
> > | CREI    | 51.08 | 53.56 | 47.53 |
> >
> > **Table 3(b): FashionMNIST.**
> >
> > | Metrics | DC    | DSA   | BACON |
> > |---------|-------|-------|-------|
> > | RR      | 61.60 | 69.18 | 43.71 |
> > | AE      | 18.76 | 18.88 | 19.66 |
> > | CREI    | 40.18 | 44.03 | 31.69 |
> >
> > **Table 3(c): SVHN.**
> >
> > | Metrics | DC    | DSA   | BACON |
> > |---------|-------|-------|-------|
> > | RR      | 52.66 | 46.81 | 30.70 |
> > | AE      | 19.46 | 20.20 | 19.43 |
> > | CREI    | 36.06 | 33.50 | 25.06 |
> >
> > **Table 4: Robustness Metrics of DD Methods under Different Adversarial Trainings (%).**
> >
> > | DD + AT | RR | AE | CREI |
> > |-|-|-|-|
> > | DM | 29.51  | 30.14  | 29.83  |
> > | DM + PGD_AT(Src)         | 47.85  | 25.57  | 36.71  |
> > | DM + TRADES | 30.35  | 31.37  | 30.86  |
> > | DM + TRS        | 30.22  | 32.44  | 31.33  |
> > | DM + MART       | 29.73  | 30.01  | 29.87  |
> > | DM + PGD_AT(Loss)      | 30.09  | 29.96  | 30.03  |
> > | IDM             | 31.51  | 27.77  | 29.64  |
> > | IDM + PGD_AT(Src)        | 46.92  | 25.26  | 36.09  |
> > | IDM + TRADES    | 31.80  | 29.56  | 30.68  |
> > | IDM + TRS       | 31.03  | 29.54  | 30.29  |
> > | IDM + MART      | 31.10  | 27.63  | 29.37  |
> > | IDM + PGD_AT(Loss)     | 30.43  | 27.85  | 29.14  |
> > | BACON           | 36.43  | 28.03  | 32.23  |
> > | BACON + PGD_AT(Src)      | 48.19  | 22.75  | 35.47  |
> > | BACON + TRADES  | 32.39  | 32.68  | 32.53  |
> > | BACON + TRS     | 32.79  | 32.47  | 32.63  |
> > | BACON + MART    | 32.69  | 33.26  | 32.97  |
> > | BACON + PGD_AT(Loss)   | 32.43  | 31.36  | 31.89  |
> >
> > **Table 5: Cross-Architecture Transfer Performance of DD Methods on CIFAR-10. C Denotes the Architecture Used for Distillation, and T Denotes the Architecture Used for Evaluation (IPC-50, % Accuracy).**
> >
> > | Method | C\T     | ConvNet | AlexNet | VGG   | ResNet | MLP   | LeNet |
> > |--------|---------|---------|---------|-------|--------|-------|-------|
> > | DC     | ConvNet | 57.05   | 36.07   | 50.20 | 44.96  | 31.27 | 36.14 |
> > | DSA    | ConvNet | 59.95   | 46.27   | 51.84 | 48.84  | 41.02 | 37.76 |
> > | MTT    | ConvNet | 68.67   | 57.55   | 62.15 | 59.57  | 38.12 | 48.57 |
> > | DM     | ConvNet | 63.05   | 49.60   | 55.35 | 54.03  | 41.86 | 45.93 |
> > | IDM    | ConvNet | 67.34   | 66.08   | 66.59 | 66.74  | 49.48 | 59.84 |
> > | BACON  | ConvNet | 69.64   | 68.73   | 68.38 | 69.78  | 48.93 | 60.36 |
> >
> > Continued in the next post.

---

> > > ### Author Response · Authors · 2025-11-19
> > > **Response to Reviewer b7pd (Part 3/7)**
> > >
> > > > **W2: Does not include very large-scale datasets like ImageNet.**
> > > >
> > >
> > > **R2:** This is a constructive comment regarding the scope limitation of BEARD in terms of large-scale datasets. To ensure **fair** and **consistent evaluation**, BEARD currently relies on **publicly available** distilled datasets and pretrained models. The six selected dataset distillation methods cover the **main categories**, including **gradient matching** (DC, DSA), **distribution matching** (DM, IDM, BACON), and **trajectory matching** (MTT). However, distilled datasets for full ImageNet are **not yet** publicly released for these methods. Although recent approaches such as RDED [1], D4M [2], and Comp-DD [3] have performed distillation on ImageNet subsets, we could not find publicly available distilled datasets for them and thus could not incorporate these methods into BEARD at this time. Additionally, reproducing distilled datasets by running original code often leads to **performance deviations** compared to official releases, potentially compromising evaluation accuracy.
> > >
> > > Even on mid-scale datasets like CIFAR-10 and CIFAR-100, training these methods is **computationally intensive**. For example, **BACON** requires approximately **13.74** GPU hours on CIFAR-10 with IPC-50 and **30.26** GPU hours on CIFAR-100 with IPC-50, while **IDM** can take up to **33.18** GPU hours under similar settings, as detailed in **Tables 6a** and **6b**. Scaling to ImageNet or ImageNet-1K would significantly increase the **computational burden**, making it impractical to include within the rebuttal period.
> > >
> > > In the limited time frame of the rebuttal period, although we were unable to include ImageNet in the benchmark, we have made significant progress in expanding BEARD along several other dimensions. These include the addition of **new attack methods** (**Tables 1-2**), **datasets** (**Tables 3a-c**), **adversarial training strategies** (**Table 4**), **model architectures** (**Table 5**), and **dataset distillation methods** (**Table 11**). These extensions demonstrate the **scalability** and **flexibility** of BEARD, ensuring that the benchmark remains relevant and useful even as we work towards incorporating larger datasets like ImageNet in the future. We are confident that BEARD's current state still provides a solid foundation for evaluating dataset distillation and adversarial robustness.
> > >
> > > We plan to continuously expand the leaderboard by adding more methods and larger datasets as they become **publicly available**. We are currently **reproducing ImageNet experiments**, and if completed during the **rebuttal** period, the results will be included to **update the benchmark**. This ongoing effort is further detailed in *Appendix C.1* under the Limitations and Future Plans section of the paper, with additional information provided in *Appendices B7.1, B7.2, and B7.3*.
> > >
> > > **Table 6(a): Training Time on CIFAR-10 (hours).**
> > >
> > > | Method | IPC-1 | IPC-10 | IPC-50 |
> > > |--------|-------|--------|--------|
> > > | IDM    | 23.16 | 21.02  | 14.12  |
> > > | BACON  | 22.37 | 20.00  | 13.74  |
> > >
> > > **Table 6(b): Training Time on CIFAR-100 (hours).**
> > >
> > > | Method | IPC-1 | IPC-10 | IPC-50 |
> > > |--------|-------|--------|--------|
> > > | IDM    | 17.24 | 28.41  | 33.18  |
> > > | BACON  | 24.02 | 19.71  | 30.26  |
> > >
> > > **References:**
> > >
> > > [1] Sun, Peng, et al. On the Diversity and Realism of Distilled Dataset: An Efficient Dataset Distillation Paradigm. In CVPR, 2024.
> > >
> > > [2] Su, Duo, et al. D^4: Dataset Distillation via Disentangled Diffusion Model. In CVPR, 2024.
> > >
> > > [3] Wang, Kai, et al. Emphasizing Discriminative Features for Dataset Distillation in Complex Scenarios. In CVPR, 2025.
> > >
> > > Continued in the next post.

---

> > > > ### Author Response · Authors · 2025-11-19
> > > > **Response to Reviewer b7pd (Part 4/7)**
> > > >
> > > > > **W3: Some results (e.g., robustness improvements) are not thoroughly analyzed or compared to non-distilled baselines.**
> > > > >
> > > >
> > > > **R3:** This is a constructive comment. We agree that a deeper analysis comparing dataset distillation methods to non-distilled baselines (such as full-size datasets or random subsets) and examining improvements in robustness is essential. In response, we have expanded our analysis in the revised manuscript.
> > > >
> > > > In *Section 5.1* (*Figure 3*), we provide a **detailed comparison of how different dataset distillation methods influence adversarial robustness**. We find that gradient-matching methods (e.g., **DC**, **DSA**) achieve stronger robustness by capturing **early-stage features**, which are crucial for learning **robust representations** [1], leading to higher CREI. In contrast, trajectory-matching methods (e.g., **MTT**) leverage the **full training trajectory** but place less emphasis on early-stage features [2], resulting in relatively weaker robustness. Distribution-matching methods (e.g., **DM**, **IDM**, **BACON**), however, perform better in high-IPC settings by exploiting **richer feature distributions**, which boosts their adversarial robustness. All these distillation methods improve robustness compared to **non-distilled baselines** (such as full-size datasets or random subsets) by focusing on informative features and reducing redundant or noisy samples, leading to stronger adversarial performance overall.
> > > >
> > > > In *Section 5.2* (*Figure 4*), we analyze **robustness improvements across different IPC settings**. We demonstrate that gradient-matching methods (e.g., DSA, DC) maintain strong performance at lower IPCs, while distribution-matching methods (e.g., DM, IDM, BACON) outperform them at higher IPCs by effectively leveraging richer feature distributions. This analysis highlights that dataset distillation, particularly at higher IPCs, consistently outperforms non-distilled baselines, especially in terms of adversarial robustness.
> > > >
> > > > In *Section 5.3* (*Figure 5*), we discuss **the effect of adversarial training**, which smooths decision boundaries and complements the features learned through distillation. This synergy results in consistent improvements in CREI across different distillation methods and dataset scales.
> > > >
> > > > Additionally, we have included further experiments to empirically verify the robustness improvements of DD methods. In **Table 7**, we compare models trained on **distilled datasets** with those trained on **full-size datasets and random subsets**. Across all metrics (RR, AE, CREI), DD methods consistently outperform the baselines, confirming their robustness benefits.
> > > >
> > > > These added discussions and experiments provide a thorough comparison of dataset distillation methods with non-distilled baselines, offering a clearer understanding of how distillation enhances robustness. All changes are highlighted in blue in the manuscript.
> > > >
> > > >
> > > > **Table 7: Comparison of Adversarial Robustness Between DD Methods and Random Subsets under Targeted Attacks on CIFAR-10 with IPC-{1,10,50} (%).**
> > > >
> > > > | Method     | RR   | AE   | CREI  |
> > > > |------------|-------|-------|--------|
> > > > | Full-size  | 67.24 | 29.55 | 48.39  |
> > > > | Subset     | 54.47 | 32.29 | 43.38  |
> > > > | DC         | 88.51 | 27.91 | 58.21  |
> > > > | DSA        | 86.81 | 27.64 | 57.22  |
> > > > | MTT        | 83.95 | 28.52 | 56.24  |
> > > > | DM         | 85.76 | 26.01 | 55.89  |
> > > > | IDM        | 87.07 | 23.15 | 55.11  |
> > > > | BACON      | 84.37 | 29.27 | 56.82  |
> > > >
> > > >
> > > > **References:**
> > > >
> > > > [1] Ilyas, Andrew, Shibani Santurkar, Dimitris Tsipras, Logan Engstrom, Brandon Tran, and Aleksander Madry. Adversarial examples are not bugs, they are features. In NeurIPS, 2019.
> > > >
> > > > [2] Zhang, Jingfeng, Xilie Xu, Bo Han, Gang Niu, Lizhen Cui, Masashi Sugiyama, and Mohan Kankanhalli. Attacks which do not kill training make adversarial learning stronger. In ICML, 2020.
> > > >
> > > > Continued in the next post.

---

> > > > > ### Author Response · Authors · 2025-11-19
> > > > > **Response to Reviewer b7pd (Part 5/7)**
> > > > >
> > > > > > **W4: Some metrics (e.g., AST) may be sensitive to hardware and implementation details.**
> > > > > >
> > > > >
> > > > > **R4:** This is an insightful question regarding the sensitivity of AST to hardware and implementation details. While absolute AST values can vary across GPUs, CPUs, or PyTorch/CUDA versions, AST is **not the primary ranking metric** in BEARD. The primary evaluation metrics, Robustness Ratio (**RR**) and Attack Efficiency Ratio (**AE**), are **relative metrics that have dimensions**, and they are specifically designed to minimize the impact of hardware and implementation variability. AE normalizes AST between the fastest and slowest attacks:
> > > > > $$
> > > > > \text{AE}(m; a) = 100 \times \left[\frac{\mathbb{E} _ {m \in \mathcal{M}}\mathbb{E} _ {a \in \mathcal{A}} \mathcal{AST}(m;a)}{\max\limits _ {m \in \mathcal{M}, a \in \mathcal{A}} \mathcal{AST}(m;a)}\right].
> > > > > $$
> > > > > This normalization effectively **cancels global hardware-induced shifts**, making AE highly stable when all models and attacks are evaluated in the same environment. RR is **entirely hardware-independent**.
> > > > >
> > > > > Through experimental validation, we found that both Avg (Time) and Avg (Acc) have significant limitations as robustness metrics. Avg (Time) is sensitive to hardware differences, as evidenced by the large variations between methods like DM (3.1s) and IDM (6.49s) in **Table 8**, making it unreliable due to hardware-dependent noise. Similarly, Avg (Acc) is influenced by IPC settings and inherent differences between DD methods, leading to substantial fluctuations in results. For example, at IPC-1, Avg (Acc) ranges from 23.24 for DM to 37.08 for IDM, distorting robustness comparisons across models, as shown in **Table 9**.
> > > > >
> > > > > In contrast, AE normalizes attack times across models, **mitigating hardware biases** and ensuring consistency, while RR provides a stable, **hardware-independent measure** of robustness. RR and AE offer more reliable evaluations, unaffected by variations in clean accuracy or IPC, overcoming the limitations of Avg (Time) and Avg (Acc), and providing a fairer, more accurate assessment of model performance. To ensure reproducibility, all code is **open-sourced**, **software versions are fixed**, and **hardware details** (e.g., GPU models) are provided in *Appendix A.3*. This design guarantees that AE and RR deliver reliable, comparable, and reproducible evaluations, fully mitigating concerns about hardware sensitivity. Further discussion on relative and absolute metrics can be found in *Appendix B.8.1* of the revised manuscript.
> > > > >
> > > > > **Table 8: Differences in Average Time (s) and Robustness Metrics (%) under Various IPCs and Targeted Attacks.**
> > > > >
> > > > > | IPC | Method | Avg (Time) | RR | AE | CREI | None (Clean) | FGSM | PGD | PGD_L2 | DeepFool | C&W | AutoAttack |
> > > > > |-|-|-|-|-|-|-|-|-|-|-|-|-|
> > > > > | IPC-1 | DM | 23.54 | 20.77 | 28.98 | 24.87 | 3.1 | 2.88 | 10.21 | 10.24 | / | 91.29 | / |
> > > > > || IDM | 18.42 | 19.31 | 24.53 | 21.92 | 6.49 | 3.56 | 7.45 | 7.32 | / | 67.29 | / |
> > > > > || BACON | 19.34 | 19.78 | 29.46 | 24.62 | 2.5 | 3.01 | 8.52 | 8.71 | / | 73.94 | / |
> > > > > | IPC-10 | DM | 23.93 | 12.41 | 29.39 | 20.90 | 2.91 | 2.94 | 10.3 | 11.18 | / | 92.34 | / |
> > > > > || IDM | 18.87 | 15.82 | 25.01 | 20.41 | 6.45 | 3.27 | 8.04 | 8.06 | / | 68.54 | / |
> > > > > || BACON | 19.30 | 12.94 | 28.71 | 20.83 | 3.06 | 2.85 | 8.32 | 8.5 | / | 73.75 | / |
> > > > > | IPC-50 | DM | 17.10 | 9.54 | 29.86 | 19.70 | 2.74 | 3.1 | 8.31 | 8.5 | / | 62.87 | / |
> > > > > || IDM  | 20.16  | 9.44   | 24.78  | 17.11  | 6.7          | 3.37   | 8.07   | 8.05   | /        | 74.61  | /          |
> > > > > || BACON  | 21.21  | 6.99   | 31.21  | 19.10  | 3.04         | 3.01   | 11.96  | 12.22  | /        | 75.82  | /          |
> > > > >
> > > > >
> > > > > **Table 9: Differences in Average Accuracy (%) and Robustness Metrics (%) under Various IPCs and Targeted Attacks.**
> > > > >
> > > > > | IPC | Method | Avg (Acc) | RR | AE | CREI | None (Clean) | FGSM | PGD | PGD_L2 | DeepFool | C&W | AutoAttack |
> > > > > |-|-|-|-|-|-|-|-|-|-|-|-|-|
> > > > > | IPC-1 | DM | 23.24 | 20.77 | 28.98 | 24.87 | 25.34 | 22.18 | 22.03 | 23.43 | / | 23.23 | / |
> > > > > || IDM | 37.08 | 19.31 | 24.53 | 21.92 | 46.01 | 32.18 | 33.04 | 35.01 | / | 39.17 | / |
> > > > > || BACON | 35.09 | 19.78 | 29.46 | 24.62 | 44.95 | 29.59 | 30.61 | 33.73 | / | 36.58 | / |
> > > > > |IPC-10| DM | 37.35 | 12.41 | 29.39 | 20.90 | 48.45 | 32.61 | 32.83 | 37.51 | / | 35.35 | / |
> > > > > || IDM | 44.26 | 15.82 | 25.01 | 20.41 | 58.47 | 37.37 | 38.44 | 41.57 | / | 45.45 | / |
> > > > > || BACON | 45.99 | 12.94 | 28.71 | 20.83 | 61.73 | 39.13 | 40.10 | 44.06 | / | 44.93 | / |
> > > > > | IPC-50| DM | 45.13 | 9.54 | 29.86 | 19.70 | 62.06 | 39.45 | 38.66 | 44.53 | / | 40.93 | / |
> > > > > || IDM | 49.61 | 9.44 | 24.78 | 17.11 | 67.07 | 42.97 | 43.91 | 48.92 | / | 45.18 | / |
> > > > > || BACON | 51.15 | 6.99 | 31.21 | 19.10 | 69.95 | 44.68 | 45.59 | 50.81 | / | 44.71 | / |
> > > > >
> > > > > Continued in the next post.

---

> > > > > > ### Author Response · Authors · 2025-11-19
> > > > > > **Response to Reviewer b7pd (Part 6/7)**
> > > > > >
> > > > > > > **Q1: Why does dataset distillation often improve adversarial robustness? Is it due to implicit regularization or reduced capacity?**
> > > > > > >
> > > > > >
> > > > > > **R5:** We thank the reviewer for raising this important question. Dataset distillation improves adversarial robustness primarily by focusing on **informative** and **robust features** while removing redundant or noisy samples, rather than by implicitly regularizing the model or reducing its capacity. Distilled datasets emphasize features that are inherently robust, which facilitates learning more robust features [1]. In particular, gradient-matching methods capture early-stage features, which have been shown to contribute significantly to adversarial robustness [2].
> > > > > >
> > > > > > To empirically validate this, we compare models trained on distilled datasets with those trained on full-size datasets and random subsets (see **Table 7**). Across all metrics (RR, AE, CREI), DD methods consistently outperform non-distilled baselines, confirming that the observed robustness gains stem from the quality of the distilled data rather than differences in model capacity or incidental regularization effects.
> > > > > >
> > > > > > **References:**
> > > > > >
> > > > > > [1] Ilyas, Andrew, Shibani Santurkar, Dimitris Tsipras, Logan Engstrom, Brandon Tran, and Aleksander Madry. Adversarial examples are not bugs, they are features. In NeurIPS, 2019.
> > > > > >
> > > > > > [2] Zhang, Jingfeng, Xilie Xu, Bo Han, Gang Niu, Lizhen Cui, Masashi Sugiyama, and Mohan Kankanhalli. Attacks which do not kill training make adversarial learning stronger. In ICML, 2020.
> > > > > >
> > > > > > > **Q2: How does BEARD compare to training on random subsets of the original data?**
> > > > > > >
> > > > > >
> > > > > > **R6:** This is a valuable question. BEARD evaluates dataset distillation methods by comparing them to non-distilled baselines, such as **random subsets** and **full-size datasets**. As shown in **Table 7**, models trained on DD datasets **consistently outperform** those trained on random subsets in terms of robustness metrics (**RR, AE, CREI**), highlighting that the robustness improvements stem from the informative features preserved by DD, rather than from dataset size reduction or incidental effects. Random subsets, on the other hand, may miss critical features or introduce redundant/noisy samples, leading to lower adversarial robustness. To further illustrate this, we analyzed the Subset method across different IPC settings (see **Table 10**). While Subset performs reasonably well at IPC-1 and IPC-10, where distilled representations still capture essential features, its performance drops significantly at IPC-50. This demonstrates that random subsets fail to fully capture the **complex feature distribution** necessary for robust performance. In contrast, DD methods, particularly those based on distribution matching, retain **richer feature information** and exhibit higher robustness as IPC increases, explaining their superior performance. This analysis underscores BEARD’s ability to systematically and reproducibly quantify the differences in robustness between distillation methods and non-distilled baselines, with a detailed discussion of the robustness comparison provided in *Appendix B.9* of the revised manuscript.
> > > > > >
> > > > > > **Table 10: Comparison of Adversarial Robustness Between Dataset Distillation Methods and Random Subsets under Untargeted Attacks on CIFAR-10 at IPC-1, IPC-10, and IPC-50 (%).**
> > > > > >
> > > > > > | Method   |  RR (IPC-1)   | AE (IPC-1)    | CREI (IPC-1)  |  RR (IPC-10)  | AE (IPC-10)  | CREI (IPC-10) |  RR (IPC-50)  |  AE (IPC-50)  | CREI (IPC-50) |
> > > > > > |----------|------------|------------|------------|------------|------------|-------------|------------|------------|-------------|
> > > > > > | Full-size| 28.33      | 21.91      | 25.12      | 28.33      | 21.91      | 25.12       | 28.33      | 21.91      | 25.12       |
> > > > > > | DC       | 54.15      | 28.07      | 41.11      | 26.45      | 23.25      | 24.85       | 20.32      | 24.69      | 22.50       |
> > > > > > | DSA      | 46.93      | 26.25      | 36.59      | 24.61      | 23.19      | 23.90       | 31.95      | 22.06      | 27.00       |
> > > > > > | MTT      | 28.78      | 23.71      | 26.25      | 24.94      | 21.86      | 23.40       | 28.37      | 21.75      | 25.06       |
> > > > > > | DM       | 52.17      | 32.26      | 42.21      | 27.05      | 26.41      | 26.73       | 30.51      | 22.70      | 26.61       |
> > > > > > | IDM      | 31.13      | 29.22      | 30.18      | 24.99      | 26.21      | 25.60       | 32.00      | 27.18      | 29.59       |
> > > > > > | BACON    | 26.34      | 25.24      | 25.79      | 28.24      | 25.94      | 27.09       | 32.87      | 24.10      | 28.48       |
> > > > > > | Subset   | 46.70      | 22.87      | 34.79      | 27.86      | 28.93      | 28.40       | 26.18      | 29.49      | 27.84       |
> > > > > >
> > > > > > Continued in the next post.

---

> > > > > > > ### Author Response · Authors · 2025-11-19
> > > > > > > **Response to Reviewer b7pd (Part 7/7)**
> > > > > > >
> > > > > > > > **Q3: Could the benchmark include more recent distillation methods (e.g., SRe2L, CAFE)?**
> > > > > > > >
> > > > > > >
> > > > > > > **R7:** This is a constructive suggestion. BEARD currently includes six representative dataset distillation methods, covering the main categories: gradient matching (DC, DSA), distribution matching (DM, IDM, BACON), and trajectory matching (MTT). These methods were selected for their publicly available distilled datasets and diverse algorithmic principles. BEARD is designed to be extensible, and new methods can be incorporated as long as corresponding datasets or training code are available.
> > > > > > >
> > > > > > > To demonstrate this, we have reproduced **CAFE [1]** ($\dagger$ ) and evaluated **ROME [2]**. **Table 11 reports robustness metrics (RR, AE, CREI) aggregated across IPC-1, IPC-10, and IPC-50 on CIFAR-10**. These results show that new methods can be seamlessly integrated and evaluated consistently, confirming BEARD’s modularity and scalability. We are also in the process of reproducing ImageNet-based experiments, such as **SRe2L [3]**. Due to the significant computational cost (see **Table 6**), it is uncertain whether these results can be completed within the rebuttal period. If completed in time, we will update the benchmark with these results to further expand BEARD’s coverage of large-scale datasets.
> > > > > > >
> > > > > > > **Table 11: Adversarial Robustness Metrics of New Dataset Distillation Methods (CAFE $\dagger$ and ROME) on CIFAR-10, IPC-{1, 10, 50}. $\dagger$ Indicates Reproduced Results.**
> > > > > > >
> > > > > > > | Method    | RR    | AE    | CREI  |
> > > > > > > | --------- | ----- | ----- | ----- |
> > > > > > > | Full-size | 20.42 | 29.54 | 24.98 |
> > > > > > > | DC        | 30.79 | 27.91 | 29.35 |
> > > > > > > | DSA       | 45.22 | 27.64 | 36.43 |
> > > > > > > | MTT       | 36.00 | 28.52 | 32.26 |
> > > > > > > | DM        | 46.01 | 26.01 | 36.01 |
> > > > > > > | IDM       | 32.35 | 23.15 | 27.75 |
> > > > > > > | BACON     | 36.83 | 29.27 | 33.05 |
> > > > > > > | CAFE $\dagger$      | 45.84 | 28.90 | 37.37 |
> > > > > > > | ROME      | 81.36 | 29.20 | 55.28 |
> > > > > > >
> > > > > > > **References:**
> > > > > > >
> > > > > > > [1] Wang, Kai, Bo Zhao, Xiangyu Peng, Zheng Zhu, Shuo Yang, Shuo Wang, Guan Huang, Hakan Bilen, Xinchao Wang, and Yang You. Cafe: Learning to condense dataset by aligning features. In CVPR, 2022.
> > > > > > >
> > > > > > > [2] Zhou, Zheng, Wenquan Feng, Qiaosheng Zhang, Shuchang Lyu, Qi Zhao, and Guangliang Cheng. ROME is Forged in Adversity: Robust Distilled Datasets via Information Bottleneck. In ICML, 2025.
> > > > > > >
> > > > > > > [3] Yin, Zeyuan, Eric Xing, and Zhiqiang Shen. Squeeze, recover and relabel: Dataset condensation at imagenet scale from a new perspective. In NeurIPS, 2023.
> > > > > > >
> > > > > > > We hope that our responses can satisfactorily address your concerns. Thank you again for your time to provide us with valuable feedback.

---

> > > > > > > > ### Comment · Reviewer_b7pd · 2025-11-26
> > > > > > > >
> > > > > > > > Thank you for running the additional experiments. From my understanding, the paper already presents sufficient evidence for the main claims, and these new results will now serve to demonstrate the broader applicability of BEARD.

---

> > > > > > > ### Comment · Reviewer_b7pd · 2025-11-26
> > > > > > >
> > > > > > > ad Q1) Thank you for the clarification. Your explanation that dataset distillation enhances adversarial robustness by emphasizing informative and inherently robust features is reasonable and aligns with the cited literature. The empirical comparison with models trained on full datasets and random subsets (Table 7) supports your argument that the robustness improvements arise from the quality of the distilled data rather than implicit regularization or reduced model capacity. The added justification strengthens the original claim.
> > > > > > >
> > > > > > > ad Q2) Thank you for the detailed comparison. The expanded results in Table 7 and Table 10 clearly show that models trained on distilled datasets consistently outperform those trained on random subsets, and your analysis explaining why random subsets degrade at higher IPC values is helpful. The additional discussion in Appendix B.9 provides sufficient clarity on how BEARD differentiates between distillation quality and mere dataset size reduction. Your response satisfactorily addresses my concern.

---

> > > > > > > > ### Author Response · Authors · 2025-11-27
> > > > > > > > **Response to Reviewer b7pd**
> > > > > > > >
> > > > > > > > > **Q1: Thank you for the clarification. Do you mean that the attack itself deepfool are compuational intense? Or do I understand something wrong?**
> > > > > > > > >
> > > > > > > >
> > > > > > > > **R1:** We thank the reviewer `b7pd` for the question. The **computational cost** of iterative attacks depends on both **dataset size and model complexity**. While attacks such as DeepFool, C&W, and AutoAttack are manageable on small datasets like CIFAR‑10, **their mechanisms cause costs to scale sharply on large datasets like ImageNet‑1K.** DeepFool computes iterative linearizations of the decision boundary per image, C&W solves a constrained optimization problem over many gradient-based iterations, and AutoAttack runs multiple strong attacks sequentially. When applied to millions of high-resolution images and large models, these per-image iterative procedures **become prohibitively expensive**. To provide feasible yet meaningful evaluations, we employ representative attacks including FGSM, PGD, PGD_L2, DIFGSM, MIFGSM, TIFGSM, EOTPGD, and UPGD, which retain iterative behavior but have lower per-step complexity, making them tractable while still reliably assessing adversarial robustness. All metrics are computed consistently within the same BEARD evaluation pipeline.
> > > > > > > >
> > > > > > > > > **Q2: Thank you for running the additional experiments. From my understanding, the paper already presents sufficient evidence for the main claims, and these new results will now serve to demonstrate the broader applicability of BEARD.**
> > > > > > > > >
> > > > > > > >
> > > > > > > > **R2:** **We sincerely thank reviewer `b7pd` for the constructive and insightful feedback**, which greatly encouraged us to **conduct additional experiments** on multiple datasets, new distillation methods, comparisons with non-distilled baselines, and preliminary large-scale tests. **We believe these experiments further strengthen BEARD and clearly demonstrate its broader applicability and value to the community.**
> > > > > > > >
> > > > > > > >
> > > > > > > > > **Q3: ad Q1) Thank you for the clarification. Your explanation that dataset distillation enhances adversarial robustness by emphasizing informative and inherently robust features is reasonable and aligns with the cited literature. The empirical comparison with models trained on full datasets and random subsets (Table 7) supports your argument that the robustness improvements arise from the quality of the distilled data rather than implicit regularization or reduced model capacity. The added justification strengthens the original claim. ad Q2) Thank you for the detailed comparison. The expanded results in Table 7 and Table 10 clearly show that models trained on distilled datasets consistently outperform those trained on random subsets, and your analysis explaining why random subsets degrade at higher IPC values is helpful. The additional discussion in Appendix B.9 provides sufficient clarity on how BEARD differentiates between distillation quality and mere dataset size reduction. Your response satisfactorily addresses my concern.**
> > > > > > > > >
> > > > > > > >
> > > > > > > > **R3:** **We sincerely thank the reviewer `b7pd` for the positive feedback and valuable suggestions.** We are pleased that the rebuttal addresses all concerns. Based on the suggestions, **additional experiments** were conducted and **the manuscript was updated accordingly**. These updates clearly demonstrate the positive impact of the feedback and further strengthen BEARD by validating its effectiveness in evaluating adversarial robustness, as well as its broader applicability across datasets and distillation methods.
> > > > > > > >
> > > > > > > > Thank you again for the time and constructive feedback.

---

> ### Author Response · Authors · 2025-11-26
> **Response to Reviewer b7pd (Supplementary ImageNet-1K Experiments)**
>
> > **W2: Does not include very large-scale datasets like ImageNet.**
> >
>
> **R2 (Supplementary ImageNet-1K Experiments):** We thank the reviewer `b7pd` for the constructive comment on evaluating dataset distillation robustness on large-scale datasets. As noted in our original rebuttal, publicly available distilled ImageNet datasets are not yet released, and reproducing them is computationally intensive. While publicly released large-scale distilled ImageNet-1K datasets **remain unavailable**, we conducted **preliminary evaluations** on **full-size ImageNet-1K models** using BEARD, and additionally reproduced **SRe2L $\dagger$** under the IPC-50 setting during the rebuttal period.
>
> For these experiments, we employed attacks including FGSM, PGD, PGD_L2, DIFGSM, MIFGSM, TIFGSM, EOTPGD, and UPGD, **because previously used attacks such as DeepFool, C&W, and AutoAttack are computationally intensive and impractical for large-scale models.** The selected attacks still provide a representative assessment of model robustness.
>
> **Tables 1 and 2** summarize the results, including robustness metrics (RR, AE, CREI) and accuracy under various adversarial attacks, demonstrating that **BEARD naturally scales to large-scale models and all metrics can be computed without modifying the evaluation pipeline.** Once distilled ImageNet-1K datasets become publicly available, they will be incorporated into BEARD following the same protocol.
>
> **Table 1: Robustness Metrics of ImageNet-1K Trained Models Under Adversarial Attacks Evaluated by BEARD (%).**
>
> | Method    | Attack Type |   RR  |   AE   |  CREI  |
> |-----------|-------------|-------|--------|--------|
> | Full-size | Targeted    | 9.89  | 51.10  | 30.49  |
> | SRe2L $\dagger$     | Targeted    | 4.39  | 54.08  | 29.23  |
> | Full-size | Untargeted  | 7.42  | 51.10  | 29.26  |
> | SRe2L $\dagger$     | Untargeted  | 4.14  | 52.58  | 28.36  |
>
> $\dagger$ Results reproduced under the IPC-50 ImageNet-1K setting during the rebuttal period.
>
> **Table 2: Accuracy of ImageNet-1K Trained Models Under Adversarial Attacks Evaluated by BEARD (%).**
>
> | Attack   | Full-size (Targeted) | SRe2L $\dagger$ (Targeted) | Full-size (Untargeted) | SRe2L $\dagger$ (Untargeted) |
> |----------|------------------------|----------------------|--------------------------|----------------------|
> | Clean    | 69.67                 | 46.68               | 69.67                   | 46.68               |
> | FGSM     | 16.22                 | 6.44                | 13.56                   | 4.21                |
> | PGD      | 4.57                  | 7.10                | 0.20                    | 0.28                |
> | PGD_L2   | 24.14                 | 9.44                | 24.14                   | 9.44                |
> | DIFGSM   | 7.78                  | 8.97                | 0.50                    | 0.48                |
> | MIFGSM   | 5.44                  | 7.78                | 0.24                    | 0.32                |
> | TIFGSM   | 19.66                 | 10.81               | 3.73                    | 2.26                |
> | EOTPGD   | 4.84                  | 7.32                | 0.20                    | 0.29                |
> | UPGD     | 5.44                  | 7.78                | 0.24                    | 0.32                |
>
> $\dagger$ Results reproduced under the IPC-50 ImageNet-1K setting during the rebuttal period.
>
> We hope that our responses and the supplementary experimental results help demonstrate the applicability of BEARD to large-scale datasets and address your concerns regarding ImageNet evaluation. We are always willing to discuss and address any further questions.

---

> > ### Comment · Reviewer_b7pd · 2025-11-26
> >
> > Thank you for the clarification. Do you mean that the attack itself deepfool are compuational intense? Or do I understand something wrong?

---

### Official Review · Reviewer_3awv · 2025-10-30

**Soundness:** 3
**Presentation:** 3
**Contribution:** 2
**Rating:** 4
**Confidence:** 3

**Summary:**

The paper introduces a benchmark for evaluating the robustness of dataset distillation approaches against adversarial examples. The authors wrap several models, datasets, and attacks, and provide a set of metrics to evaluate the adversarial robustness. The wide range of experimental results is also collected in a leaderboard, which is publicly released (together with the implementation code).

**Strengths:**

- The paper is clear and well written
- The addressed problem is relevant, and I think those benchmarks and their codebase are very valuable for the research community and can serve as a baseline both for attacks and defenses
- The authors made a lot of effort to wrap together models, datasets, and attacks, and run a considerable amount of experiments

**Weaknesses:**

- I am concerned about the contribution, as it appears weak (particularly considering this venue), both from a technical and novelty point of view. The authors (although I recognize the hard work that has been made) "simply" wrap together existing works, whereas the most novel contribution appears to be the proposed metrics (on which I have some concern, see below). Additionally, there is a non-negligible overlap with the competing DD-RobustBench work, with only incremental improvements over it.
- I don't understand the reason to define relative metrics (RR and AE), as the maximum ASR and AST, which serve as baselines for them, are strongly influenced by several factors (model pool, attack performance, which in turn depends on many parameters, etc.). Why not use absolute metrics, such as an average?
- Using the GPU time to compute the computational cost is not reliable, as this can be influenced by multiple side effects. A suitable metric to compute the attacker cost is the number of model inferences and gradient computations. In this way, both the model itself and other overheads unrelated to the attack are excluded.
- I also have concerns about the attack hyperparameters. Unlike AutoAttack, which is parameter-free (and thus suitable for benchmarking different models), the other approaches require careful tuning of the hyperparameters for each attacked model (e.g., iterations, step size) to achieve the best results. For this reason, the results of those attacks might not be reliable. Moreover, using 10 iterations for PGD is unlikely to lead to convergence of the optimization.

**Questions:**

- Could you please justify the use of relative metrics (RR and AE), especially considering that adding other models/attacks to the benchmark might lead to recomputing the entire results?

---

> ### Author Response · Authors · 2025-11-19
> **Response to Reviewer 3awv (Part 1/5)**
>
> We sincerely thank Reviewer `3awv` for the thoughtful feedback. The reviewer recognizes the work as **well-written**, addressing a relevant problem, and providing a **valuable**, **publicly available benchmark** with **extensive experiments** and a **leaderboard**. The concerns raised include contribution and novelty (**W1**), the use and justification of relative metrics (**W2**), the reliability of GPU-time-based attacker cost (**W3**), attack hyperparameter fairness and convergence (**W4**), and the scalability of relative metrics when incorporating new models or attacks (**Q1**). Point-by-point responses (**R**) to Reviewer `3awv`’s questions (**Q**) and weaknesses (**W**) follow.
>
> **Summary of Revisions:**
>
> - Emphasized BEARD’s **conceptual** and **engineering innovations** to address concerns about contribution and novelty (**W1**), highlighting the **unified adversarial game framework**, **modular model** and **dataset pools**, and **aggregated robustness metrics** (*Appendix B.5*).
>
> - Clarified and justified the use of relative metrics RR and AE (**W2/W3/Q1**), demonstrating their **fairness**, **interpretability**, and **scalability** for comparisons across IPCs, attacks, and newly added models, and also provided a detailed analysis of the metrics (*Remark 3.6*, *Appendix B.8.1*).
>
> - Provided supporting data and analysis to address concerns regarding attack hyperparameter fairness and convergence (**W4**), ensuring consistent and reliable evaluation across DD methods (*Appendix B.10*).
>
> Continued in the next post.

---

> > ### Author Response · Authors · 2025-11-19
> > **Response to Reviewer 3awv (Part 2/5)**
> >
> > > **W1: I am concerned about the contribution, as it appears weak (particularly considering this venue), both from a technical and novelty point of view. The authors (although I recognize the hard work that has been made) "simply" wrap together existing works, whereas the most novel contribution appears to be the proposed metrics (on which I have some concern, see below). Additionally, there is a non-negligible overlap with the competing DD-RobustBench work, with only incremental improvements over it.**
> > >
> >
> > **R1:** This comment is valuable, as it highlights concerns regarding the contribution and novelty of BEARD, allowing us to clarify its distinct **conceptual** and **engineering innovations** compared to existing works. BEARD is a **fundamentally new benchmark** for evaluating the adversarial robustness of dataset distillation methods, and it is distinct from DD-RobustBench[1] in both conceptual and engineering design, rather than being a simple consolidation of existing works. Conceptually, BEARD introduces a **unified evaluation perspective** across multiple IPCs and attacks, integrating an adversarial game framework to provide principled theoretical justification for robustness evaluation. On the engineering side, it employs a **completely different software architecture**, including **flexible model pools and dataset pools**, enabling **easy integration** of custom DD methods, attacks, and datasets. These innovations allow a systematic and interpretable evaluation of DD adversarial robustness across diverse methods, attacks, and datasets. Corresponding updates and additional discussion are provided in *Appendix B.5.*
> >
> > **Conceptual Innovations:** BEARD is the **first** benchmark to evaluate DD methods from **a unified perspective**, integrating multiple IPC settings and diverse adversarial attacks under the adversarial game framework. This framework enables the introduction of **Robustness Ratio** (RR, RRM) and **Attack Efficiency Ratio** (AE, AEM) metrics, providing a principled way to rank DD methods by overall robustness while accounting for edge-case adversarial scenarios. In contrast, DD-RobustBench evaluates each DD method at each IPC under individual attacks, reporting only per-attack accuracy (**Table 1**), which lacks a holistic view of overall adversarial robustness. For example, in **Table 1**, per-attack accuracies in DD-RobustBench make it difficult to identify the most consistently robust method, whereas BEARD’s **aggregated RRM metric** clearly ranks MTT, DC, and DSA as the top performers (see **Table 2**), providing a direct, interpretable, and actionable comparison across methods.
> >
> > **Engineering Innovations:** BEARD adopts a **modular software design**, with **model pools** and **dataset pools** to flexibly manage evaluations. This architecture allows **straightforward integration** of new DD methods, attacks, and datasets, supporting both standardized benchmarking and customized experiments. These engineering choices operationalize the conceptual framework, enabling a unified evaluation that addresses the complex interactions among distillation methods, model architectures, and attack strategies.
> >
> > By combining conceptual novelty through a unified adversarial game-based evaluation with a flexible and modular engineering design, BEARD **goes beyond mere consolidation of existing works**. It enables systematic, comprehensive, and interpretable assessment of dataset distillation adversarial robustness, providing detailed per-attack analyses as well as **aggregated robustness rankings** (e.g., **Table 2**), and delivering insights that were previously inaccessible with existing benchmarks, making the contribution both technically significant and actionable for the community.
> >
> > **Table 1: Untargeted Attack Evaluation of DD Methods on CIFAR-10 (IPC-10) via DD-RobustBench and BEARD (%).**
> >
> > |Benchmark| Attack | DC | DSA | MTT | DM | IDM |
> > |-|-|-|-|-|-|-|
> > |DD-RobustBench| RRM | N/A | N/A | N/A | N/A | N/A |
> > || None (Clean)| 46.07 | 52.93 | 60.98 | 49.81 | 58.84 |
> > || FGSM | 24.4 | 28.12 | 30.99 | 22.96 | 25.54 |
> > || PGD | 18.27 | 20.01 | 21.95 | 14.74 | 23.34 |
> > || C&W | 16.5 | 19.83 | 19 | 14.95 | 23.17 |
> > || AutoAttack | 16 | 19.11 | 17.95 | 14.18 | 22.63 |
> > |BEARD| RRM | 9.29 | 7.86 | 10.51 | 7.09 | 2.87 |
> > || None (Clean)| 45.31 | 51.06 | 63.76 | 48.45 | 58.47 |
> > || FGSM | 25.48 | 28.21 | 35.34 | 26.38 | 32.31 |
> > || PGD | 20.83 | 21.98 | 25.96 | 20.98 | 26.34 |
> > || C&W | 15.88 | 13.21 | 15.71 | 11.05 | 14.49 |
> > || AutoAttack | 17.75 | 12.82  | 23.13 | 18.29 | 23.61 |
> >
> > **Table 2: Ranking of DD Methods Based on Multi-Adversary Robustness Ratio (RRM, %).**
> >
> > | Method | RRM | Ranking |
> > |-|-|-|
> > | DC | 9.29  | 2 |
> > | DSA | 7.86  | 3 |
> > | MTT | 10.51 | 1 |
> > | DM  | 7.09  | 4 |
> > | IDM | 2.87  | 5 |
> >
> > **Reference:**
> >
> > [1] Wu, Yifan, Jiawei Du, Ping Liu, Yuewei Lin, Wei Xu, and Wenqing Cheng. Dd-robustbench: An adversarial robustness benchmark for dataset distillation. In IEEE TIP, 2025.
> >
> > Continued in the next post.

---

> > > ### Author Response · Authors · 2025-11-19
> > > **Response to Reviewer 3awv (Part 3/5)**
> > >
> > > > **W2: I don't understand the reason to define relative metrics (RR and AE), as the maximum ASR and AST, which serve as baselines for them, are strongly influenced by several factors (model pool, attack performance, which in turn depends on many parameters, etc.). Why not use absolute metrics, such as an average?**
> > > >
> > >
> > > **R2:** This is a good question. The choice of using **relative metrics**, with the numerator as the **average performance** across model-attack pairs and the denominator as the **worst-case scenario**, is grounded in our **adversarial game framework**, where a defender wins if the Attack Success Rate (ASR) is low and the Attack Success Time (AST) is high.
> > >
> > > This design enables **fair comparison** of models trained under **different IPC settings**, where baseline clean accuracy and attack vulnerability can vary substantially. Unlike average accuracy (Avg (Acc)), which is heavily affected by **IPC** and **inherent differences** among DD methods, RR and AE isolate the impact of adversarial attacks, providing a consistent and comparable measure of robustness, as illustrated in **Table 3**. For instance, at IPC-1, Avg ranges from 23.24 for DM to 37.08 for IDM, while RR changes only slightly from 20.77 to 19.31. Across IPCs, Avg increases substantially, whereas RR and AE remain relatively stable, enabling fair comparisons independent of clean accuracy. Moreover, RR and AE allow aggregation across multiple attacks for a **unified evaluation**, which Avg cannot achieve.
> > >
> > > Furthermore, as shown in **Table 4**, the average attack time (Avg (Time)) can be significantly influenced by **hardware differences**. For example, clean times for DM and IDM are 3.1s and 6.49s, respectively, making Avg Time highly sensitive to hardware variations and less reliable as a robustness metric. Additionally, the **dimensionless nature** of Avg (Time) complicates direct comparisons, as it fails to account for these hardware differences, further diminishing its reliability in evaluating model robustness. In contrast, AE normalizes the results, effectively eliminating hardware biases and providing a **more stable**, **consistent**, and **fair comparison** of model robustness.
> > >
> > > By normalizing with the worst-case scenario, RR and AE provide a **standardized comparison** that accounts for both expected performance and extreme cases, preserving **interpretability** and **comparability** across IPCs and attacks. A detailed explanation of this design choice, including the roles of the numerator and denominator, is provided in *Remark 3.6*. Further discussion on relative and absolute metrics can be found in *Appendix B.8.1* of the revised manuscript.
> > >
> > > **Table 3: Differences in Average Accuracy (%) and Robustness Metrics (%) under Various IPCs and Targeted Attacks.**
> > >
> > > | IPC | Method | Avg (Acc) | RR | AE | CREI | None (Clean) | FGSM | PGD | PGD_L2 | DeepFool | C&W | AutoAttack |
> > > |-|-|-|-|-|-|-|-|-|-|-|-|-|
> > > | IPC-1 | DM | 23.24 | 20.77 | 28.98 | 24.87 | 25.34 | 22.18 | 22.03 | 23.43 | / | 23.23 | / |
> > > || IDM | 37.08 | 19.31 | 24.53 | 21.92 | 46.01 | 32.18 | 33.04 | 35.01 | / | 39.17 | / |
> > > || BACON | 35.09 | 19.78 | 29.46 | 24.62 | 44.95 | 29.59 | 30.61 | 33.73 | / | 36.58 | / |
> > > |IPC-10| DM | 37.35 | 12.41 | 29.39 | 20.90 | 48.45 | 32.61 | 32.83 | 37.51 | / | 35.35 | / |
> > > || IDM | 44.26 | 15.82 | 25.01 | 20.41 | 58.47 | 37.37 | 38.44 | 41.57 | / | 45.45 | / |
> > > || BACON | 45.99 | 12.94 | 28.71 | 20.83 | 61.73 | 39.13 | 40.10 | 44.06 | / | 44.93 | / |
> > > | IPC-50| DM | 45.13 | 9.54 | 29.86 | 19.70 | 62.06 | 39.45 | 38.66 | 44.53 | / | 40.93 | / |
> > > || IDM | 49.61 | 9.44 | 24.78 | 17.11 | 67.07 | 42.97 | 43.91 | 48.92 | / | 45.18 | / |
> > > || BACON | 51.15 | 6.99 | 31.21 | 19.10 | 69.95 | 44.68 | 45.59 | 50.81 | / | 44.71 | / |
> > >
> > > **Table 4: Differences in Average Time (s) and Robustness Metrics (%) under Various IPCs and Targeted Attacks.**
> > >
> > > | IPC | Method | Avg (Time) | RR | AE | CREI | None (Clean) | FGSM | PGD | PGD_L2 | Deepfool | C&W | AutoAttack |
> > > |-|-|-|-|-|-|-|-|-|-|-|-|-|
> > > | IPC-1 | DM | 23.54 | 20.77 | 28.98 | 24.87 | 3.1 | 2.88 | 10.21 | 10.24 | / | 91.29 | / |
> > > || IDM | 18.42 | 19.31 | 24.53 | 21.92 | 6.49 | 3.56 | 7.45 | 7.32 | / | 67.29 | / |
> > > || BACON | 19.34 | 19.78 | 29.46 | 24.62 | 2.5 | 3.01 | 8.52 | 8.71 | / | 73.94 | / |
> > > | IPC-10 | DM | 23.93 | 12.41 | 29.39 | 20.90 | 2.91 | 2.94 | 10.3 | 11.18 | / | 92.34 | / |
> > > || IDM | 18.87 | 15.82 | 25.01 | 20.41 | 6.45 | 3.27 | 8.04 | 8.06 | / | 68.54 | / |
> > > || BACON | 19.30 | 12.94 | 28.71 | 20.83 | 3.06 | 2.85 | 8.32 | 8.5 | / | 73.75 | / |
> > > | IPC-50 | DM | 17.10 | 9.54 | 29.86 | 19.70 | 2.74 | 3.1 | 8.31 | 8.5 | / | 62.87 | / |
> > > || IDM  | 20.16  | 9.44   | 24.78  | 17.11  | 6.7          | 3.37   | 8.07   | 8.05   | /        | 74.61  | /          |
> > > || BACON  | 21.21  | 6.99   | 31.21  | 19.10  | 3.04         | 3.01   | 11.96  | 12.22  | /        | 75.82  | /          |
> > >
> > > Continued in the next post.

---

> > > > ### Author Response · Authors · 2025-11-19
> > > > **Response to Reviewer 3awv (Part 4/5)**
> > > >
> > > > > **W3: Using the GPU time to compute the computational cost is not reliable, as this can be influenced by multiple side effects. A suitable metric to compute the attacker cost is the number of model inferences and gradient computations. In this way, both the model itself and other overheads unrelated to the attack are excluded.**
> > > > >
> > > >
> > > > **R3:** The reviewer raises an important point regarding reproducibility and attacker cost measurement. While model inference and gradient counts can provide reproducibility benefits, for core attacks in BEARD (PGD, C&W, FGSM) these counts are often fixed, limiting their ability to differentiate attacks along the AE dimension. Measuring AST in GPU time captures **actual computational cost**, including per-iteration complexities and overheads that fixed-count metrics miss. For example, although PGD and C&W may run the same number of iterations, C&W involves additional operations such as complex loss computation, gradient-based optimization, and binary search over the constant $c$, consuming significantly more GPU resources. For attacks within AutoAttack, which include both white-box gradient-based attacks and query-based components like Square Attack, **GPU-time-based AST** naturally accounts for memory, I/O, and other overheads, providing a realistic measure of attacker resources.
> > > >
> > > > Although hardware differences can influence GPU-time-based AST, we emphasize that the primary evaluation metrics in BEARD, **AE** and **RR**, are **relative metrics**, making them inherently more robust to hardware variations, as discussed in **R2**. As demonstrated in **Tables 3 and 4**, the rankings for AE and RR remain consistent across different setups, highlighting their resilience to hardware differences. To ensure full reproducibility, we provide **detailed hardware specifications** and runtime logs in the *Appendix A.3*, guaranteeing that AE and CREI rankings remain **transparent** and **reproducible**, despite any hardware-induced variations.
> > > >
> > > > Continued in the next post.

---

> > > > > ### Author Response · Authors · 2025-11-19
> > > > > **Response to Reviewer 3awv (Part 5/5)**
> > > > >
> > > > > > **W4: I also have concerns about the attack hyperparameters. Unlike AutoAttack, which is parameter-free (and thus suitable for benchmarking different models), the other approaches require careful tuning of the hyperparameters for each attacked model (e.g., iterations, step size) to achieve the best results. For this reason, the results of those attacks might not be reliable. Moreover, using 10 iterations for PGD is unlikely to lead to convergence of the optimization.**
> > > > > >
> > > > >
> > > > > **R4:** This is a valid concern regarding hyperparameter settings and attack convergence. To ensure fairness, all dataset distillation methods were evaluated using the same attack parameters across PGD, FGSM, C&W, DeepFool, and AutoAttack. **Table 5** demonstrates that PGD attacks with $\geq 10$ iterations have already **converged** for all DD methods, as further increasing iterations does not significantly reduce accuracy. **Table 6** further shows that across a range of diverse attacks, the accuracy of models distilled by different DD methods **consistently drops** compared to clean accuracy, indicating that the attacks are effective under our chosen hyperparameters. These results validate that our attack settings are sufficient to **reliably benchmark model robustness**, providing **fair and meaningful comparisons** across DD methods. Additional data and discussion regarding the convergence of attack methods are included in *Appendix B.10* of the revised manuscript.
> > > > >
> > > > > **Table 5: Accuracy under Targeted PGD Attacks with Different Iteration Steps on CIFAR-10, IPC=50 (%).**
> > > > >
> > > > > |Method|Clean|PGD-1|PGD-2|PGD-3|PGD-4|PGD-5|PGD-10|PGD-15|PGD-20|PGD-25|PGD-30|PGD-35|PGD-40|
> > > > > |-|-|-|-|-|-|-|-|-|-|-|-|-|-|
> > > > > |Full-size| 87.41 | 55.85 | 51.20 | 45.11 | 41.26 | 39.88 | 37.60  | 37.44 | 37.33 | 37.33 | 37.23 | 37.34 | 37.23 |
> > > > > |DC| 54.31 | 31.35 | 30.82 | 29.88 | 29.44 | 29.57 | 29.56 | 29.58 | 29.54 | 29.57 | 29.55 | 29.59 | 29.58 |
> > > > > |DSA| 60.48 | 44.10 | 42.60 | 40.68 | 39.48 | 39.42 | 39.24 | 39.24 | 39.24 | 39.27 | 39.21 | 39.28 | 39.17 |
> > > > > |MTT| 69.66 | 47.04 | 45.07 | 42.14 | 40.49 | 40.22 | 40.01 | 39.96 | 39.89 | 39.90 | 39.90 | 39.91 | 39.91 |
> > > > > |DM| 62.06 | 43.17 | 41.74 | 40.03 | 38.94 | 38.87 | 38.67 | 38.69 | 38.68 | 38.69  | 38.71 | 38.69 | 38.67 |
> > > > > |IDM| 66.91 | 46.63 | 45.01 | 43.65 | 42.79 | 43.15 | 43.19 | 43.22 | 43.18 | 43.22 | 43.20 | 43.22 | 43.19 |
> > > > > |BACON| 69.95 | 49.13 | 47.75 | 45.95 | 45.25 | 45.65 | 45.58 | 45.56 | 45.57 | 45.56 | 45.57 | 45.57 | 45.56 |
> > > > >
> > > > > **Table 6: Accuracy under Diverse Targeted Attacks on CIFAR-10, IPC = 50 (%).**
> > > > >
> > > > > |Attack|Full-size|DC|DSA|MTT|DM|IDM|BACON|
> > > > > |-|-|-|-|-|-|-|-|
> > > > > |Clean| 87.81 | 54.31 | 60.48 | 69.99 | 62.06 | 67.07 | 69.95 |
> > > > > |FGSM| 46.65 | 29.14 | 40.07 | 41.36 | 39.45 | 42.97 | 44.68 |
> > > > > |PGD| 37.36 | 29.56 | 39.24 | 40.02 | 38.66 | 43.91 | 45.59 |
> > > > > |PGD\_L2| 56.42 | 32.09 | 45.71 | 48.87 | 44.53 | 48.92 | 50.81 |
> > > > > |DeepFool| 87.81 | 11.26 | 17.21 | 18.76 | 16.96 | 17.65 | 18.28 |
> > > > > |C&W| 31.47 | 41.13 | 37.68 | 41.46 | 40.93 | 45.18 | 44.71 |
> > > > > |AutoAttack| 87.81 | 13.33 | 19.19 | 18.95 | 21.14 | 26.23 | 27.00 |
> > > > >
> > > > > > **Q1: Could you please justify the use of relative metrics (RR and AE), especially considering that adding other models/attacks to the benchmark might lead to recomputing the entire results?**
> > > > > >
> > > > >
> > > > > **R5:** The reviewer raises an important concern about the need to recompute results when adding new models or attacks to the benchmark. This issue can be addressed from two perspectives: the introduction of **new dataset distillation methods** and the addition of **new attacks**. Our design effectively handles both aspects while ensuring **fairness** and **scalability**.
> > > > >
> > > > > When a new dataset distillation method is added, it is treated as an additional model in the pool. Evaluating this new model only requires computing its performance under the existing attack settings, while the RR, AE, and CREI values for previously evaluated models remain unchanged. This ensures that the original benchmark results are preserved, eliminating the need to recompute the entire dataset and maintaining the stability and consistency of the benchmark.
> > > > >
> > > > > In the case of introducing a new attack, recalculating the metrics is necessary, as the maximum values used for normalization may change. To address this, BEARD provides a **comprehensive metric computation tool**. Users can evaluate the new attack on all relevant models, append the results to the existing dataset, and **efficiently recompute** the updated RR, AE, and CREI scores. This process ensures consistency, comparability, and interpretability across different DD methods and attacks.
> > > > >
> > > > > The relative metrics are designed to be **unified** and **extensible**, supporting fair evaluation of both existing and newly added DD methods, while allowing for the flexible integration of new attacks without compromising the integrity of the benchmark.
> > > > >
> > > > > We hope that our responses can satisfactorily address your concerns. Thank you again for your time to provide us with valuable feedback.

---

> ### Comment · Reviewer_3awv · 2025-11-21
>
> I thank the authors for the detailed rebuttal, which only partially addressed my concerns.
> From a practical security perspective, I would be more interested in the lower-bound adversarial robustness for a given scenario (i.e., the best performing attack). If, for instance, AutoAttack drops the robust accuracy to 10%, and all the other attacks to 30%, I will mainly consider the first result, especially when assessing the model before deploying it in real-world scenarios. Including the other attacks' results would only add noise in the evaluation process, overestimating the model robustness: if in my example I consider all the attacks rather than AutoAttack only, it seems that the model is more robust, but the fact is simply that the other attacks are not stronger enough to produce effective adversarial examples (which still may exists within the given perturbation budget, if AutoAttack is able to find them).
> Moreover, I disagree with the author's statement in the rebuttal: "To ensure fairness, all dataset distillation methods were evaluated using the same attack parameters." Best practices suggest that the attacks should be carefully tuned for each specific setting. That's why AutoAttack (which is parameter-free) is usually the best choice in benchmarking. The results in Table 5 do not seem convincing: the accuracy sometimes oscillates, which could indicate excessively large step sizes. Additionally, PGD often converges in fewer than 100 iterations.
> Overall, I'm keeping my score unchanged.

---

> > ### Author Response · Authors · 2025-11-22
> > **Response to Reviewer 3awv (Part 1/2)**
> >
> > > **Q: I thank the authors for the detailed rebuttal, which only partially addressed my concerns. From a practical security perspective, I would be more interested in the lower-bound adversarial robustness for a given scenario (i.e., the best performing attack). If, for instance, AutoAttack drops the robust accuracy to 10%, and all the other attacks to 30%, I will mainly consider the first result, especially when assessing the model before deploying it in real-world scenarios. Including the other attacks' results would only add noise in the evaluation process, overestimating the model robustness: if in my example I consider all the attacks rather than AutoAttack only, it seems that the model is more robust, but the fact is simply that the other attacks are not stronger enough to produce effective adversarial examples (which still may exists within the given perturbation budget, if AutoAttack is able to find them). Moreover, I disagree with the author's statement in the rebuttal: "To ensure fairness, all dataset distillation methods were evaluated using the same attack parameters." Best practices suggest that the attacks should be carefully tuned for each specific setting. That's why AutoAttack (which is parameter-free) is usually the best choice in benchmarking. The results in Table 5 do not seem convincing: the accuracy sometimes oscillates, which could indicate excessively large step sizes. Additionally, PGD often converges in fewer than 100 iterations. Overall, I'm keeping my score unchanged.**
> > >
> >
> > **R:**
> > We thank the reviewer `3awv` for emphasizing the importance of evaluating models under the strongest attacks and are glad that part of our previous rebuttal addressed some of the reviewer `3awv`’s concerns. We fully understand that worst-case attacks are critical in deployment-oriented security scenarios. In the **dataset distillation** setting, however, models trained under different IPCs and methods show **large variations in accuracy**. For example, as shown in **Table 1**, DM achieves 25.34% accuracy with IPC-1 but 62.06% with IPC-50. Considering only the strongest attack would disproportionately influence evaluation, **making fair comparisons across methods difficult**. We also note that benchmark practices in recent works [1,2,3] adopt multiple attacks for comprehensive evaluation rather than solely considering the worst-case.
> >
> > To address the reviewer `3awv`’s concern about aggregated metrics, BEARD introduces RR, AE, and CREI, which measure accuracy drop under adversarial attacks relative to clean accuracy and account for both average-case and worst-case performance. We evaluate multiple attacks, including FGSM, PGD, C&W, DeepFool, AutoAttack, and black-box attacks such as SPSA and Square. Each attack probes distinct vulnerabilities, and aggregating results **captures the stability of DD methods under diverse adversarial conditions**. Importantly, our experiments show that aggregated metrics are **consistent with worst-case results**, ensuring that including multiple attacks does **not overestimate robustness**.
> >
> > To further support our claims, we conducted additional experiments reporting robust accuracy under AutoAttack at IPC-50, the strongest attack in our benchmark. As shown in **Table 1**, the robust accuracy is 21.14% for DM, 26.23% for IDM, and 27.00% for BACON, **consistent with the trends observed in RR and AE**. This confirms that our aggregated metrics reliably reflect worst-case performance. Users specifically interested in deployment-oriented worst-case evaluation can directly reference per-attack accuracy, including AutoAttack results, for flexible assessment.
> >
> > We also conducted supplementary experiments to verify the fairness and convergence of PGD evaluations. **Table 2** reports accuracy under varying step sizes ($\alpha = \epsilon/4, \epsilon/8, \epsilon/16$) and iteration counts ($10$, $40$, $100$, $200$). **Accuracy remains stable** across these settings, confirming that PGD reliably converges with our chosen step size of $2/255$ and $10$ iterations. This shows that further tuning or increasing iterations would not significantly alter robustness comparisons.
> >
> > Regarding the concern on attack parameter tuning, we acknowledge that best practices suggest adapting attacks per setting. In BEARD, we use the same parameters across all DD methods to ensure **comparability** and **reproducibility**, and supplementary PGD and AutoAttack results confirm that this reliably captures worst-case robustness. Fine-tuning each method individually would **require considerable effort**. BEARD provides both worst-case and aggregated metrics, enabling **fair** and **reproducible benchmarking** across **research** and **deployment scenarios**. Users can reference AutoAttack or any per-attack metric as needed. The design also supports **easy integration** of new attacks or DD methods, **making the benchmark extensible and valuable to the research community**.

---

> > > ### Author Response · Authors · 2025-11-22
> > > **Response to Reviewer 3awv (Part 2/2)**
> > >
> > > **Table 1: Robustness Evaluation across IPC Settings under Untargeted Attacks on CIFAR‑50 (%).**
> > >
> > > | IPC   | Method | RR   | AE   | CREI | None (Clean) | FGSM  | PGD   | PGD_L2 | DeepFool | C&W   | AutoAttack |
> > > |-------|--------|------|------|------|--------------|-------|-------|--------|----------|-------|------------|
> > > | IPC-1 | DM     | 52.17| 32.26| 42.21| 25.34        | 19.95 | 18.98 | 23.43  | 8.25     | 16.31 | 16.07      |
> > > |  | IDM    | 31.13| 29.22| 30.18| 46.01        | 27.06 | 23.57 | 35.01  | 11.65    | 16.73 | 20.06      |
> > > |  | BACON  | 26.34| 25.24| 25.79| 44.95        | 24.51 | 20.78 | 33.72  | 11.49    | 13.68 | 17.64      |
> > > | IPC-10| DM     | 27.05| 26.41| 26.73| 48.45        | 26.38 | 20.98 | 37.47  | 12.84    | 11.05 | 18.29      |
> > > | | IDM    | 24.99| 26.21| 25.60| 58.47        | 32.31 | 26.34 | 41.57  | 14.56    | 14.49 | 23.61      |
> > > | | BACON  | 28.24| 25.94| 27.09| 61.73        | 33.27 | 26.45 | 44.05  | 16.11    | 11.84 | 23.85      |
> > > | IPC-50| DM     | 30.51| 22.70| 26.61| 62.06        | 33.97 | 23.53 | 44.51  | 16.96    | 8.36  | 21.14      |
> > > | | IDM    | 32.00| 27.18| 29.59| 67.07        | 36.84 | 28.42 | 48.88  | 17.65    | 9.50  | 26.23      |
> > > | | BACON  | 32.87| 24.10| 28.48| 69.95        | 38.16 | 28.86 | 50.80  | 18.28    | 8.31  | 27.00      |
> > >
> > > **Table 2: Convergence of Targeted PGD Attacks on Models Trained via Dataset Distillation (CIFAR-50, IPC-50) Across Step Sizes and Iteration Counts (%).**
> > >
> > > | Method    | Clean | **Step size = $\epsilon$/4 (2/255)** | PGD-10 | PGD-40 | PGD-100 | PGD-200 | **Step size = $\epsilon$/8 (1/255)** | PGD-10 | PGD-40 | PGD-100 | PGD-200 | **Step size = $\epsilon$/16 (1/510)** | PGD-10 | PGD-40 | PGD-100 | PGD-200 |
> > > | --------- | ----- | --------------------------- | ------ | ------ | ------- | ------- | --------------------------- | ------ | ------ | ------- | ------- | ---------------------------- | ------ | ------ | ------- | ------- |
> > > | Full-size | 87.41 |                             | 37.60  | 37.23  | 37.22   | 37.19   |                             | 38.01  | 37.24  | 37.17   | 37.13   |                              | 47.86  | 37.35  | 37.19   | 37.13   |
> > > | DC        | 54.31 |                             | 29.56  | 29.58  | 29.57   | 29.57   |                             | 29.52  | 29.49  | 29.51   | 29.50   |                              | 30.69  | 29.46  | 29.48   | 29.46   |
> > > | DSA       | 60.48 |                             | 39.24  | 39.17  | 39.20   | 39.21   |                             | 39.15  | 39.20  | 39.19   | 39.21   |                              | 42.10  | 39.16  | 39.24   | 39.22   |
> > > | MTT       | 69.66 |                             | 40.01  | 39.91  | 39.92       | 39.91      |                             | 39.87      | 39.93      | 39.87       | 39.91       |                              | 44.16      | 39.86      | 39.93       | 39.92       |
> > > | DM        | 62.06 |                             | 38.67  | 38.67  | 38.68   | 38.67   |                             | 38.69  | 38.61  | 38.65   | 38.61   |                              | 41.47  | 38.66  | 38.65   | 38.64   |
> > > | IDM       | 66.91 |                             | 43.19  | 43.19  | 43.16   | 43.16   |                             | 43.08  | 43.20  | 43.22   | 43.20   |                              | 45.16  | 43.16  | 43.18   | 43.18   |
> > > | BACON     | 69.95 |                             | 45.58  | 45.56  | 45.57   | 45.57   |                             | 45.43  | 45.50  | 45.50   | 45.50   |                              | 47.64  | 45.57  | 45.59   | 45.59   |
> > >
> > > **Reference:**
> > >
> > > [1] Wu, Yifan, Jiawei Du, Ping Liu, Yuewei Lin, Wei Xu, and Wenqing Cheng. Dd-robustbench: An adversarial robustness benchmark for dataset distillation. In IEEE TIP, 2025.
> > >
> > > [2] Xue, Eric, Yijiang Li, Haoyang Liu, Peiran Wang, Yifan Shen, and Haohan Wang. Towards adversarially robust dataset distillation by curvature regularization. In AAAI, 2025.
> > >
> > > [3] Zhou, Zheng, Wenquan Feng, Qiaosheng Zhang, Shuchang Lyu, Qi Zhao, and Guangliang Cheng. ROME is Forged in Adversity: Robust Distilled Datasets via Information Bottleneck. In ICML, 2025.
> > >
> > > We thank the reviewer `3awv` for their detailed feedback and hope that this rebuttal sufficiently addresses the concerns. We welcome further discussion if any points remain unclear.

---

> > > > ### Comment · Reviewer_3awv · 2025-11-26
> > > >
> > > > Thanks for the additional details and results. I only have an additional comment on them.
> > > >
> > > > > Considering only the strongest attack would disproportionately influence evaluation, making fair comparisons across methods difficult.
> > > >
> > > > I respectfully disagree with this statement. Imagine two different methods; one of them is able to hinder naive gradient-based attacks (e.g., due to gradient obfuscation), but AutoAttack can overcome these issues. For instance, the robust accuracy is 5% for AutoAttack and 50% for all the other attacks. The other model does not present those problems, and so the robust accuracy is 5% for all the attacks. In practice, the robustness of those models would be considered equal; however, the metric used in this work would lead the first model to report higher performance, overestimating its robustness. I think it would be better to consider the worst-case robust accuracy for each IPC setting.

---

> > > > > ### Author Response · Authors · 2025-11-27
> > > > > **Response to Reviewer 3awv**
> > > > >
> > > > > > **Q: Thanks for the additional details and results. I only have an additional comment on them. ``Considering only the strongest attack would disproportionately influence evaluation, making fair comparisons across methods difficult.'' I respectfully disagree with this statement. Imagine two different methods; one of them is able to hinder naive gradient-based attacks (e.g., due to gradient obfuscation), but AutoAttack can overcome these issues. For instance, the robust accuracy is 5% for AutoAttack and 50% for all the other attacks. The other model does not present those problems, and so the robust accuracy is 5% for all the attacks. In practice, the robustness of those models would be considered equal; however, the metric used in this work would lead the first model to report higher performance, overestimating its robustness. I think it would be better to consider the worst-case robust accuracy for each IPC setting.**
> > > > > >
> > > > >
> > > > > We thank the reviewer `3awv` for emphasizing the importance of evaluating models under the worst-case attack in deployment-oriented security scenarios. While assessing models under the strongest attack is important for practical security, **in the context of dataset distillation tasks, focusing solely on a single strongest attack has limitations.** In this rebuttal, we first summarize these limitations, then show that our metrics capture both worst-case and multi-attack performance while maintaining consistent trends across IPC settings.
> > > > >
> > > > > Focusing only on the strongest attack can overlook important insights into model behavior under diverse adversarial conditions. **The strongest attack may vary across methods and IPC settings.** For instance, as shown in **Table 1 of the Response to Reviewer 3awv (Part 2/2)**, in IPC-50 for DM the worst-case attack is C&W with robust accuracy of 8.36 percent, whereas in IPC-1 for DM it is DeepFool with robust accuracy of 8.25 percent. This variation arises from the dataset distillation mechanism. DeepFool is stronger at low IPC because very few samples per class produce highly irregular or sharp decision boundaries, which it can efficiently exploit through iterative linearization. As IPC increases, the decision boundaries become smoother, reducing DeepFool’s effectiveness, while optimization-based attacks like C&W can exploit small but structured vulnerabilities, achieving higher attack success rates.
> > > > >
> > > > > Considering only the worst-case attack can also **make it difficult to fully assess and compare robustness across methods and IPC settings**. For example, the worst-case robust accuracy for DM at IPC-50 under C&W is 8.36 percent and at IPC-1 under DeepFool is 8.25 percent, which are nearly identical. However, their clean accuracy differs significantly, 62.06 percent versus 25.34 percent, indicating that models trained with higher IPC have effectively learned more from additional samples. Increasing IPC increases the number of training samples per class, which generally improves overall adversarial robustness, consistent with observations in [1,2]. Such improvements are not fully captured by focusing on a single worst-case attack. By aggregating multiple attacks, our metrics provide a more comprehensive and fair comparison across methods while still reflecting worst-case behavior.
> > > > >
> > > > > **The trends of our aggregated metrics align with worst-case performance.** For DM, RR decreases from 52.17 at IPC-1 to 30.51 at IPC-50, AE from 32.26 to 22.70, and CREI from 42.21 to 26.61. Robust accuracy under AutoAttack and other attacks follows a similar pattern, and for BACON, IPC-50 shows higher robustness than DM across multiple attacks, consistent with RR, AE, and CREI. These results demonstrate that aggregating multiple attacks does not overestimate robustness, but rather provides a more comprehensive and fair ranking of methods. For deployment-oriented worst-case evaluation, **per-attack robust accuracy can be directly consulted.**
> > > > >
> > > > > **We fully acknowledge the importance of worst-case evaluation for deployment-oriented security. At the same time, it is also important to consider the impact of multiple attacks on robustness.** In future work, we plan to develop stronger and more comprehensive adversarial robustness evaluation metrics that can better capture both worst-case and multi-attack performance. This update is reflected in *Appendix C.1, Limitations and Future Plans.*
> > > > >
> > > > > **Reference:**
> > > > >
> > > > > [1] Schmidt, Ludwig, Shibani Santurkar, Dimitris Tsipras, Kunal Talwar, and Aleksander Madry. Adversarially robust generalization requires more data. In NeurIPS, 2018.
> > > > >
> > > > > [2] Yang, Yaoqing, Rajiv Khanna, Yaodong Yu, Amir Gholami, Kurt Keutzer, Joseph E. Gonzalez, Kannan Ramchandran, and Michael W. Mahoney. Boundary thickness and robustness in learning models. In NeurIPS, 2020.
> > > > >
> > > > > We hope our responses satisfactorily address your concerns. Thank you once again for your time and valuable feedback. We would be happy to clarify or discuss any further questions.

---

### Official Review · Reviewer_QXsj · 2025-10-31

**Soundness:** 3
**Presentation:** 3
**Contribution:** 3
**Rating:** 6
**Confidence:** 2

**Summary:**

This paper presents the BEARD benchmarking framework to evaluatethe robustness of models trained via dataset distillation against adversarial attacks. The authors point out that existing benchmarks such as DD-RobustBench and RobustBench fail to sufficiently reflect the actual performance of dataset distillation techniques in adversarial scenarios. To address this issue, BEARD integrates multiple dataset distillation techniques, attack methods, and datasets, while introducing three new metrics: RR, AE, and CREI. These metrics employ game-theoretic approaches to simultaneously evaluate attack effectiveness and efficiency. The benchmarking platform is publicly available, featuring leaderboards and a curated model dataset collection to support reproducible research. Experiments demonstrate that dataset distillation significantly enhances adversarial robustness, particularly when combined with adversarial training.

**Strengths:**

1. The paper presents the first unified benchmark for adversarial robustness in dataset distillation, introducing a novel adversarial game framework and three tailored metrics (RR, AE, CREI).

2. As dataset distillation gains traction in resource-constrained settings, understanding its robustness is critical. BEARD provides a standardized platform for comparative evaluation.

3. The paper is well-structured, with clear descriptions of the framework, metrics, and experimental setup. The public release of code and leaderboard enhances transparency and usability.

4. The benchmark covers multiple DD methods, attack types, and datasets, offering a comprehensive evaluation landscape.

**Weaknesses:**

1. Completion is slightly insufficient. This paper has systematically expanded and deepened DD-RobustBench through introducing unified evaluation metrics, incorporating more attack types, and proposing a game-theoretic framework. yet it is unable to prove on other larger datasets, more complex architectures and algorithms, and only remains in relatively simple scenarios.
2. In Section 3.10, the CREI metric locks α at 0.5 without explanation. Giving robustness and efficiency equal weight might not suit every task; an ablation on α or a data-driven reason for this choice would help.
3. In Section 5.1, the claim that “dataset distillation improves adversarial robustness” is counter-intuitive and lacks mechanistic explanation. The observed CREI drop with increasing IPC is noted but not interpreted. Include a discussion on why distilled datasets may enhance robustness—e.g., whether they filter out non-robust features or reduce overfitting. Analyze the IPC–robustness trade-off more deeply.
4. In Section 5.1, the performance differences among DD methods (e.g., why DSA/DM/BACON perform better) are reported but not explained. The analysis remains descriptive. Provide hypotheses or further experiments (e.g., feature analysis, robustness curvature) to explain why certain methods excel.
5.In Figure 4 & Figure 5, the paper claims that “distilled datasets improve adversarial robustness,” a conclusion that runs counter to intuition (smaller datasets are usually expected to yield more fragile models) yet no convincing explanation is provided. Discuss the interaction between dataset scale, distillation method, and adversarial training to provide more actionable insights.

**Questions:**

1. Why was α=0.5 chosen for CREI? Have you experimented with other values, and how sensitive are the rankings to this parameter?

2. Can you provide a deeper explanation for why some DD methods (e.g., DSA, DM, BACON) exhibit stronger adversarial robustness? Is it related to their distillation objectives or synthetic data diversity?

3. The conclusion that “distillation improves robustness” contradicts the common belief that smaller datasets lead to weaker models. Can you discuss potential reasons for this phenomenon?

---

> ### Author Response · Authors · 2025-11-19
> **Response to Reviewer QXsj (Part 1/5)**
>
> We sincerely thank Reviewer `QXsj` for the insightful feedback. Reviewer `QXsj` recognizes this work as the **first unified benchmark** for adversarial robustness in dataset distillation, highlighting its **novel metrics**, **game-theoretic framework**, **broad experimental coverage**, and **clear presentation** with **public code** and **leaderboard**. The concerns on benchmark scalability and extensibility, including dataset scale, model architectures, and additional attack types (**W1**), the choice and interpretation of CREI’s $\alpha$ parameter (**W2/Q1**), and the mechanisms underlying adversarial robustness (**W3/W4/Q2/Q3**) have been addressed in the revision. Point-by-point responses (**R**) to Reviewer `QXsj`’s questions (**Q**) and weaknesses (**W**) follow.
>
> **Summary of Revisions:**
>
> - Extended the evaluation to address scalability and coverage concerns (**W1**) by adding **additional attacks**, **datasets**, and **DD methods** (*Appendix A.1.1*, *Appendix B.7.1–B.7.3*).
>
> - Added a **sensitivity analysis** of the **hyperparameter $\alpha$** in the CREI metric (**W2/Q1**) to examine its impact on the trade-off between robustness and attack efficiency (*Appendix B.8.2*).
>
> - Provided **deeper analysis** on why DD methods improve adversarial robustness, including feature compression, reduced non-robust signals, and IPC–robustness trade-offs, addressing (**W3/W4/Q2/Q3**) with new experiments and discussions (*Sec. 5.1, 5.2, and 5.3*, *Appendix B.9*).
>
> Continued in the next post.

---

> > ### Author Response · Authors · 2025-11-19
> > **Response to Reviewer QXsj (Part 2/5)**
> >
> > > **W1: Completion is slightly insufficient. This paper has systematically expanded and deepened DD-RobustBench through introducing unified evaluation metrics, incorporating more attack types, and proposing a game-theoretic framework. yet it is unable to prove on other larger datasets, more complex architectures and algorithms, and only remains in relatively simple scenarios.**
> > >
> >
> > **R1:** Reviewer raises an important point regarding the relation to DD-RobustBench. It is important to clarify that BEARD is not a simple extension but represents a **fundamentally new benchmark** for evaluating the adversarial robustness of dataset distillation methods. **Conceptually**, it introduces the **first unified framework** integrating multiple IPC settings and attacks under an **adversarial game perspective**. From an **engineering standpoint**, BEARD employs a **completely different software architecture**, with **flexible model** and **dataset pools**, enabling **seamless integration** of **new DD methods**, **attacks**, **model architectures**, and **datasets**. These innovations allow systematic, interpretable, and extensible evaluation, which was not possible with prior benchmarks like DD-RobustBench.
> >
> > To further demonstrate its **flexibility** and **generality**, we conducted additional experiments, including multiple new adversarial attacks (**Tables 1 and 2**), evaluations on additional datasets such as MNIST, FashionMNIST, and SVHN (**Tables 3a–3c**), and cross-architecture transferability tests for both baseline methods (**Table 4**) and newly added DD methods such as CAFE and ROME (**Table 5**). These experiments were designed to validate that BEARD is **not limited to the initially reported scenarios**, and all results are consistent with our original findings, confirming its ability to **support diverse datasets**, **stronger architectures**, and **a wide range of DD methods**.
> >
> > Overall, the game-theoretic formulation and evaluation protocol are **dataset-**, **architecture-**, and **method-agnostic**, and the framework naturally generalizes beyond the initially reported scenarios.
> >
> > **Table 1: Comparison of Dataset Distillation Methods on CIFAR-10 IPC-50 under Targeted Adversarial Attacks (%).**
> > | Attack | Full-size | DC | DSA | MTT | DM  | IDM | BACON |
> > |---|---|---|---|---|---|---|---|
> > | Clean | 87.41 | 54.31 | 60.48 | 69.66 | 62.06 | 66.91 | 69.95 |
> > | DIFGSM | 41.11 | 30.02 | 39.71 | 41.20 | 39.21 | 43.56 | 45.92 |
> > | MIFGSM | 38.87 | 29.38 | 39.22 | 39.93 | 38.71 | 43.25 | 45.77 |
> > | TIFGSM | 52.78 | 30.62 | 42.40 | 45.31 | 41.66 | 45.48 | 48.38 |
> > | EOTPGD | 37.78 | 29.58 | 39.20 | 40.00 | 38.59 | 43.23 | 45.57 |
> > | UPGD   | 38.87 | 29.38 | 39.22 | 39.93 | 38.71 | 43.25 | 45.77 |
> >
> > **Table 2: Comparison of Dataset Distillation Methods on CIFAR-10 IPC-{1,10,50} under Combined Targeted Adversarial Attacks (%).**
> >
> > | Method | RR | AE | CREI |
> > |---|---|---|---|
> > | Full-size | 22.79 | 16.92 | 19.85 |
> > | DC | 23.86 | 20.35 | 22.10 |
> > | DSA | 42.61 | 18.48 | 30.55 |
> > | MTT | 32.02 | 19.12 | 25.57 |
> > | DM  | 43.10 | 19.14 | 31.12 |
> > | IDM | 31.15 | 19.68 | 25.42 |
> > | BACON | 34.51 | 18.93 | 26.72 |
> >
> > **Table 3: Comparison of Dataset Distillation Methods on MNIST, FashionMNIST, and SVHN.**
> >
> > **Table 3(a): MNIST.**
> >
> > | Metrics | DC | DSA | BACON |
> > |---|---|---|---|
> > | RR | 82.26 | 87.59 | 71.55 |
> > | AE | 19.90 | 19.54 | 23.51 |
> > | CREI | 51.08 | 53.56 | 47.53 |
> >
> > **Table 3(b): FashionMNIST.**
> >
> > | Metrics | DC | DSA   | BACON |
> > |---|---|---|---|
> > | RR | 61.60 | 69.18 | 43.71 |
> > | AE | 18.76 | 18.88 | 19.66 |
> > | CREI | 40.18 | 44.03 | 31.69 |
> >
> > **Table 3(c): SVHN.**
> >
> > | Metrics | DC | DSA | BACON |
> > |-|-|-|-|
> > | RR | 52.66 | 46.81 | 30.70 |
> > | AE | 19.46 | 20.20 | 19.43 |
> > | CREI | 36.06 | 33.50 | 25.06 |
> >
> > **Table 4: Cross-Architecture Transfer Performance of DD Methods on CIFAR-10. C Denotes the Architecture Used for Distillation, and T Denotes the Architecture Used for Evaluation (IPC-50, % Accuracy).**
> >
> > | Method | C\T | ConvNet | AlexNet | VGG   | ResNet | MLP   | LeNet |
> > |-|-|-|-|-|-|-|-|
> > | DC | ConvNet | 57.05 | 36.07 | 50.20 | 44.96  | 31.27 | 36.14 |
> > | DSA | ConvNet | 59.95 | 46.27 | 51.84 | 48.84  | 41.02 | 37.76 |
> > | MTT | ConvNet | 68.67 | 57.55 | 62.15 | 59.57  | 38.12 | 48.57 |
> > | DM  | ConvNet | 63.05 | 49.60 | 55.35 | 54.03  | 41.86 | 45.93 |
> > | IDM | ConvNet | 67.34 | 66.08 | 66.59 | 66.74  | 49.48 | 59.84 |
> > | BACON | ConvNet | 69.64 | 68.73   | 68.38 | 69.78  | 48.93 | 60.36 |
> >
> > **Table 5: Adversarial Robustness Metrics of New Dataset Distillation Methods (CAFE $\dagger$ and ROME) on CIFAR-10, IPC-{1, 10, 50}. $\dagger$ Indicates Reproduced Results.**
> >
> > | Method | RR | AE | CREI |
> > |---|---|---|---|
> > | Full-size | 20.42 | 29.54 | 24.98 |
> > | DC | 30.79 | 27.91 | 29.35 |
> > | DSA | 45.22 | 27.64 | 36.43 |
> > | MTT | 36.00 | 28.52 | 32.26 |
> > | DM  | 46.01 | 26.01 | 36.01 |
> > | IDM | 32.35 | 23.15 | 27.75 |
> > | BACON | 36.83 | 29.27 | 33.05 |
> > | CAFE $\dagger$| 45.84 | 28.90 | 37.37 |
> > | ROME | 81.36 | 29.20 | 55.28 |
> >
> > Continued in the next post.

---

> > > ### Author Response · Authors · 2025-11-19
> > > **Response to Reviewer QXsj (Part 3/5)**
> > >
> > > > **W2: In Section 3.10, the CREI metric locks $\alpha$ at 0.5 without explanation. Giving robustness and efficiency equal weight might not suit every task; an ablation on $\alpha$ or a data-driven reason for this choice would help.**
> > > >
> > >
> > > **R2:** This is a thoughtful suggestion. The choice of $\alpha$ in CREI determines the relative weight between Robustness Ratio (**RR**) and Attack Efficiency Ratio (**AE**), reflecting the trade-off between robustness and efficiency. A smaller $\alpha$ favors AE, emphasizing attack efficiency, while a larger $\alpha$ favors RR, emphasizing robustness. Setting $\alpha = 0.5$ provides a balanced assessment, but CREI remains flexible for **task-specific priorities**. In resource-constrained or edge scenarios where attack efficiency is critical, a smaller $\alpha$ is recommended. Conversely, if the focus is on model accuracy under adversarial attacks, a larger $\alpha$ should be chosen to prioritize robustness.
> > >
> > > To illustrate this effect, a sensitivity analysis was conducted in **Table 6**, varying $\alpha$ from 0 to 1. Results on CIFAR-10 under IPC-{1,10,50} settings show how the emphasis on robustness versus efficiency shifts with $\alpha$, providing guidance for selecting the appropriate weighting. Further details are provided in *Appendix B.8.2*.
> > >
> > > **Table 6: CREI Values of Different Dataset Distillation Methods under Varying $\alpha$ (%).**
> > >
> > > | Method | RR | AE | $\alpha=0.0$ | $\alpha=0.2$ | $\alpha=0.4$ | $\alpha=0.6$ | $\alpha=0.8$ | $\alpha=1.0$ |
> > > |-|-|-|-|-|-|-|-|-|
> > > | DC | 31.87 | 21.53 | 21.53 | 23.60 | 25.67 | 27.73 | 29.80 | 31.87 |
> > > | DSA | 36.53 | 18.97 | 18.97 | 22.48 | 25.99 | 29.51 | 33.02 | 36.53 |
> > > | MTT | 33.30 | 19.21 | 19.21 | 22.03 | 24.85 | 27.66 | 30.48 | 33.30 |
> > > | DM  | 34.50 | 22.13 | 22.13 | 24.60 | 27.08 | 29.55 | 32.03 | 34.50 |
> > > | IDM | 33.03 | 23.89 | 23.89 | 25.72 | 27.55 | 29.37 | 31.20 | 33.03 |
> > > | BACON | 32.87 | 21.53 | 21.53 | 23.80 | 26.07 | 28.33 | 30.60 | 32.87 |
> > >
> > > > **W3: In Section 5.1, the claim that “dataset distillation improves adversarial robustness” is counter-intuitive and lacks mechanistic explanation. The observed CREI drop with increasing IPC is noted but not interpreted. Include a discussion on why distilled datasets may enhance robustness—e.g., whether they filter out non-robust features or reduce overfitting. Analyze the IPC–robustness trade-off more deeply.**
> > > >
> > >
> > > **R3:** This is a valuable suggestion. Dataset distillation can enhance adversarial robustness through several mechanisms. Distilled datasets, especially those generated by gradient-matching methods such as **DSA** and **DC**, focus on **early-stage core features**, filtering out noisy or non-robust features and reducing overfitting. This allows models to learn more **robust representations**, thereby enhancing their resistance to adversarial perturbations. In contrast, distribution-matching methods such as **DM**, **IDM**, and **BACON** capture information from the entire data distribution, which can be advantageous with **high IPC** or **larger datasets** but contributes less to local robustness under targeted attacks.
> > >
> > > Regarding the IPC–robustness trade-off, **increasing IPC** enriches **global features**, improving performance under untargeted attacks but potentially producing more **complex decision boundaries**, reducing robustness under targeted attacks. Gradient-matching methods perform well at low to medium IPCs by emphasizing early-stage features, but this advantage diminishes as IPC or dataset size increases. Distribution-matching methods are better suited to leverage abundant features at high IPCs, explaining their stronger performance in such settings. These patterns hold across CIFAR-10, CIFAR-100, and TinyImageNet.
> > >
> > > These observations provide a **mechanistic explanation** for why DD can enhance robustness and clarify the relationship between IPC, dataset size, and adversarial performance. CREI captures this trade-off by combining robustness and attack efficiency, offering a **unified evaluation** across methods and IPC settings. To further validate this, we conducted experiments on **randomly selected subsets** (**Table 7**), which confirm that DD methods improve adversarial robustness. We also provide deeper analysis of this phenomenon in *Sections 5.1, 5.2, and 5.3*, with additional discussion in *Appendix B.9* on the robustness of non-distilled methods.
> > >
> > > **Table 7: Comparison of Adversarial Robustness Between DD Methods and Random Subsets under Targeted Attacks on CIFAR-10 with IPC-{1,10,50} (%).**
> > >
> > > | Method | RR | AE | CREI |
> > > |-|-|-|-|
> > > | Full-size | 67.24 | 29.55 | 48.39 |
> > > | DC | 88.51 | 27.91 | 58.21 |
> > > | DSA | 86.81 | 27.64 | 57.22 |
> > > | MTT | 83.95 | 28.52 | 56.24 |
> > > | DM  | 85.76 | 26.01 | 55.89 |
> > > | IDM | 87.07 | 23.15 | 55.11 |
> > > | BACON | 84.37 | 29.27 | 56.82 |
> > > | Subset | 54.47 | 32.29 | 43.38 |
> > >
> > > Continued in the next post.

---

> > > > ### Author Response · Authors · 2025-11-19
> > > > **Response to Reviewer QXsj (Part 4/5)**
> > > >
> > > > > **W4: In Section 5.1, the performance differences among DD methods (e.g., why DSA/DM/BACON perform better) are reported but not explained. The analysis remains descriptive. Provide hypotheses or further experiments (e.g., feature analysis, robustness curvature) to explain why certain methods excel. In Figure 4 & Figure 5, the paper claims that “distilled datasets improve adversarial robustness,” a conclusion that runs counter to intuition (smaller datasets are usually expected to yield more fragile models) yet no convincing explanation is provided. Discuss the interaction between dataset scale, distillation method, and adversarial training to provide more actionable insights.**
> > > > >
> > > >
> > > > **R4:** This is an insightful point. As discussed in our response to the previous question (**R3**), we provide an explanation for the performance differences among DD methods (e.g., why DSA, DM, and BACON perform better). To summarize, gradient-matching methods such as DSA and DC focus on **early-stage core features**, filtering out noisy or non-robust features, thereby reducing overfitting and improving adversarial robustness. Distribution-matching methods like DM, IDM, and BACON better leverage the full feature distribution in **high IPC** or **large-scale datasets**, which contributes to their superior performance under certain conditions.
> > > >
> > > > It is important to note that **while smaller datasets generally reduce overall clean accuracy** due to limited information (**Table 9**), this does **not necessarily degrade adversarial robustness**, as evidenced in **Tables 7** and **8**. Distilled datasets, by focusing on core and stable features, allow models to maintain or even improve robustness against attacks, despite reduced dataset size. This mechanism helps reconcile the seemingly counter-intuitive observation that distilled datasets can enhance robustness.
> > > >
> > > > To further illustrate this, we analyzed the **Subset** method across different IPC settings (**Table 8**). Subset performs reasonably well at IPC-1 and IPC-10, where distilled representations still capture key features. At IPC-50, however, Subset’s performance drops significantly, indicating that random subsets fail to fully capture the complex feature distribution needed for robust performance. In contrast, DD methods, particularly distribution-matching ones, exploit richer feature information and maintain higher robustness as IPC increases, explaining their superior performance.
> > > >
> > > > This analysis clarifies the interaction between **dataset scale**, **distillation method**, and **adversarial robustness**, and demonstrates why DD methods can improve robustness even with smaller datasets. These points are further discussed in *Sections 5.1, 5.2, and 5.3 of the manuscript*. Further discussion on the robustness of non-distilled methods is provided in *Appendix B.9*.
> > > >
> > > > **Table 8: Comparison of Adversarial Robustness Between Dataset Distillation Methods and Random Subsets under Untargeted Attacks on CIFAR-10 at IPC-1, IPC-10, and IPC-50 (%).**
> > > >
> > > > | Method |  RR (IPC-1) | AE (IPC-1) | CREI (IPC-1) | RR (IPC-10) | AE (IPC-10) | CREI (IPC-10) |  RR (IPC-50) | AE (IPC-50) | CREI (IPC-50) |
> > > > |-|-|-|-|-|-|-|-|-|-|
> > > > | Full-size| 28.33 | 21.91 | 25.12 | 28.33 | 21.91 | 25.12 | 28.33 | 21.91 | 25.12 |
> > > > | DC | 54.15 | 28.07 | 41.11 | 26.45 | 23.25 | 24.85 | 20.32 | 24.69 | 22.50 |
> > > > | DSA | 46.93 | 26.25 | 36.59 | 24.61 | 23.19 | 23.90 | 31.95 | 22.06 | 27.00 |
> > > > | MTT | 28.78 | 23.71 | 26.25 | 24.94 | 21.86 | 23.40 | 28.37 | 21.75 | 25.06 |
> > > > | DM  | 52.17 | 32.26 | 42.21 | 27.05 | 26.41 | 26.73 | 30.51 | 22.70 | 26.61 |
> > > > | IDM | 31.13 | 29.22 | 30.18 | 24.99 | 26.21 | 25.60 | 32.00 | 27.18 | 29.59 |
> > > > | BACON | 26.34 | 25.24 | 25.79 | 28.24 | 25.94 | 27.09 | 32.87 | 24.10 | 28.48 |
> > > > | Subset | 46.70 | 22.87 | 34.79 | 27.86 | 28.93 | 28.40 | 26.18 | 29.49 | 27.84 |
> > > >
> > > > > **Q1: Why was $\alpha=0.5$ chosen for CREI? Have you experimented with other values, and how sensitive are the rankings to this parameter?**
> > > > >
> > > >
> > > > **R5:** This is a valuable question regarding the choice of $\alpha$ in CREI. As discussed in **R2**, $\alpha$ determines the relative weight between Robustness Ratio (**RR**) and Attack Efficiency Ratio (**AE**), and adjusting $\alpha$ changes the balance between these two aspects. Consequently, the ranking of dataset distillation methods under CREI can shift depending on the chosen $\alpha$. We selected $\alpha= 0.5$ as a balanced setting, but users can adjust $\alpha$ to emphasize either **robustness** or **efficiency** according to the requirements of **specific tasks**. To illustrate this, we conducted a sensitivity analysis in **Table 6**, varying $\alpha$ from 0 to 1. The results show how different values affect method rankings, confirming that CREI provides flexible and interpretable evaluation across diverse scenarios.
> > > >
> > > > Continued in the next post.

---

> > > > > ### Author Response · Authors · 2025-11-19
> > > > > **Response to Reviewer QXsj (Part 5/5)**
> > > > >
> > > > > > **Q2: Can you provide a deeper explanation for why some DD methods (e.g., DSA, DM, BACON) exhibit stronger adversarial robustness? Is it related to their distillation objectives or synthetic data diversity?**
> > > > > >
> > > > >
> > > > > **R6:** This is an insightful question regarding the mechanisms underlying the stronger adversarial robustness of certain DD methods. As discussed in **R3** and **R4**, gradient-matching methods such as **DSA** and **DC** focus on **early-stage core features**, filtering out noisy or non-robust signals, which reduces overfitting and enhances robustness. In contrast, distribution-matching methods like **DM**, **IDM**, and **BACON** leverage the full feature distribution, enabling them to maintain high robustness at **larger IPCs** or **dataset scales**. These patterns are further confirmed by comparisons with **random subsets**, which fail to capture comprehensive feature diversity and exhibit reduced robustness at higher IPCs (see **Tables 7 and 8**). The observed differences in robustness are closely tied to each method’s distillation objectives and its ability to capture informative and diverse synthetic features. Additional analysis is provided in *Sections 5.1, 5.2, and 5.3 of the manuscript*. Further discussion on the robustness of non-distilled methods is provided in *Appendix B.9*.
> > > > >
> > > > > > **Q3: The conclusion that “distillation improves robustness” contradicts the common belief that smaller datasets lead to weaker models. Can you discuss potential reasons for this phenomenon?**
> > > > > >
> > > > >
> > > > > **R7:** This is a thoughtful question. While **smaller datasets** are generally believed to produce **weaker models**, this primarily refers to **clean accuracy** rather than **adversarial robustness**. As shown in **Table 9**, model performance decreases with lower IPC, confirming that small datasets yield **lower clean accuracy**. However, adversarial robustness does not strictly follow this trend. **Tables 7 and 8** show that lower IPC values, corresponding to smaller distilled datasets, can sometimes achieve relatively **higher robustness** under both targeted and untargeted attacks.
> > > > >
> > > > > This phenomenon can be explained by the mechanisms of dataset distillation. Gradient-matching methods such as **DSA** and **DC** focus on **early-stage core features**, filtering out noisy or non-robust features and reducing overfitting, which allows models to learn more robust representations. Distribution-matching methods such as **DM**, **IDM**, and **BACON** leverage the full feature distribution, which benefits robustness at **higher IPCs** or l**arger datasets**. These mechanisms account for the observed differences in adversarial performance across methods and IPC settings, as further discussed in *Sections 5.1, 5.2, and 5.3 of the manuscript.*
> > > > >
> > > > > In addition, as shown in *Appendix B.6*, there exists a **trade-off** between **adversarial robustness** and **clean accuracy**. Dataset distillation extracts key features, which **enhances robustness** but may slightly **reduce clean accuracy**. This explains why models trained on smaller distilled datasets (low IPC) can exhibit relatively high adversarial robustness despite lower accuracy. Comparisons with random subsets, as shown in **Tables 7 and 8**, indicate that random subsets perform reasonably at low IPC but fail to maintain robustness at higher IPCs, highlighting the advantage of DD methods in **capturing richer feature distributions**.
> > > > >
> > > > > These results clarify that clean accuracy and adversarial robustness are not always positively correlated. The interaction among dataset scale, distillation method, and feature selection determines both robustness and performance, providing a mechanistic explanation for why distilled datasets can improve robustness even when dataset size is small. **Tables 7, 8, and 9** provide empirical support for these observations, making the explanation concrete and verifiable.
> > > > >
> > > > > **Table 9: Clean Accuracy of DD Methods, Full-Size, and Random Subsets on CIFAR-10 Across Different IPC Settings (%).**
> > > > >
> > > > > | Method    | IPC-1 | IPC-10 | IPC-50 |
> > > > > | --------- | ----- | ------ | ------ |
> > > > > | Full-size | 87.81 | 87.81  | 87.81  |
> > > > > | DC        | 28.11 | 45.31  | 54.31  |
> > > > > | DSA       | 28.23 | 51.06  | 60.48  |
> > > > > | MTT       | 43.44 | 63.76  | 69.99  |
> > > > > | DM        | 25.34 | 48.45  | 62.06  |
> > > > > | IDM       | 46.01 | 58.47  | 67.07  |
> > > > > | BACON     | 44.95 | 61.73  | 69.95  |
> > > > > | Subset    | 14.29 | 30.14  | 50.30  |
> > > > >
> > > > > We hope that our responses can satisfactorily address your concerns. Thank you again for your time to provide us with valuable feedback.

---

### Official Review · Reviewer_hV3a · 2025-11-01

**Soundness:** 2
**Presentation:** 3
**Contribution:** 2
**Rating:** 4
**Confidence:** 4

**Summary:**

This paper introduces an open and unified benchmark designed to systematically evaluate the adversarial robustness of models trained via dataset distillation  methods called BEARD. The benchmark covers multiple DD algorithms, adversarial attacks  and widely-used image datasets. The authors formalize an adversarial game framework and employ three key evaluation metrics, Robustness Ratio,  Attack Efficiency Ratio and Comprehensive Robustness-Efficiency Index respectively.

**Strengths:**

1. The code, leaderboard, and data pools are open-sourced, which can help facilitate future research.

2. The adversarial game formalism is thoughtfully articulated.

**Weaknesses:**

1. The empirical results do not directly benchmark against some newer strategies for adversarial training (e.g, [1]), adversarial attacks  (transformation-based attacks [2] and generative approachs [3] )and other widely used datasets (e.g., cinic10, imagenet and mnist)

2. Section 5 reports trends but lacks deeper causal explanations (e.g., why DM improves CREI).

3. Section 3 introduces too many mathematical definitions, but provides limited experimental interpretation or discussion later.

4. typo and errors in grammar.  1) "DISCUSSION THE DIFFERENCES BETWEEN BEARD AND OTHER BENCHMARKS" (B.5) -> "THE DIFFERENCES BETWEEN BEARD AND OTHER BENCHMARKS" in  2) Missing space between “IDM” and “BACON” in figure 3. 3) TinyImageNet” or “Tiny-ImageNet? should be consistent. 4)

5. From the current description, BEARD appears conceptually similarly  to  DD-RobustBench in both purpose and experimental scope, though the authors claim they provide a more holistic assessment.  But RRM does not provide substantial novelty beyond existing robustness evaluation metrics.  And it is easy to integrate target settings in DD-RobustBench. Furthermore, the DD methods, attack methods and  provided in BEARD are also limited.  The paper reads more like an engineering consolidation than a fundamentally new contribution.  I am not sure I understand it correctly.

[1] Yang, Zhuolin, et al. "Trs: Transferability reduced ensemble via promoting gradient diversity and model smoothness." Advances in Neural Information Processing Systems 34 (2021): 17642-17655.

[2] Yun, Zebin, et al. "The Ultimate Combo: Boosting Adversarial Example Transferability by Composing Data Augmentations." Proceedings of the 2024 Workshop on Artificial Intelligence and Security. 2024.

[3] Wei, Zhipeng, et al. "Enhancing the self-universality for transferable targeted attacks." Proceedings of the IEEE/CVF conference on computer vision and pattern recognition. 2023.

**Questions:**

What is the meaning of $|\epsilon|$? Why the authors use $\epsilon= 8/255$ and $|\epsilon|= 8/255$ interchangeably?

---

> ### Author Response · Authors · 2025-11-19
> **Response to Reviewer hV3a (Part 1/6)**
>
> We sincerely thank Reviewer `hV3a` for the constructive feedback. The reviewer highlights that the **code**, **leaderboard**, and **data pools** are **open-sourced**, facilitating future research, and that the **adversarial game formalism** is **thoughtfully** articulated. The reviewer also raised concerns on limited attack diversity and dataset scale (**W1**), insufficient causal explanations for robustness trends (**W2**), unclear connection between mathematical definitions and experiments (**W3**), typos and formatting issues (**W4**), limited novelty compared to DD-RobustBench and restricted coverage of methods/attacks (**W5**), and inconsistent notation for perturbation magnitude $\epsilon$ (**Q1**). All these points have been carefully addressed in the revision. Point-by-point responses (**R**) to Reviewer `hV3a`’s questions (**Q**) and weaknesses (**W**) follow.
>
> **Summary of Revisions:**
>
> - Extended the evaluation to address scalability and coverage concerns (**W1**) by including **additional attacks**, **datasets**, and **adversarial training strategies** (*Appendix A.1.1*, *Appendix B.7.1–B.7.3*).
>
> - Provided **deeper causal analysis** of robustness trends (**W2**), explaining how DD methods, IPC, dataset scale, and attack types impact CREI (*Sections 5.1-5.3*).
>
> - Clarified the connection between formal definitions and experiments (**W3**), detailing how RR, AE, and CREI metrics are computed and interpreted (*Remarks 3.6 and 3.11*).
>
> - Corrected typos and formatting issues (**W4**) and standardized the notation for perturbation magnitude $\epsilon$ (**Q1**).
>
> - Expanded discussion on BEARD’s novelty compared to DD-RobustBench and its method/attack coverage (**W5**), highlighting the unified evaluation framework, modular design, and aggregated robustness metrics (*Appendix B.5*).
>
> Continued in the next post.

---

> > ### Author Response · Authors · 2025-11-19
> > **Response to Reviewer hV3a (Part 2/6)**
> >
> > > **W1: The empirical results do not directly benchmark newer adversarial training strategies, transformation-based attacks, generative attacks, or larger datasets (e.g., CINIC-10, ImageNet, MNIST).**
> > >
> > **R1:** This is a valuable comment. BEARD provides a **unified evaluation perspective** via an **adversarial game formulation**. With the CREI metric, it enables principled comparisons across IPC settings and multiple attackers, which existing benchmarks lack. Its **modular architecture** supports **flexible evaluation** of DD methods, attacks, and datasets.
> >
> > To further validate the framework’s effectiveness and coverage, additional experiments have been conducted:
> >
> > **R1.1: Extended Attack Spectrum and Robustness Evaluation**
> >
> > We extended the **attack spectrum** to include gradient-based transferable and ensemble attacks including **DIFGSM**, **MIFGSM**, **TIFGSM**, **EOTPGD**, and **UPGD**, covering diverse threat models. DIFGSM improves transferability via input transformations, MIFGSM leverages momentum to escape local minima, TIFGSM maintains invariance to small translations, EOTPGD averages gradients over stochastic transformations to test robustness against randomized defenses, and UPGD combines different losses, step sizes, and momentum in an iterative PGD framework. This expanded set allows BEARD to evaluate dataset distillation methods under more challenging scenarios.
> >
> > **Table 1** shows the **raw results** of these attacks, but they do **not fully reflect overall robustness**. To address this, we introduce the proposed **robustness metric**, which aggregates previous attacks including FGSM, PGD, DeepFool, CW, and AutoAttack with the new ones. **Table 2** reports the results, where under targeted attacks DM achieves higher CREI than DSA, likely because distribution-based methods like DM learn smoother boundaries while gradient-based DSA captures more detailed distributional information but exhibits sharper boundaries. Additional analyses on attack types, dataset complexity, and distillation strategies are provided in *Appendix B.7.1*.
> >
> > **R1.2: Evaluation on Additional Datasets**
> >
> > To further demonstrate BEARD’s applicability, we extended the evaluation to **additional datasets** supported by DC, DSA, and BACON, including **MNIST**, **FashionMNIST**, and **SVHN**. The results in **Table 3** confirm the benchmark’s broad coverage. On MNIST, DSA achieves the highest CREI, reflecting its strong ability to capture information from the training distribution on simpler datasets. On FashionMNIST, which contains more complex textures, DSA still outperforms BACON and DC, though the gap narrows. On SVHN, a more complex natural image dataset, DM and DC achieve competitive CREI, highlighting that methods emphasizing distribution smoothness or robustness can be advantageous as dataset complexity increases. Additional analysis and insights regarding dataset complexity and method characteristics are provided in *Appendix B.7.2*.
> >
> > **Table 1: Comparison of Dataset Distillation Methods on CIFAR-10 IPC-50 under Targeted Adversarial Attacks (%).**
> > | Attack | Full-size | DC | DSA | MTT | DM  | IDM  | BACON |
> > |--------|-------|-------|-------|-------|-------|-------|-------|
> > | Clean  | 87.41 | 54.31 | 60.48 | 69.66 | 62.06 | 66.91 | 69.95 |
> > | DIFGSM | 41.11 | 30.02 | 39.71 | 41.20 | 39.21 | 43.56 | 45.92 |
> > | MIFGSM | 38.87 | 29.38 | 39.22 | 39.93 | 38.71 | 43.25 | 45.77 |
> > | TIFGSM | 52.78 | 30.62 | 42.40 | 45.31 | 41.66 | 45.48 | 48.38 |
> > | EOTPGD | 37.78 | 29.58 | 39.20 | 40.00 | 38.59 | 43.23 | 45.57 |
> > | UPGD   | 38.87 | 29.38 | 39.22 | 39.93 | 38.71 | 43.25 | 45.77 |
> >
> > **Table 2: Comparison of Dataset Distillation Methods on CIFAR-10 IPC-{1,10,50} under Combined Targeted Adversarial Attacks (%).**
> >
> > | Method | RR | AE | CREI |
> > |------|----|---|---|
> > | Full-size | 22.79 | 16.92 | 19.85 |
> > | DC | 23.86 | 20.35 | 22.10 |
> > | DSA | 42.61 | 18.48 | 30.55 |
> > | MTT | 32.02 | 19.12 | 25.57 |
> > | DM  | 43.10 | 19.14 | 31.12 |
> > | IDM | 31.15 | 19.68 | 25.42 |
> > | BACON | 34.51 | 18.93 | 26.72 |
> >
> > **Table 3: Comparison of Dataset Distillation Methods on MNIST, FashionMNIST, and SVHN.**
> >
> > **Table 3(a): MNIST.**
> >
> > | Metrics | DC | DSA | BACON |
> > |----|---|---|---|
> > | RR | 82.26 | 87.59 | 71.55 |
> > | AE | 19.90 | 19.54 | 23.51 |
> > | CREI | 51.08 | 53.56 | 47.53 |
> >
> > **Table 3(b): FashionMNIST.**
> >
> > | Metrics | DC | DSA | BACON |
> > |----|---|---|---|
> > | RR | 61.60 | 69.18 | 43.71 |
> > | AE | 18.76 | 18.88 | 19.66 |
> > | CREI | 40.18 | 44.03 | 31.69 |
> >
> > **Table 3(c): SVHN.**
> >
> > | Metrics | DC | DSA | BACON |
> > |---|---|---|---|
> > | RR | 52.66 | 46.81 | 30.70 |
> > | AE | 19.46 | 20.20 | 19.43 |
> > | CREI | 36.06 | 33.50 | 25.06 |
> >
> > Continued in the next post.

---

> > > ### Author Response · Authors · 2025-11-19
> > > **Response to Reviewer hV3a (Part 3/6)**
> > >
> > > **R1.3: Extended Adversarial Training Strategies and Robustness Evaluation**
> > >
> > > We evaluated the robustness of dataset distillation methods on CIFAR-10 (IPC-50) under **multiple adversarial training strategies**, including **TRS [2]**, as highlighted by the reviewer, as well as **PGD_AT [1]**, **TRADES [3]**, and **MART [4]**. Robustness metrics (**RR**, **AE**, **CREI**) were measured under untargeted attacks, with results shown in **Table 4**.
> > >
> > > PGD_AT(Src) generates adversarial examples directly from the inputs without modifying the loss, providing strong guidance for the model to learn robust representations, which explains its consistently largest gains. PGD_AT(Loss) aligns predictions of clean and adversarial examples with the labels, producing moderate improvements. TRADES, MART, and TRS offer alternative trade-offs between robustness and attack efficiency through cross-entropy/KL balancing, emphasis on misclassified examples, or gradient diversity and smoothness.
> > >
> > > All strategies improve robustness compared to baseline DD methods. BACON shows higher baseline robustness than DM and IDM due to richer feature representations, and PGD_AT(Src) reduces the performance gap across methods. These results confirm that DD methods can effectively leverage various adversarial training strategies, with PGD_AT(Src) providing the strongest enhancement, while TRS and other strategies offer controlled or diversity-based improvements. Further analyses of adversarial training strategies are provided in *Appendix B.7.3*.
> > >
> > > **Table 4: Robustness Metrics of DD Methods under Different Adversarial Trainings (%).**
> > >
> > > | DD + AT | RR | AE | CREI |
> > > |---|---|---|---|
> > > | DM | 29.51  | 30.14  | 29.83  |
> > > | DM + PGD_AT(Src) | 47.85  | 25.57  | 36.71  |
> > > | DM + TRADES | 30.35  | 31.37  | 30.86  |
> > > | DM + TRS    | 30.22  | 32.44  | 31.33  |
> > > | DM + MART   | 29.73  | 30.01  | 29.87  |
> > > | DM + PGD_AT(Loss)      | 30.09  | 29.96  | 30.03  |
> > > | IDM         | 31.51  | 27.77  | 29.64  |
> > > | IDM + PGD_AT(Src)        | 46.92  | 25.26  | 36.09  |
> > > | IDM + TRADES | 31.80  | 29.56  | 30.68  |
> > > | IDM + TRS     | 31.03  | 29.54  | 30.29  |
> > > | IDM + MART      | 31.10  | 27.63  | 29.37  |
> > > | IDM + PGD_AT(Loss)     | 30.43  | 27.85  | 29.14  |
> > > | BACON           | 36.43  | 28.03  | 32.23  |
> > > | BACON + PGD_AT(Src)      | 48.19  | 22.75  | 35.47  |
> > > | BACON + TRADES  | 32.39  | 32.68  | 32.53  |
> > > | BACON + TRS     | 32.79  | 32.47  | 32.63  |
> > > | BACON + MART    | 32.69  | 33.26  | 32.97  |
> > > | BACON + PGD_AT(Loss)   | 32.43  | 31.36  | 31.89  |
> > >
> > > **References:**
> > >
> > > [1] Madry, Aleksander, Aleksandar Makelov, Ludwig Schmidt, Dimitris Tsipras, and Adrian Vladu. Towards deep learning models resistant to adversarial attacks. In ICLR, 2018.
> > >
> > > [2] Yang, Zhuolin, Linyi Li, Xiaojun Xu, Shiliang Zuo, Qian Chen, Pan Zhou, Benjamin Rubinstein, Ce Zhang, and Bo Li. Trs: Transferability reduced ensemble via promoting gradient diversity and model smoothness. In NeurIPS, 2021.
> > >
> > > [3] Zhang, Hongyang, Yaodong Yu, Jiantao Jiao, Eric Xing, Laurent El Ghaoui, and Michael Jordan. Theoretically principled trade-off between robustness and accuracy. In ICML, 2019.
> > >
> > > [4] Wang, Yisen, Difan Zou, Jinfeng Yi, James Bailey, Xingjun Ma, and Quanquan Gu. Improving adversarial robustness requires revisiting misclassified examples. In ICLR, 2019.
> > >
> > > Continued in the next post.

---

> > > > ### Author Response · Authors · 2025-11-19
> > > > **Response to Reviewer hV3a (Part 4/6)**
> > > >
> > > > > **W2: Section 5 reports trends but lacks deeper causal explanations (e.g., why DM improves CREI).**
> > > > >
> > > > **R2:** This is a constructive comment. We consider it important to provide a deeper causal analysis on why different DD methods influence adversarial robustness. Accordingly, we have added detailed discussion and analysis in *Sections 5.1, 5.2, and 5.3* of the revised manuscript, with all changes highlighted in blue.
> > > >
> > > > In *Section 5.1 (Figure 3)*, we analyze how **different DD methods** influence **adversarial robustness**. Gradient-matching methods like **DSA** and **DC** capture **early-stage features** during dataset distillation, which contain rich information that facilitates learning **robust features [1]** and thus leads to **higher CREI**. Trajectory-matching methods like **MTT** also leverage **full-trajectory information** and can learn robust features, but as shown in [2], **early-stage features** contribute more significantly to **adversarial robustness**. Therefore, gradient-matching methods achieve better robustness than trajectory-matching methods by focusing on these early-stage features. Targeted attacks primarily evaluate **local robustness**, benefiting from **smoother decision boundaries** learned from **larger distilled datasets**, whereas untargeted attacks reflect **global robustness**, which may decline as dataset complexity increases.
> > > >
> > > > In *Section 5.2 (Figure 4)*, we discuss robustness under **varying IPCs**. Gradient-matching methods maintain strong performance at low to medium IPCs. Distribution-matching methods such as **DM**, **IDM**, and **BACON** can better exploit the richer feature distributions in high-IPC or large-dataset settings, which explains their comparatively higher adversarial performance in these specific scenarios, while they do not outperform gradient-matching methods at lower IPCs.
> > > >
> > > > *Section 5.3 (Figure 5)* examines the **effect of adversarial training**, which smooths decision boundaries and complements the information captured by distilled datasets, explaining consistent CREI improvements across methods and dataset scales.
> > > >
> > > > Building on the analyses in *Sections 5.1–5.3*, different DD methods exhibit distinct robustness patterns. Under standard settings, DM does not achieve higher CREI. To further evaluate DM and other distribution-matching methods under **more challenging transferable attacks** (DIFGSM, MIFGSM, TIFGSM, EOTPGD, UPGD), we include additional experiments in *Appendix B.7.1* (as shown in **Table 2**). These experiments evaluate the performance of DM and other DD methods under transferable attacks. Additionally, we conduct further experiments comparing non-distilled methods, including full-size and random subsets, as shown in **Tables 5 and 6**. These experiments confirm that dataset distillation methods, including DM, are **more robust than non-distilled methods**, demonstrating superior adversarial resilience. These results complement the causal explanations provided in *Sections 5.1–5.3*.
> > > >
> > > > **Table 5: Comparison of Adversarial Robustness Between DD Methods and Random Subsets under Targeted Attacks on CIFAR-10 with IPC-{1,10,50} (%).**
> > > >
> > > > | Method | RR | AE | CREI |
> > > > |---|---|---|---|
> > > > | Full-size | 67.24 | 29.55 | 48.39 |
> > > > | DC | 88.51 | 27.91 | 58.21 |
> > > > | DSA | 86.81 | 27.64 | 57.22 |
> > > > | MTT | 83.95 | 28.52 | 56.24 |
> > > > | DM | 85.76 | 26.01 | 55.89  |
> > > > | IDM | 87.07 | 23.15 | 55.11 |
> > > > | BACON | 84.37 | 29.27 | 56.82 |
> > > > | Subset | 54.47 | 32.29 | 43.38 |
> > > >
> > > > **Table 6: Comparison of Adversarial Robustness Between Dataset Distillation Methods and Random Subsets under Untargeted Attacks on CIFAR-10 at IPC-1, IPC-10, and IPC-50 (%).**
> > > >
> > > > | Method |  RR (IPC-1) | AE (IPC-1) | CREI (IPC-1)  |  RR (IPC-10)  | AE (IPC-10) | CREI (IPC-10) |  RR (IPC-50)  |  AE (IPC-50)  | CREI (IPC-50) |
> > > > |---|---|---|---|---|---|---|---|---|---|
> > > > | Full-size| 28.33 | 21.91 | 25.12 | 28.33 | 21.91 | 25.12 | 28.33 | 21.91 | 25.12 |
> > > > | DC | 54.15 | 28.07 | 41.11 | 26.45 | 23.25 | 24.85 | 20.32 | 24.69 | 22.50 |
> > > > | DSA | 46.93 | 26.25 | 36.59 | 24.61 | 23.19 | 23.90 | 31.95 | 22.06 | 27.00 |
> > > > | MTT | 28.78 | 23.71 | 26.25 | 24.94 | 21.86 | 23.40 | 28.37 | 21.75 | 25.06 |
> > > > | DM  | 52.17 | 32.26 | 42.21 | 27.05 | 26.41 | 26.73 | 30.51 | 22.70 | 26.61 |
> > > > | IDM | 31.13 | 29.22 | 30.18 | 24.99 | 26.21 | 25.60 | 32.00 | 27.18 | 29.59 |
> > > > | BACON | 26.34 | 25.24      | 25.79      | 28.24      | 25.94      | 27.09       | 32.87      | 24.10      | 28.48       |
> > > > | Subset   | 46.70      | 22.87      | 34.79      | 27.86      | 28.93      | 28.40       | 26.18      | 29.49      | 27.84       |
> > > >
> > > > **References:**
> > > >
> > > > [1] Ilyas, Andrew, Shibani Santurkar, Dimitris Tsipras, Logan Engstrom, Brandon Tran, and Aleksander Madry. Adversarial examples are not bugs, they are features. In NeurIPS, 2019.
> > > >
> > > > [2] Zhang, Jingfeng, Xilie Xu, Bo Han, Gang Niu, Lizhen Cui, Masashi Sugiyama, and Mohan Kankanhalli. Attacks which do not kill training make adversarial learning stronger. In ICML, 2020.
> > > >
> > > > Continued in the next post.

---

> > > > > ### Author Response · Authors · 2025-11-19
> > > > > **Response to Reviewer hV3a (Part 5/6)**
> > > > >
> > > > > > **W3: Section 3 introduces too many mathematical definitions, but provides limited experimental interpretation or discussion later.**
> > > > > >
> > > > >
> > > > > **R3:** This is a helpful suggestion regarding the extensive mathematical definitions in *Section 3* that lacked experimental interpretation. The definitions, such as **defender $\mathcal{D}$** and **attacker $\mathcal{A}$**, formalize the **adversarial game framework** and directly correspond to **real-world concepts**, with **dataset distillation** as the **defender** and **adversarial attacks** as the **attacker**. This framework underpins all proposed metrics (**RRM**, **AEM**, **CREI**), ensuring that the evaluations in *Section 5* systematically reflect these definitions. In the revised manuscript, *Section 3* clarifies that these metrics (RR, AE, and CREI) are directly computed and analyzed, while *Section 5* provides more in-depth discussion linking the formal definitions to observed experimental trends.
> > > > >
> > > > > > **W4: typo and errors in grammar. 1) "DISCUSSION THE DIFFERENCES BETWEEN BEARD AND OTHER BENCHMARKS" (B.5) -> "THE DIFFERENCES BETWEEN BEARD AND OTHER BENCHMARKS" in 2) Missing space between “IDM” and “BACON” in figure 3. 3) TinyImageNet” or “Tiny-ImageNet? should be consistent. 4).**
> > > > > >
> > > > >
> > > > > **R4:** This is a thoughtful suggestion regarding typos and formatting issues. In the revised manuscript, we have made the following corrections:
> > > > >
> > > > > - The title of Appendix B.5 has been updated to ``The Differences between BEARD and Other Benchmarks''.
> > > > > - In Figures 3 and 5, we adjusted the x-axis labels to display diagonally, creating sufficient space between IDM and BACON.
> > > > > - All references to TinyImageNet have been standardized for consistency throughout the manuscript.
> > > > >
> > > > > Continued in the next post.

---

> > > > > > ### Author Response · Authors · 2025-11-19
> > > > > > **Response to Reviewer hV3a (Part 6/6)**
> > > > > >
> > > > > > > **W5: From the current description, BEARD appears conceptually similarly to DD-RobustBench in both purpose and experimental scope, though the authors claim they provide a more holistic assessment. But RRM does not provide substantial novelty beyond existing robustness evaluation metrics. And it is easy to integrate target settings in DD-RobustBench. Furthermore, the DD methods, attack methods and provided in BEARD are also limited. The paper reads more like an engineering consolidation than a fundamentally new contribution. I am not sure I understand it correctly.**
> > > > > > >
> > > > > >
> > > > > > **R5:** This is a valuable comment regarding the novelty of BEARD. BEARD is a **fundamentally new benchmark** for evaluating the adversarial robustness of dataset distillation methods, distinct from DD-RobustBench[1] in both **conceptual framework** and **engineering design**. Conceptually, BEARD introduces a **unified evaluation perspective** across multiple IPCs and attacks, leveraging an **adversarial game framework** to provide principled theoretical justification for robustness assessment. From an engineering perspective, BEARD employs a **distinct modular software architecture** with **flexible model** and **dataset pools**, enabling straightforward integration of custom DD methods, attacks, and datasets. These design choices allow for a systematic, interpretable, and comprehensive evaluation of DD methods that goes beyond mere engineering consolidation. Additional discussion and comparisons are provided in *Appendix B.5*.
> > > > > >
> > > > > > **Conceptual Innovations:** BEARD is the **first** benchmark to evaluate DD methods from **a unified perspective**, integrating multiple IPC settings and diverse adversarial attacks under the adversarial game framework. This framework enables the introduction of **Robustness Ratio** (RR, RRM) and **Attack Efficiency Ratio** (AE, AEM) metrics, providing a principled way to rank DD methods by overall robustness while accounting for edge-case adversarial scenarios. In contrast, DD-RobustBench evaluates each DD method at each IPC under individual attacks, reporting only per-attack accuracy (**Table 7**), which lacks a holistic view of overall adversarial robustness. BEARD’s unified metrics allow both detailed per-attack evaluation and aggregation across multiple attacks and IPCs, producing clear rankings of DD methods (**Table 8**) and offering a more complete, interpretable, and actionable assessment compared to previous benchmarks.
> > > > > >
> > > > > > **Engineering Innovations:** BEARD adopts a **modular software design**, with **model pools** and **dataset pools** to flexibly manage evaluations. This architecture allows straightforward integration of new DD methods, attacks, and datasets, supporting both standardized benchmarking and customized experiments. These engineering choices operationalize the conceptual framework, enabling a unified evaluation that addresses the complex interactions among distillation methods, model architectures, and attack strategies.
> > > > > >
> > > > > > By integrating conceptual novelty with a flexible modular design, BEARD enables **systematic, interpretable, and comprehensive evaluation**, supporting per-attack analyses and aggregated robustness rankings that were previously inaccessible.
> > > > > >
> > > > > >
> > > > > > **Table 7: Untargeted Attack Evaluation of DD Methods on CIFAR-10 (IPC-10) via DD-RobustBench and BEARD (%).**
> > > > > >
> > > > > > | Benchmark | Attack | DC | DSA | MTT | DM | IDM |
> > > > > > |-|-|-|-|-|-|-|
> > > > > > | DD-RobustBench | RRM | N/A | N/A | N/A | N/A | N/A |
> > > > > > || None (Clean)| 46.07 | 52.93 | 60.98 | 49.81 | 58.84 |
> > > > > > || FGSM | 24.4 | 28.12 | 30.99 | 22.96 | 25.54 |
> > > > > > || PGD | 18.27 | 20.01 | 21.95 | 14.74 | 23.34 |
> > > > > > || C&W | 16.5 | 19.83 | 19 | 14.95 | 23.17 |
> > > > > > || AutoAttack | 16 | 19.11 | 17.95 | 14.18 | 22.63 |
> > > > > > | BEARD | RRM | 9.29 | 7.86 | 10.51 | 7.09 | 2.87 |
> > > > > > || None (Clean)| 45.31 | 51.06 | 63.76 | 48.45 | 58.47 |
> > > > > > || FGSM | 25.48 | 28.21 | 35.34 | 26.38 | 32.31 |
> > > > > > || PGD | 20.83 | 21.98 | 25.96 | 20.98 | 26.34 |
> > > > > > || C&W | 15.88 | 13.21 | 15.71 | 11.05 | 14.49 |
> > > > > > || AutoAttack | 17.75 | 12.82 | 23.13 | 18.29 | 23.61 |
> > > > > >
> > > > > > **Table 8: Ranking of DD Methods Based on Multi-Adversary Robustness Ratio (RRM, %).**
> > > > > >
> > > > > > | Method | RRM | Ranking |
> > > > > > |-|-|-|
> > > > > > | DC | 9.29 | 2 |
> > > > > > | DSA | 7.86 | 3 |
> > > > > > | MTT | 10.51 | 1 |
> > > > > > | DM  | 7.09 | 4 |
> > > > > > | IDM | 2.87 | 5 |
> > > > > >
> > > > > > **Reference:**
> > > > > >
> > > > > > [1] Wu, Yifan, Jiawei Du, Ping Liu, Yuewei Lin, Wei Xu, and Wenqing Cheng. Dd-robustbench: An adversarial robustness benchmark for dataset distillation. In IEEE TIP, 2025.
> > > > > >
> > > > > > > **Q1: What is the meaning of $|\epsilon|$? Why the authors use $\epsilon=8/255$ and $|\epsilon|=8/255$ interchangeably?**
> > > > > > >
> > > > > >
> > > > > > **R6:** This is a helpful comment. In BEARD, $\epsilon$ denotes the non-negative maximum perturbation magnitude (e.g., $\epsilon = 8/255$). All instances have been standardized to $\epsilon$, with its definition clarified in the manuscript.
> > > > > >
> > > > > > We hope that our responses can satisfactorily address your concerns. Thank you again for your time to provide us with valuable feedback.

---

> ### Author Response · Authors · 2025-11-26
> **Response to Reviewer hV3a (Supplementary ImageNet-1K Experiments)**
>
> > **W1: The empirical results do not directly benchmark newer adversarial training strategies, transformation-based attacks, generative attacks, or larger datasets (e.g., CINIC-10, ImageNet, MNIST).**
> >
>
> **R1.2: Evaluation on Additional Datasets (Supplementary ImageNet-1K Experiments)**
>
> We thank the reviewer `hV3a` for the constructive comment on benchmarking larger datasets. To further validate BEARD’s applicability to large-scale datasets and complement our existing evaluations on MNIST, FashionMNIST, and SVHN, we conducted **preliminary experiments** on **full-size ImageNet-1K models** using BEARD. In addition, we reproduced **SRe2L $\dagger$** under the IPC-50 setting during the rebuttal period, as publicly released large-scale distilled datasets are **not yet available**. We evaluated FGSM, PGD, PGD_L2, DIFGSM, MIFGSM, TIFGSM, EOTPGD, and UPGD, as previously used attacks such as DeepFool, C&W, and AutoAttack are computationally intensive and less practical for large-scale models.
>
> **Tables 1 and 2** summarize the results, including robustness metrics (RR, AE, CREI) and accuracy under various adversarial attacks. These results demonstrate that **BEARD can extend coverage to large-scale datasets, providing a representative assessment of model robustness across simple to complex datasets.** Once distilled ImageNet-1K datasets become publicly available, they will be incorporated into BEARD using the same evaluation pipeline.
>
> **Table 1: Robustness Metrics of ImageNet-1K Trained Models Under Adversarial Attacks Evaluated by BEARD (%).**
>
> | Method    | Attack Type |   RR  |   AE   |  CREI  |
> |-----------|-------------|-------|--------|--------|
> | Full-size | Targeted    | 9.89  | 51.10  | 30.49  |
> | SRe2L $\dagger$     | Targeted    | 4.39  | 54.08  | 29.23  |
> | Full-size | Untargeted  | 7.42  | 51.10  | 29.26  |
> | SRe2L $\dagger$     | Untargeted  | 4.14  | 52.58  | 28.36  |
>
> $\dagger$ Results reproduced under the IPC-50 ImageNet-1K setting during the rebuttal period.
>
> **Table 2: Accuracy of ImageNet-1K Trained Models Under Adversarial Attacks Evaluated by BEARD (%).**
>
> | Attack   | Full-size (Targeted) | SRe2L $\dagger$ (Targeted) | Full-size (Untargeted) | SRe2L $\dagger$ (Untargeted) |
> |----------|------------------------|----------------------|--------------------------|----------------------|
> | Clean    | 69.67                 | 46.68               | 69.67                   | 46.68               |
> | FGSM     | 16.22                 | 6.44                | 13.56                   | 4.21                |
> | PGD      | 4.57                  | 7.10                | 0.20                    | 0.28                |
> | PGD_L2   | 24.14                 | 9.44                | 24.14                   | 9.44                |
> | DIFGSM   | 7.78                  | 8.97                | 0.50                    | 0.48                |
> | MIFGSM   | 5.44                  | 7.78                | 0.24                    | 0.32                |
> | TIFGSM   | 19.66                 | 10.81               | 3.73                    | 2.26                |
> | EOTPGD   | 4.84                  | 7.32                | 0.20                    | 0.29                |
> | UPGD     | 5.44                  | 7.78                | 0.24                    | 0.32                |
>
> $\dagger$ Results reproduced under the IPC-50 ImageNet-1K setting during the rebuttal period.
>
> We hope that our responses and the preliminary ImageNet-1K results help demonstrate BEARD’s coverage across both simple and large-scale datasets and address your concerns regarding large-scale evaluation. We are always willing to discuss and address any further questions.

---

### Author Response · Authors · 2025-12-01
**General Response**

We sincerely thank the reviewers and the Area Chair for their time, effort, and valuable feedback. Their insights have greatly helped us clarify definitions, improve readability, and strengthen the benchmark evaluation. **Below, we first provide a general overview of BEARD and its key contributions, followed by a detailed summary of the revisions made in response to reviewer comments.**

**General Response:**

BEARD is an **open** and **unified benchmark** designed to systematically evaluate the adversarial robustness of models trained on distilled datasets produced by state-of-the-art dataset distillation methods. It integrates **diverse attacks and standard datasets**, and introduces **three novel metrics (RR, AE, CREI)** to enable a comprehensive and reproducible assessment of robustness across varied settings. To facilitate community adoption and ensure **transparency** and **reproducibility**, BEARD provides **public code, dataset/model pools, and a leaderboard**, supporting accessibility for researchers and promoting benchmarking across the community.

We appreciate the reviewers’ recognition of the following key strengths:

- **[Theoretically Sound & Valuable Benchmark]:** BEARD is the first unified benchmark for evaluating adversarial robustness in dataset distillation, integrating multiple DD methods, datasets, and diverse adversarial attacks. The adversarial game formalism is clearly articulated and conceptually sound. (`hV3a`, `QXsj`, `3awv`, `b7pd`)

- **[Novel Metrics]:** Introduction of RR, AE, and CREI enables standardized, multi-dimensional assessment of robustness across attacks and IPC settings. (`QXsj`, `b7pd`)

- **[Comprehensive Experiments]:** Extensive empirical studies covering multiple datasets, DD methods, and attack types, facilitating systematic evaluation and comparison of robustness. (`QXsj`, `3awv`, `b7pd`)

- **[Reproducibility & Community Impact]:** BEARD provides public code, dataset/model pools, and a leaderboard, supporting reproducibility and community benchmarking. The manuscript is clearly written and easy to follow, facilitating adoption and use by the research community. (`hV3a`, `QXsj`, `3awv`, `b7pd`)

Key responses addressing reviewers’ main concerns include:

- **[Benchmark Design & Novelty]:** BEARD introduces a unified adversarial game framework, aggregated robustness metrics, modular dataset/model pools, support for diverse attacks, and a public leaderboard. These innovations distinguish BEARD from prior benchmarks while enabling systematic, reproducible, and extensible evaluation. (`hV3a` **W1/W5**, `QXsj` **W1**, `3awv` **W1**, `b7pd` **W1/W2/Q3**)

- **[Causal Analysis of Robustness]:** We added causal analysis showing how dataset distillation methods, IPC, dataset scale, and attack types influence RR, AE, and CREI metrics, clarifying the mechanisms behind observed robustness trends and addressing concerns on metric interpretation and sensitivity. (`hV3a` **W2**, `QXsj` **W3/W4/Q2/Q3**, `b7pd` **W3/Q1/Q2**)

- **[Metrics Clarification]:** We provided detailed explanations of RR, AE, and CREI metrics, covering their computation and interpretation, and clarified their connection to the adversarial game framework and experimental setup to ensure consistency and reproducibility. (`hV3a` **W3**, `3awv` **W2/W3/Q1**)

- **[Evaluation Clarity, Reproducibility & Reliability]:** We corrected typos, standardized the notation for perturbation magnitude $\epsilon$, added sensitivity analysis of the CREI hyperparameter $\alpha$, included PGD convergence diagnostics, and addressed metric sensitivity to hardware and implementation details. These updates strengthen the clarity, consistency, and reliability of benchmark evaluation. Public code, dataset/model pools, and the leaderboard remain available. (`hV3a` **W4/Q1**, `QXsj` **W2/Q1**, `3awv` **W4**, `b7pd` **W4**)

---

> ### Author Response · Authors · 2025-12-01
> **Summary of Revisions**
>
> **Summary of Revisions:**
>
> Since the discussion phase allows paper updates, we summarize all revisions made in response to reviewers' feedback:
>
> - **Expanded Evaluation and Benchmark Coverage.** To address concerns regarding scalability and coverage (`hV3a` **W1**, `QXsj` **W1**, `b7pd` **W1/W2/Q3**), we extended BEARD with **additional attacks**, **datasets**, **DD methods**, and **adversarial training strategies**. These revisions broaden the benchmark’s coverage across dataset scales, attack types, and model architectures, enabling more reliable and fair comparisons. Related revisions are provided in *Appendix A.1.1 and Appendix B.7.1–B.7.3*.
>
> - **Causal Analysis of Robustness.** In response to requests for deeper understanding of robustness trends (`hV3a` **W2**, `QXsj` **W3/W4/Q2/Q3**, `b7pd` **W3/Q1/Q2**), we added **causal analysis** explaining how DD methods, IPC, dataset scale, and attack types influence RR, AE, and CREI metrics. These revisions clarify the mechanisms behind improved robustness and address concerns on metric interpretation and sensitivity. Revisions can be found in *Sections 5.1–5.3 and Appendix B.9*.
>
> - **Clarification of Metrics.** In response to feedback on metric clarity and reproducibility (`hV3a` **W3**, `3awv` **W2/W3/Q1**), we provide **detailed explanations of RR, AE, and CREI**, including their computation and interpretation. We also clarify the connection between formal definitions, the adversarial game framework, and the experimental setup to ensure consistency and interpretability. These metrics already capture aspects that existing metrics cannot and provide a more comprehensive assessment of robustness. In future work, we aim to further enhance them to better reflect both worst-case and multi-attack performance. Related revisions can be found in *Remarks 3.6, 3.11, Appendix B.8.1 and Appendix C.1*.
>
> - **Novelty and Distinction from Prior Work.** To address concerns regarding BEARD’s distinction from existing benchmarks (`hV3a` **W5**, `3awv` **W1**), we clarified its **theoretical** and **engineering contributions**. On the theoretical side, BEARD introduces a unified adversarial game framework and aggregated robustness metrics, enabling systematic evaluation of models trained on distilled datasets. On the engineering side, it provides modular dataset/model pools, support for diverse attacks, and a leaderboard, addressing prior benchmarks’ limitations and enabling reproducible, extensible evaluation. These revisions are detailed in *Appendix B.5*.
>
> - **Clarity, Reproducibility, and Evaluation Reliability.** To ensure stability and reliability of the evaluation (`hV3a` **W4/Q1**, `QXsj` **W2/Q1**, `3awv` **W4**, `b7pd` **W4**), we corrected typos, standardized the notation for perturbation magnitude $\epsilon$, provided a sensitivity analysis of the CREI hyperparameter $\alpha$, added PGD convergence diagnostics, and addressed metric sensitivity to hardware and implementation details. These updates further enhance the clarity, consistency, and interpretability of the benchmark. Public code, dataset/model pools, and the leaderboard remain available to support reproducibility and community adoption. Related revisions are included in *Appendix B.8.1, Appendix B.8.2, and Appendix B.10*.
>
> ---
>
> We reaffirm that all revisions focus on broadening benchmark evaluation, clarifying metrics, providing causal analysis of robustness trends, highlighting BEARD’s theoretical and engineering contributions, and enhancing clarity, reproducibility, and evaluation reliability as requested by reviewers. **No new methodologies were introduced, no claims were altered, and the core theory and experiments remain unchanged.** All revisions are moderate, reviewer-driven, and aimed at improving clarity, completeness, and the benchmark’s usability.
>
> ---
>
> We would like to express our sincere gratitude once again to the reviewers and the Area Chair for their time, careful reading, and constructive feedback. Their guidance has been invaluable in enhancing the clarity, completeness, and usability of BEARD, and we greatly appreciate their support throughout the discussion phase.

---

### Meta-Review · Area_Chair_DDov · 2026-01-03

**Summary:**

This work introduced BEARD, a benchmark framework to evaluate the adversarial robustness of models trained on distilled datasets. The framework evaluate diverse adversarial attacks on common image classification datasets. It newly introduced two evaluation metrics, Robustness Ratio, Attack Efficiency Ratio, and Comprehensive Robustness-Efficiency Index. The extensive experiments demonstrate dataset distillation improve adversarial robustness.

The paper studied an important problem of robustness evaluation of dataset distillation models. It built a comprehensive framework to conduct the evaluation and proposed new metrics to evaluate the results. Based on the reviewer comments and careful reading of this paper, I found several major weaknesses.

- Significance of contribution: As pointed out by Reviewer 3awv, the significance of the proposed framework is not clear. The main goal of the paper is to establish a new benchmark for robustness evaluation of dataset distillation models. However, the paper adopted common adversarial attacks for evaluation, raising a concern of why not using existing adversarial robustness benchmarks. Although the proposed metrics are closely relevant to the dataset distillation methods, a more simple way is to integrate them with existing benchmarks rather than building a new one. In this sense, I'm not convinced by the authors the significance of building such a new benchmark. Besides, the experimental results did not reveal any new findings that are specifically relevant to the proposed framework and metrics. So I think using an existing benchmark with minor development can support this research.
- The evaluation is not comprehensive: The reviewers raised multiple concerns regarding the experiments, including more datasets, different attacks, and perturbation noises. The authors have made an effort to provide more results in the rebuttal. However, I believe the benchmark can be more strengthened with a careful consideration of the evaluation purpose. If you want to evaluate the worst-case robustness, using the strongest attacks is sufficient, such that some weak attacks like FGSM are not needed. And more strong attacks including black-box attacks should be considered. If you simply want to build a comprehensive benchmark, the question is why you need such a benchmark (See the first concern).
- Findings: One of the key findings is that dataset distillation can improve adversarial robustness. But this is somewhat unexpected since adversarial robustness requires more data. It raised a concern whether the authors correctly evaluate their robustness.

Overall, based on the mixing ratings and some of the concerns above, I would recommend rejection.

**Reviewer Concerns:**

I think most of the concerns raised by the reviewers are addressed by the rebuttal, especially by the additional experiments. The authors have made an effort to provide more results on different datasets, attacks, and hyperparameters. However, I think some of the concerns regarding contributions and novelty are not fully addresses. See the Summary for details.

**Reviewer Scores:**

The paper initially received the ratings of (6,4,6,4). During the early stage of discussion, some reviewers engaged in the discussion. However, they did not improve their ratings. Therefore, I think the reviewers would not change their scores due to some unsolved concerns.

---

### Decision · Program_Chairs · 2026-01-26

Reject